# Tracking lake drainage events and drained lake basin vegetation dynamics across the Arctic

Yating Chen [1,2,3] ✉, Xiao Cheng[2,4] ✉, Aobo Liu [1,2,3] ✉, Qingfeng Chen[1] & Chengxin Wang[1,5]

Widespread lake drainage can lead to large-scale drying in Arctic lake-rich areas, affecting hydrology, ecosystems and permafrost carbon dynamics. To date, the spatio-temporal distribution, driving factors, and post-drainage dynamics of lake drainage events across the Arctic remain unclear. Using satellite remote sensing and surface water products, we identify over 35,000 (~0.6% of all lakes) lake drainage events in the northern permafrost zone between 1984 and 2020, with approximately half being relatively understudied non-thermokarst lakes. Smaller, thermokarst, and discontinuous permafrost area lakes are more susceptible to drainage compared to their larger, non-thermokarst, and continuous permafrost area counterparts. Over time, discontinuous permafrost areas contribute more drained lakes annually than continuous permafrost areas. Following drainage, vegetation rapidly colonizes drained lake basins, with thermokarst drained lake basins showing significantly higher vegetation growth rates and greenness levels than their non-thermokarst counterparts. Under warming, drained lake basins are likely to become more prevalent and serve as greening hotspots, playing an important role in shaping Arctic ecosystems.

The Arctic has been warming almost four times faster than the rest of the world[1], posing great challenges to ecosystem stability and the well-being of indigenous communities and wildlife[2,3]. Lakes, an essential part of Arctic ecosystems, play a key role in the carbon cycle and regional energy balance through their life cycle of initiation, expansion, drainage and re-initiation[4–7]. Satellite observations[8–10] indicate that Arctic lake-rich areas (areas with at least 5% lake coverage) have experienced a decline in lake area over the last 20 years, contrary to model projections[11,12] that suggested an increase due to widespread permafrost thaw. This indicates that the area of land exposed through lake drainage currently surpasses the area of water gained through

lake initiation and expansion, enhancing the dominance of drained lake basins (DLBs).

Lakes are natural sources of methane emissions and are highly sensitive to climate change[13,14]. Lakes within permafrost regions are susceptible to drainage through degradation of the surrounding and underlying permafrost[4,15]. The transition from lakes to DLBs exposes fresh land surfaces for permafrost aggradation and tundra vegetation colonization through reduced water storage[16–18]. This transition not only significantly reduces carbon fluxes from DLBs but also increases permafrost carbon sequestration, potentially transforming them into net carbon sinks[19,20] and diversifying the habitat

[1]College of Geography and Environment, Shandong Normal University, Jinan 250014, China. [2]Key Laboratory of Comprehensive Observation of Polar Environment (Sun Yat-sen University), Ministry of Education, Zhuhai 519082, China. [3]College of Global Change and Earth System Science, Beijing Normal University, 100875 Beijing, China. [4]School of Geospatial Engineering and Science, Sun Yat-sen University, and Southern Marine Science and Engineering Guangdong Laboratory (Zhuhai), Zhuhai 519082, China. [5]Key Research Institute of Yellow River Civilization and Sustainable Development & Yellow River Civilization by Provincial and Ministerial Co-construction of Collaborative Innovation Center, Henan University, Kaifeng 475001, China. ✉e-mail: chenyt2016bnu@gmail.com; chengxiao9@mail.sysu.edu.cn; lab2016bnu@foxmail.com

mosaic in the northern permafrost zone[4]. Monitoring lake drainage and subsequent DLB evolution in Arctic and Boreal regions through detailed observations can provide valuable insights for studies of permafrost ecosystem characteristics, vegetation dynamics and carbon storage[21–23].

Remote sensing is a valuable tool for large-scale monitoring of lake dynamics, providing essential data support for analyzing surface changes related to lake drainage processes and DLB evolution[9]. Existing remote sensing studies on lake drainage mostly focus on a limited area within the northern permafrost zone[24–30], which hinders their capacity to provide a holistic understanding of the distribution patterns and underlying drivers of drained lakes due to the presence of spatial heterogeneity. Webb et al.[8] conducted an analysis of surface water trends at a 12-km pixel resolution and found that the drying trend in lake area across Arctic lake-rich areas was correlated with increasing annual air temperatures and autumn rainfall. However, the spatio-temporal distribution of specific lake drainage events that contribute to the observed drying trend remains unclear, which is crucial for understanding the transition dynamics from lakes to DLBs and accurately identifying the climatic and landscape attributes associated with lake drainage.

Lake drainage is a complex process influenced by various factors including surrounding permafrost properties, lake characteristics, topography, climate and human activity[4,15,31]. Lakes across the northern permafrost region can be divided into thermokarst and non-thermokarst lakes, depending on their origin[4,32]. Thermokarst lakes develop in ice-rich permafrost zones through the thawing of sediments, melting of ice wedges, and localized ground subsidence[15,33,34], while non-thermokarst lakes form by accumulating water in pre-existing topographic depressions without significant thaw-induced subsidence. Compared to non-thermokarst lakes surrounded by ice-poor permafrost, thermokarst lakes usually undergo faster lateral erosion and bottom talik (unfrozen ground) development[31,35], making them more likely to drain[4]. Regional studies have shown that after lake drainage, the tundra vegetation growing in the moist and nutrient-rich sediments of thermokarst DLBs may be more luxuriant than in surrounding areas[16,17]. Hence, distinguishing between thermokarst and non-thermokarst lakes is essential to understand their distinct drainage processes, evolution trajectories and vegetation dynamics, as well as to investigate the differential impacts of climate change on them.

In this study, we ask the questions: How many lakes in the circum-Arctic permafrost region are experiencing drainage, and what are the key driving factors behind this phenomenon? After drainage events in these lakes, what are the growth dynamics of vegetation in the DLBs, and which factors affect vegetation greenness levels? To answer these questions, we conduct a comprehensive analysis of lake drainage events across the northern permafrost zone over a span of three and a half decades (1984–2020). By leveraging existing surface water products[36,37] and satellite remote sensing, we accurately detect over 35,000 lake drainage events and are able to delineate their spatial distribution and determine the corresponding drainage years. We distinguish between thermokarst and non-thermokarst lakes based on the published thermokarst lake coverage dataset[33]. We find that thermokarst lakes are more prone to drainage compared to non-thermokarst lakes, but they contribute a comparable amount to all drainage events. We further track the year-to-year dynamics of vegetation greenness in DLBs following lake drainage based on the detected drainage years for individual lakes, and conduct spatio-temporal analyses to examine the variations. We find that there is a significant difference in the vegetation dynamics between thermokarst and non-thermokarst DLBs. We construct machine learning models to quantify the impact of environmental factors on predicting lake drainage and vegetation greenness in DLBs (details in Methods).

## Results and discussion
### Spatial patterns of drained lakes in the northern permafrost zone

Using an object-based image analysis approach with well-established surface water products[36,37], we delineated lake objects in the northern permafrost region and identified those that experienced varying degrees of water loss during 1984–2020. Unlike the pixel-based analysis of surface water dynamics, our lake-object-based analysis allows for the identification of specific lake drainage events, enabling us to calculate regional lake drainage probabilities and provide more detailed insights into the primary drainage years and post-drainage vegetation dynamics for individual lakes. A total of 35,337 lake drainage events were detected, the largest of which had an area of approximately 6000 ha. Here, lake drainage events refer to those that involve lakes with an initial area of >1 ha and a loss of >50% of the original lake area. By continuously monitoring the dynamics of each lake using Landsat time series images, we are able to identify the main years of drainage and track vegetation growth dynamics in DLBs (Fig. 1a–c shows an example).

The distribution of drained lakes appears to exhibit spatial clustering, with a concentration in coastal lowlands and river delta areas (Supplementary Figs. S1–S3). These are generally lake-rich regions, as lakes and DLBs account for 21% of the northern permafrost regions and 49% of the lowland permafrost regions[4,33]. Typically, lake drainage is thought to be dominated by thermokarst lake processes[4], which cover about 20% of the continuous permafrost zone[33]. However, we have found that only about half of the drained lakes are situated within thermokarst lake landscapes (Fig. 1d), with the other half having non-thermokarst origins. The number of small (1–10 ha), medium (10–100 ha), and large (>100 ha) drained lakes accounted for 83.5%, 15.1% and 1.4% of the total, respectively, with a cumulative drainage area share of 23.5%, 40.9% and 35.6%, respectively. Despite the relatively small number of medium and large drained lakes, they account for a relatively large proportion of the total lake drainage area. Surface water in permafrost regions is critical to northern communities, with lakes and DLBs being a focus of local agriculture, livestock, and industrial activities[4]. Therefore, exploring the causes of drainage in large lakes is of great value for the effective management and conservation of Arctic water resources.

We used two metrics to calculate the density of drained lakes: spatial density (number of drained lakes per unit area) and lake-wise density (calculated as the ratio of drained lakes to the total number of lakes in the region). These metrics aim to illustrate the spatial distribution patterns and likelihood of lake drainage events in different regions. Based on the ecoregion delineation[38] (Supplementary Fig. S4 and Supplementary Table S1), we categorized the region and performed a statistical analysis of both the count and lake-wise density of drained lakes, revealing notable spatial variability in the distribution of lake drainage events (Supplementary Table S2). For instance, along the southern permafrost margin at the Russia-Mongolia border, this area is densely populated with thousands of drained lakes (Fig. 1e), even though it is not situated within the lake-rich areas. Here, the lake-wise density reaches nearly 8%, which is more than ten times higher than the average level across the entire permafrost zone. In contrast, in the Northeast Siberian coastal tundra ecoregion (Fig. 1f), although the overall lake-wise density of drained lakes is lower at one-third of the study area average, large drained lakes have a lake-wise density twice the average, leading to more than half of the drained lakes being medium to large in size. In addition, we have discovered previously underreported spatial clusters of drained lakes, such as St. Lawrence Island in the Bering Sea, where the spatial density of drained lakes is approximately 80 times that of the entire permafrost zone, with 655 drained lakes densely distributed over an area of 4640 km$^2$ (Fig. 1g).

We grouped drained lakes based on published geospatial datasets of the Yedoma region[39], permafrost extent[40], ground ice content[41] and

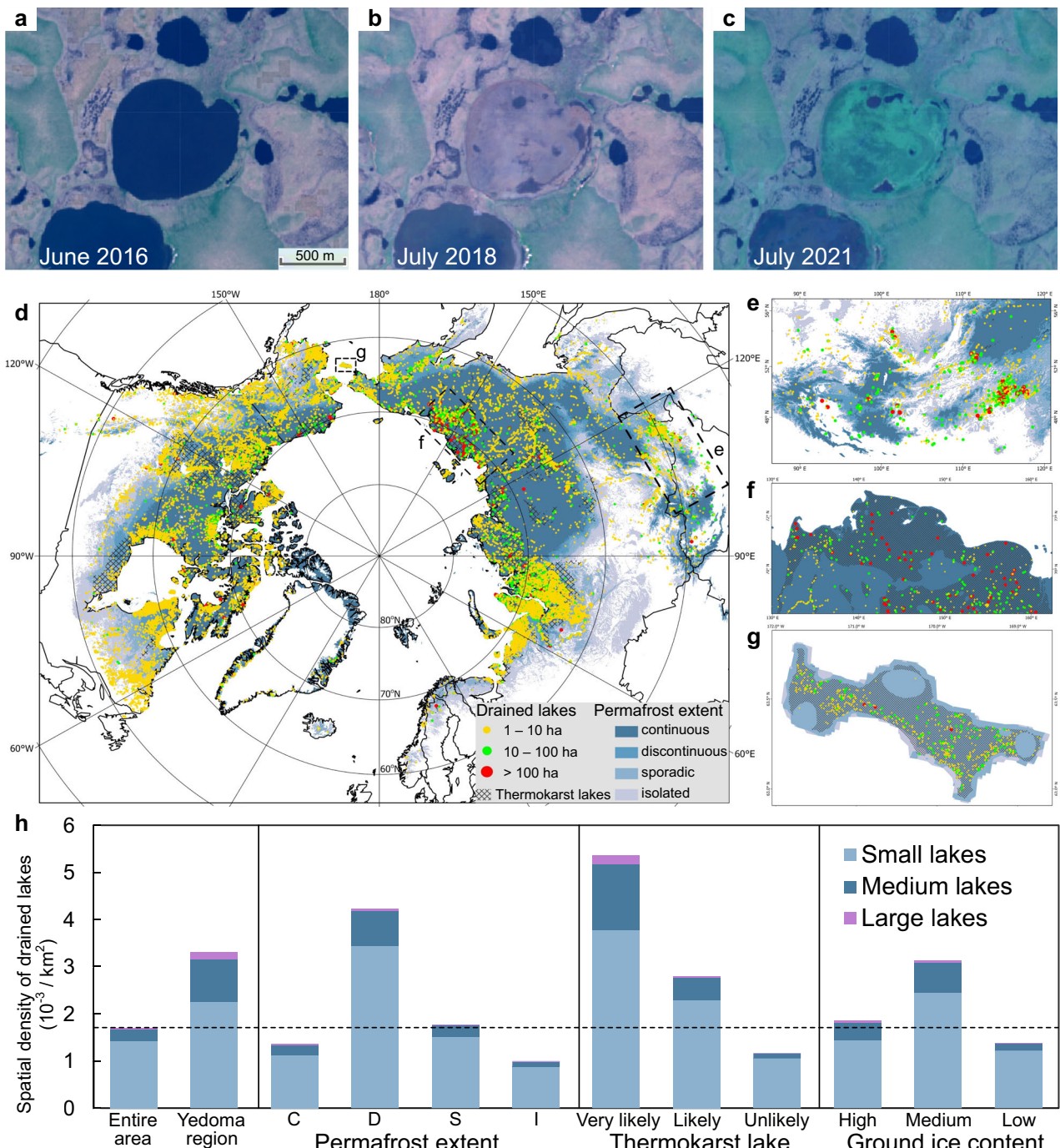

**Fig. 1 | Spatial distribution of lake drainage events detected by remote sensing in the northern permafrost zone. a–c** Satellite images of an Arctic lake at 164°45′ W, 66°27′ N showing the occurrence of a lake drainage event and subsequent vegetation growth. **d** Map of lake drainage events (*n* = 35,337) in the northern permafrost region during 1984–2020, categorized by initial lake size (small, medium, and large). The bottom map illustrates the permafrost zonation, with thermokarst lake landscape highlighted. **e–g** Enlarged maps illustrating the diversity of drained lake distribution in different regions: southern permafrost margin at the Russia-Mongolia border, Northeast Siberian coastal tundra, and St. Lawrence Island. **h** Spatial density of drained lakes in relation to permafrost extent, thermokarst lake likelihoods, ground ice content, and the Yedoma region. Yedoma is an organic-rich, ice-rich Pleistocene-age permafrost found primarily in eastern Siberia, Alaska, and the Yukon. The dashed line shows the average reference for the entire study area. C: continuous, D: discontinuous, S: sporadic, I: isolated. Refer to Supplementary Fig. S5 for classification distribution patterns and Supplementary Fig. S6 for the proportion of drained lakes of different sizes in each region.

thermokarst lake coverage[33] (Supplementary Fig. S5), and calculated their spatial density (Fig. 1h). On average, there have been approximately $1.7 \times 10^{-3}$ lake drainage events per km² in the circum-Arctic permafrost region. The Yedoma region, discontinuous permafrost zone, and lakes classified as very likely to have a thermokarst origin (hereafter referred to as 'very likely thermokarst lakes') are densely

drained, with spatial densities approximately 2, 2.5, and 3 times higher than average, respectively (Fig. 1h). Drained lakes in the northern permafrost zone can be divided into two primary categories: lateral drainage and internal drainage[34]. In the continuous permafrost zone, lake drainage often results from lateral expansion of lakes into low-lying areas, driven by mechanisms such as thawing of ice barriers,

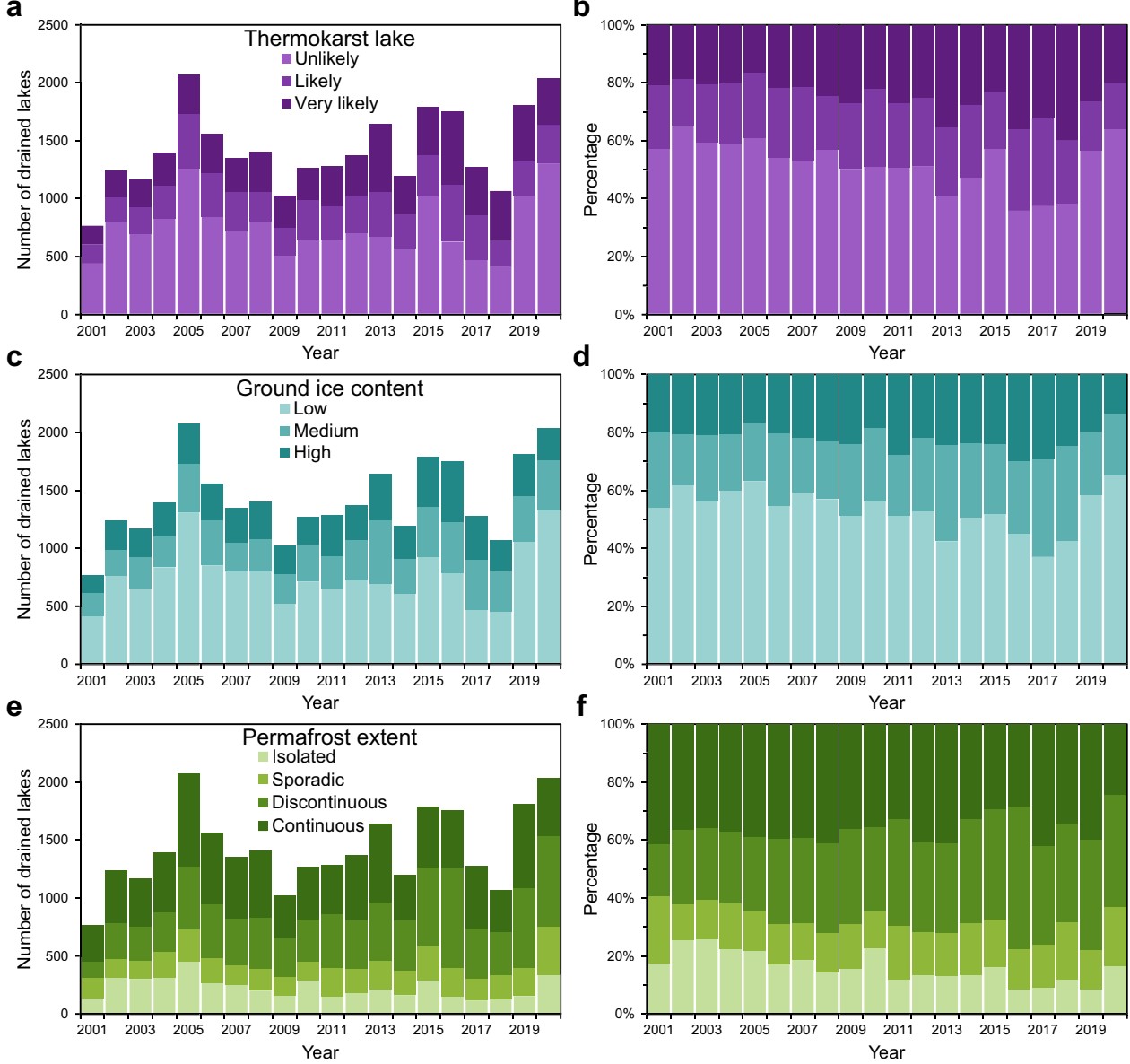

**Fig. 2 | Annual variation in the frequency of lake drainage event during 2001–2020.** Stacked and percentage stacked bar charts illustrating the frequencies of lake drainage events over time under different classifications of (**a**, **b**) thermokarst lake likelihoods, (**c**, **d**) ground ice contents, and (**e**, **f**) permafrost extents. The occurrence year of a lake drainage event was determined as the year with the maximum drainage proportion.

headward stream erosion, coastal erosion, and bank overtopping due to rapid snowmelt, extreme precipitation, and flooding[30,42,43]. While in the discontinuous permafrost zone, in addition to lateral drainage, internal drainage related to ice wedge degradation frequently occurs, where taliks or thawed zones beneath lakes penetrate the permafrost, allowing for drainage subterraneously[24,34,44]. Drained lakes are less spatially dense in areas that are ice-poor and unlikely to form thermokarst lakes (hereafter referred to as 'unlikely thermokarst lakes'), where lake abundance is relatively low and limited ice wedge melting does not readily create drainage channels[4].

## Temporal patterns in lake drainage events for 2001–2020

Based on the identified lake drainage events, we employed a temporal segmentation and change detection algorithm[45,46] to determine the primary years of drainage for individual lakes. We identified 6858 and 28,479 lake drainage events for the periods 1984–2000 and 2001–2020, respectively. However, the fractured nature of Landsat observations across space and time[36] prior to 2000 limited the

detection of lake drainage events that occurred during 1984–2000. Therefore, in this study we only analyzed changes in the frequency of lake drainage events over the period 2001–2020 (Fig. 2). The results showed that the mean annual count of drained lakes in the northern permafrost region was 1424 (range: 767–2073), with a standard deviation of 332. Despite annual fluctuations, the lake drainage frequency exhibited a slight upward trend (slope of 23; $p = 0.08$; unpaired two-tailed Student's $t$ test) throughout the study period.

Regional statistics show that temporal peaks in lake drainage frequency are not synchronized across landscape attributes (Fig. 2). For example, in 2020, the number of drained lakes peaked in areas with low ground ice content (1329) and for unlikely thermokarst lakes (1305), while drained lakes in other categorized regions remained close to the annual average level. In contrast, in 2016, likely (495) and very likely (630) thermokarst lakes had a peak drainage frequency, while the number of unlikely thermokarst drained lakes was below the annual average. The elevated count of drained lakes in these specific years suggests the potential prevalence of triggering factors for lake

drainage. Thermokarst and non-thermokarst lakes making distinct contributions to total drainage in various years. Despite comprising over half of all drained lakes (Fig. 2c), non-thermokarst lakes have garnered relatively less research attention[4]. The prevailing emphasis on thermokarst-related phenomena[7,11,33] has led to a comparative scarcity of studies exploring the characteristics, dynamics, and ecological implications of these non-thermokarst lakes[4,28,47]. Given their substantial number and potential contributions to our understanding of diverse ecosystems, there exists a compelling need for future investigations to delve into the unique dynamics and roles of these less explored lake types.

From 2001 to 2020, lake drainage events exhibited significant upward trends ($p < 0.001$) in discontinuous permafrost zones and very likely thermokarst lakes, with slopes of 21 and 14, respectively. This indicates that over the 20-year period, the mean annual number of drained lakes increased by 420 in discontinuous permafrost zones and 280 for very likely thermokarst lakes. Over time, the contribution of the discontinuous permafrost zone to the annual count of drained lakes gradually surpasses that of the continuous permafrost zone (Fig. 2f), suggesting that the impacts of climate change and permafrost degradation on the stability of lakes in these sensitive regions appear to be intensifying[24,34,42]. Permafrost acts as a barrier to water exchange between surface and groundwater systems. In continuous permafrost regions, suprapermafrost groundwater often accumulates above the permafrost layer within the active layer, including closed subaerial and open subaqueous taliks[4]. In discontinuous permafrost regions, where permafrost continuity is lacking, direct connections between surface and groundwater systems can occur, influencing the hydrological dynamics of lakes[24,44]. Therefore, lake drainage events are influenced by both the long-term legacy of permafrost characteristics, such as permafrost properties, distribution, and degradation extent, as well as short-term environmental features, including active layer dynamics, ground thermal conditions, and variations in water supply[4].

## Environmental drivers of Lake drainage events

To reveal the key environmental drivers behind lake drainage events from a variety of candidate explanatory variables such as climate, topography, and permafrost characteristics (Supplementary Table S3), we developed a binary classification model to predict lake drainage. We conducted a diagnostic evaluation of the lake drainage prediction model and obtained an area under the curve (AUC) value of 0.92 and an average precision (AP) value of 0.88 (Supplementary Fig. S7). The results showed that the model has a strong classification ability and can effectively use explanatory variables to predict whether or not individual lakes will be drained.

Further feature importance assessment indicates that air temperature and elevation are important drivers of lake drainage events (Fig. 3a). The effect of annual air temperature trend on lake drainage events is nonlinear (Fig. 3b), with the primary clusters between 0–0.04 °C/year inhibiting lake drainage, and secondary clusters near 0.1 °C/year promoting lake drainage. The relationship between annual air temperature and the prediction of lake drainage demonstrates an almost monotonically increasing trend (Fig. 3c). As the annual air temperature rises, the risk of lakes experiencing drainage events also increases. This can be attributed to various mechanisms, including the thermal erosion of permafrost around lakes and the formation of drainage channels due to melting of ground ice[27,31,48]. Development of terrestrial taliks is likely to result in widespread lake drainage when mean annual air temperature approaches 0 °C[43]. In addition to promoting lake drainage, this warmer regime may adversely affect permafrost aggradation on newly exposed surfaces, inhibit ground ice accumulation, and bring about substantial changes in the landscape conditions of the DLB system[4].

Increasing elevation has an overall negative impact on the likelihood of lake drainage (Fig. 3d), with the primary cluster located in the range of 0–150 meters above sea level. We found that this lowland area covers approximately 29.6% of the northern permafrost zone, yet contributes to about 57.1% of the drained lakes, with both spatial and lake-wise densities of drained lakes exceeding the average level (Supplementary Fig. S8). We examined the impact of active layer depth on lake drainage (Fig. 3e) and observed a consistent rise in lake drainage likelihood as the active layer depth increases. Specifically, when the active layer depth in the area exceeds 0.6 meters, the likelihood of lake drainage tends to be higher than the average level. This may be attributed to increased erosion and other thermal processes, such as the formation of taliks at the lake bottom[47,49]. The mechanisms and factors driving lake drainage are highly diverse, and with future Arctic warming and permafrost degradation, the frequency of lake drainage will likely increase[20,25,43].

## Lake-wise density of drained lakes

For categorical features such as permafrost extent, ground ice content, and thermokarst lake likelihood, their relative importance may be underestimated due to the fact that they are treated as time-invariant attributes and the imbalanced distribution of categories. Given the known importance of these landscape variables in driving surface water changes associated with permafrost degradation, we conducted an enumeration of all lake objects within our study area and calculated lake-wise density (the ratio of drained lakes to the total number of lakes in the region) for various sizes of drained lakes and geographical regions (Fig. 4 and Supplementary Table S2). A total of approximately 5.83 million lakes (with an area larger than 1 ha) were identified across the northern permafrost region. The average lake-wise density of drained lakes in the study area was 0.61%, with 0.64%, 0.50% and 0.38% for small, medium and large lakes, respectively, indicating a relatively lower likelihood of drainage for larger lakes. One potential explanation for this is that smaller lakes, due to their shallower water columns and lower water storage capacity[29], are more susceptible to lateral drainage events triggered by extreme precipitation or rapid snowmelt[43,50]. In contrast, larger lakes have the capacity to accumulate more heat and water, thus showing greater resilience against hydrological disturbances[28]. This provides a thermal inertia and water storage buffer when temperatures and precipitation fluctuate, enabling larger lakes to better maintain thermal equilibrium and shoreline stability compared to smaller water bodies.

Significant spatial heterogeneity in lake-wise density of drained lakes has been observed across ecoregions (Supplementary Table S2), highlighting the complexity of lake drainage dynamics. For instance, within the Canadian Aspen forests and parklands (spanning $1.6 \times 10^4$ km²), 16.33% of lakes experienced drainage, while in the Central Canadian Shield forests (spanning $16.5 \times 10^4$ km²), only 0.11% of lakes underwent drainage. This diverse lake-wise density of drained lakes underscores the intricate interplay of local environmental factors. Further analysis indicates that lake-wise density of drained lakes is associated with thermokarst lake likelihoods, permafrost extent, and ground ice content (Fig. 4). Notably, for very likely thermokarst lakes and the Yedoma region, the lake-wise density for drained lakes of all sizes exceeds the regional averages, with large lakes anomalously having higher lake-wise density than small and medium-sized lakes. Large drained lakes are disproportionately concentrated in these areas, with 6–8 times the average spatial density (Fig. 1h) and 3–4 times the average lake-wise density (Fig. 4b, d), suggesting well-developed erosional drainage systems may have formed. The prevalent landscape in these areas is a mosaic of lakes and streams, interconnected through an underground network of ice[15,42]. When expanding lakes encounter the ice network, thermal erosion along the network results in the melting of ice wedges, creating drainage channels through which the lake water drains[7,34].

The lake-wise density for likely thermokarst lakes fall between those for very likely and unlikely thermokarst lakes (Fig. 4d), consistent

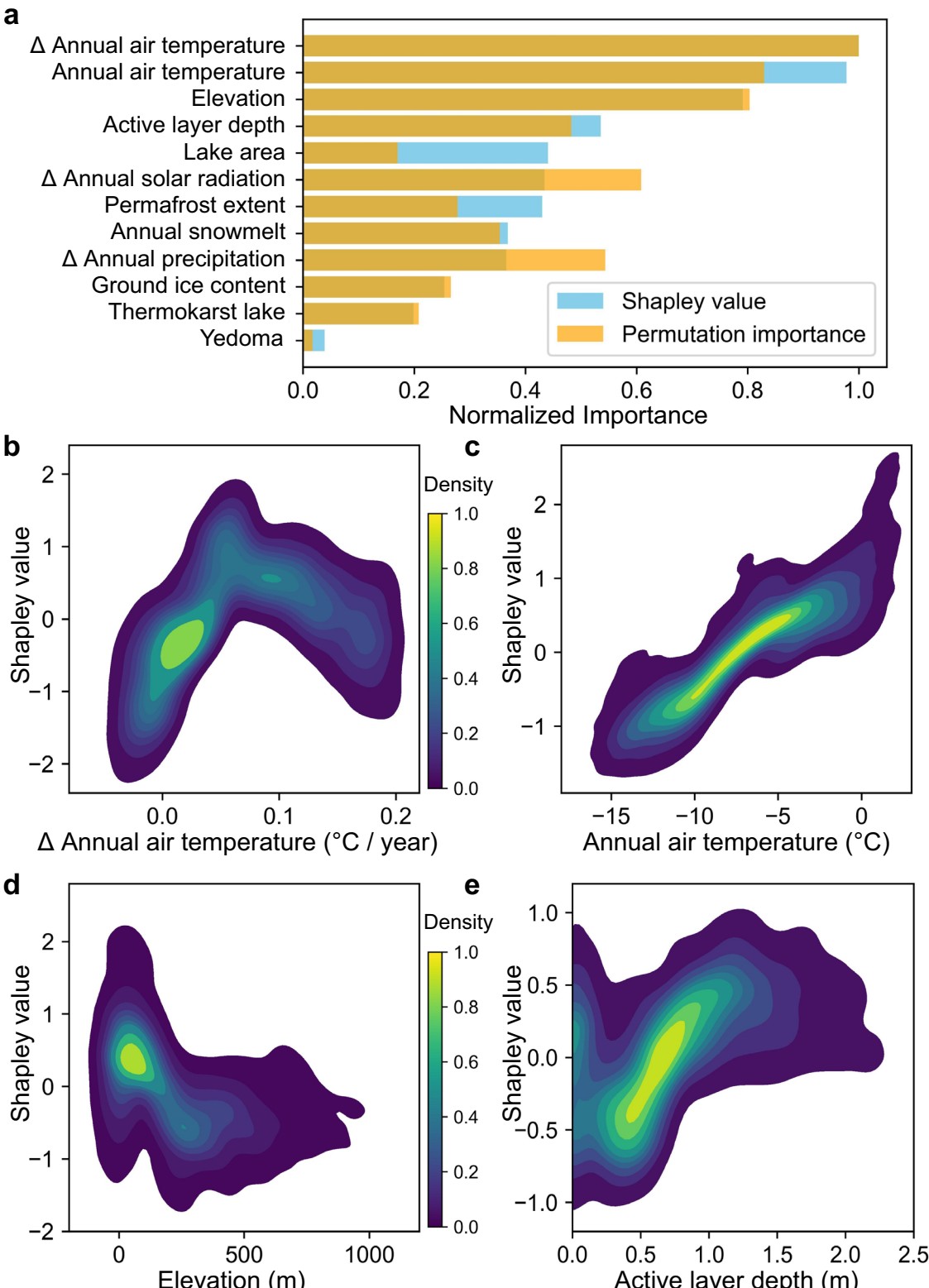

**Fig. 3 | Identification of key environmental factors driving lake drainage. a** The relative importance of predictive variables in predicting the occurrence of lake drainage events, assessed using permutation importance and Shapley values8. The two methods are superimposed, with dark orange representing overlap. Both indicators identify slope of annual air temperature, annual air temperature and elevation as the most important predictors of lake drainage events. Kernel density estimations of Shapley value distributions for **b** slope of annual air temperature, **c** annual air temperature, **d** elevation, and **e** active layer depth. The density of sample distribution is normalized, with yellow indicating the most concentrated regions of data points. The magnitude of Shapley values reflects their impact on lake drainage predictions, with negative values representing lower likelihood. Refer to Supplementary Fig. S7 for model diagnostics.

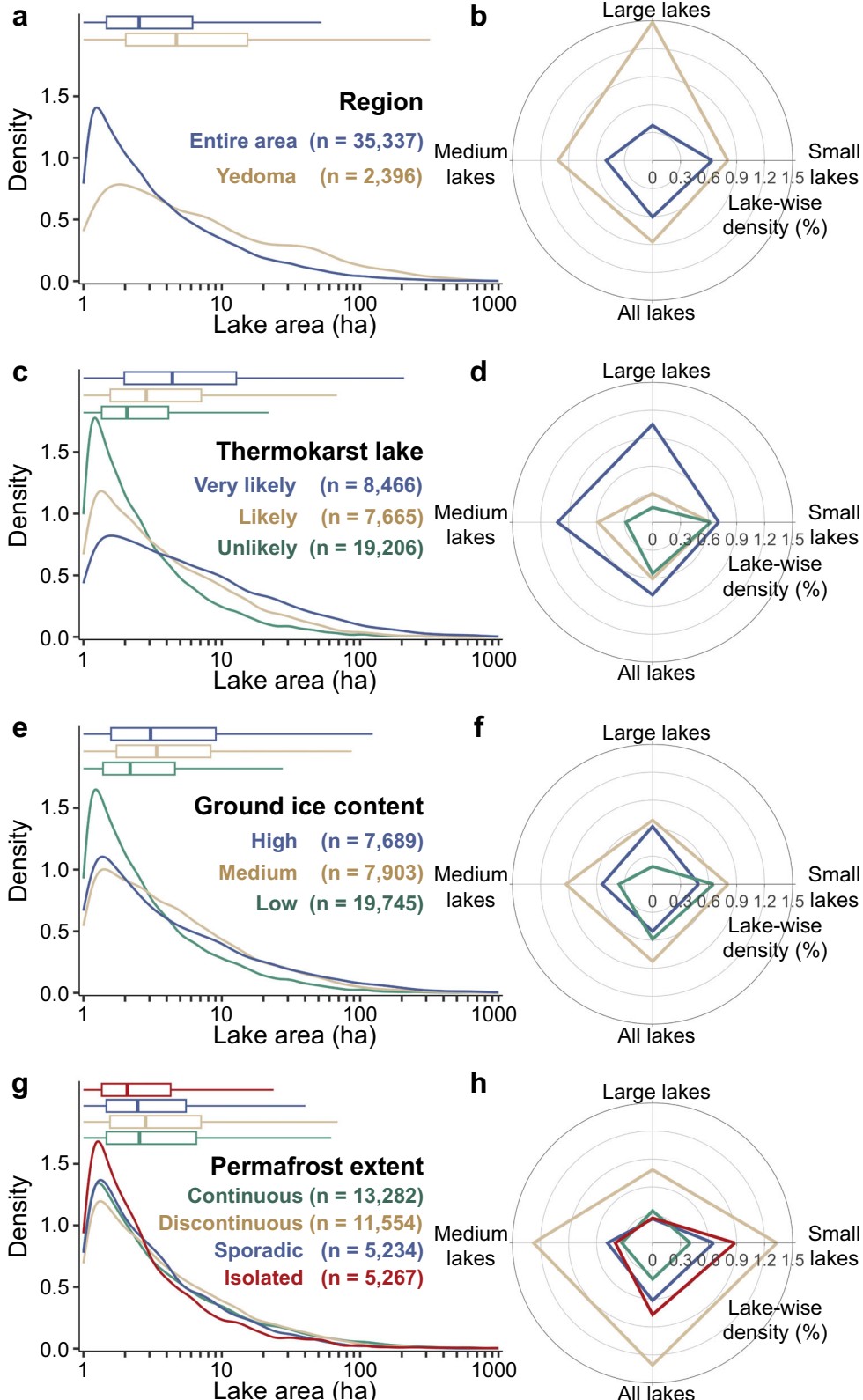

**Fig. 4 | Lake-wise density of drained lakes across various lake sizes and regions.** Analysis of drained lake area distribution and lake-wise density for drained lakes of different sizes in (**a**, **b**) the Yedoma region and the entire study area, along with different classifications of (**c**, **d**) thermokarst lake likelihoods, (**e**, **f**) ground ice contents, and (**g**, **h**) permafrost extents. The left-hand panels depict kernel density estimation plots with a logarithmic x-axis to display lake areas, along with numerical annotations denoting the number of drained lakes. Boxplots show the statistics – horizontal lines: median; boxes: interquartile range; whiskers: 1.5 times the interquartile range. The right-hand panels present radar charts illustrating the percentage of drained lakes relative to the total number of lakes within the respective area or attribute type.

with existing knowledge that thermokarst lakes are more susceptible to drainage[4]. The impact of ground ice content on lake-wise density of drained lakes is relatively intricate (Fig. 4f). In regions with medium ground ice content, the lake-wise density of drained lakes is the highest, approximately 1.4 times the average level. In regions with high and low ground ice content, the overall lake-wise density is slightly below average, while the lake-wise density for large drained lakes in these regions are 1.6 and 0.5 times the average, respectively. Variations in the lake-wise density of drained lakes under different ground ice content levels may be associated with the thermal inertia of latent heat fusion and the formation of subsurface drainage channels[28,51]. Considering that the ground ice content dataset covering the northern permafrost region has not been updated for over 20 years, and during this period, the permafrost region has undergone significant warming, we need to exercise caution regarding the timeliness of the data and its potential impact on the current research findings. In future studies, the consideration of updating the ground ice content data may aid in a more accurate assessment of the relationship between ground ice content and lake drainage probabilities[4].

Furthermore, the discontinuous permafrost zone exhibits the highest overall lake-wise density of drained lakes (more than double the average; Fig. 4h), with lake-wise density for small, medium, and large lakes at 1.33%, 1.28%, and 0.78% respectively. There are over 13,000 drained lakes in the continuous permafrost zone (Fig. 4g), but the lake-wise density in this area is lower than the average level (Fig. 4h). However, the susceptibility is rising in certain southern areas where permafrost is becoming more discontinuous due to climate warming[9,20].

## Post-drainage vegetation dynamics in DLBs

After a lake drains, plants colonize the DLB, shifting the landscape from exposed lacustrine sediments to tundra vegetation. During the initial colonization stage in the years following drainage, pioneer plant species like sedges grow quickly in DLBs due to their strong adaptability and high reproductive rates[16,19,52]. Edaphic conditions in DLBs change as the pioneer plants thrive, creating a more hospitable environment for plant succession. The early succession stage may last for decades, with a diversity of plant species beginning to colonize the DLBs, including dwarf shrubs, herbs, mosses, and lichens[16,52]. We quantified the vegetation growth dynamics in DLBs using the Normalized Difference Vegetation Index (NDVI) (Fig. 5), which is sensitive to the chlorophyll content of plants and can serve as a good indicator of tundra plant productivity and aboveground biomass[53,54]. Taking into account the potential modulation of NDVI by standing water, we examined the Tasseled Cap Greenness (TCG) Index, which is less affected by surface water than NDVI[55], finding similar patterns (Supplementary Fig. S9).

Considering data availability and vegetation growth status, we chose the tenth year after drainage as a reference point to conduct a vegetation greenness assessment within DLBs and their surrounding areas, along with an analysis of various potential influencing factors (Fig. 5a–i). The results showed that the vegetation greenness in very likely, likely, and unlikely thermokarst DLBs is higher, similar, and lower than surrounding vegetation, respectively, suggesting that vegetation greenness in thermokarst DLBs is generally greater than in non-thermokarst DLBs (Fig. 5a). Chen et al.[17] analyzed vegetation dynamics in thermokarst DLBs in northern Alaska and found that tundra vegetation growing on wet and nutrient-rich lake sediments was more luxuriant (with 0.15 or 25% higher NDVI) than in surrounding areas. Here we find that across the northern permafrost region, the NDVI of very likely thermokarst DLBs is higher by 0.06 or 10% compared to the surrounding areas, while the NDVI of unlikely thermokarst DLBs is lower by 0.09 or 15% than the surrounding areas (Fig. 5a).

We analyzed the annual differences in NDVI between DLBs and surrounding vegetation, revealing significant variations in vegetation dynamics between thermokarst and non-thermokarst DLBs (Fig. 5j). For very likely thermokarst lakes, vegetation rapidly spread and covered DLBs after drainage[16,22], leading to a rapid increase in NDVI, reaching levels similar to the surrounding areas approximately 2 years after drainage. As succession progressed, the relative abundance of high-productivity plant communities in thermokarst DLBs increased, resulting in a slow NDVI growth that reached slightly higher levels than the surrounding vegetation. In contrast, the NDVI of likely thermokarst DLBs reached a stable state similar to the surrounding vegetation around the 6th year after drainage, while the NDVI of unlikely thermokarst DLBs exhibited very slow growth, remaining significantly lower than the surrounding levels even 15 years after drainage (Fig. 5j). Note that the vegetation dynamics beyond our analysis period remain uncertain due to the limitations of observations. In the tenth year after lake drainage, the (likely and very likely) thermokarst and non-thermokarst DLBs had median NDVI values of 0.72 and 0.42, with corresponding 25%–75% ranges of 0.64–0.77 and 0.29–0.59, and 5%–95% ranges of 0.39–0.83 and 0.15–0.79, respectively.

The median areas of very likely, likely, and unlikely thermokarst drained lakes are 4.4 (25%–75% range: 2.0–12.7), 2.9 (1.6–7.1), and 2.1 (1.4–4.1) ha, respectively (Fig. 4c). Moreover, the lake-wise density of large and medium-sized thermokarst drained lakes is markedly higher than that of non-thermokarst lakes (Fig. 4d), leading to proportions of small lakes within very likely, likely, and unlikely thermokarst drained lakes of 70%, 82%, and 90%, respectively (Supplementary Fig. S6). For drained lakes of various sizes, the greenness variability of DLBs tends to be higher than that of the surrounding vegetation, and the overall trend is that the vegetation greenness of large DLBs is higher than that of small DLBs (Fig. 5b). Large DLBs, with more extensive lakebed sediment exposure, can support more diverse plant communities[4], resulting in NDVI around 0.04 higher than the surrounding areas (25%–75% range: −0.09–0.08). Additionally, vegetation greenness in DLBs is related to drainage area ratio (Fig. 5c): a higher ratio corresponds to greener vegetation than the surrounding area, while a lower ratio indicates less greenness than the surrounding area. A higher drainage area ratio implies a greater amount of exposed land for plant growth, and in these areas, earlier summer thaw and elevated concentrations of dissolved soil carbon and nitrogen create favorable conditions for promoting vegetation growth[56].

Regionally, DLBs in discontinuous permafrost regions exhibit higher greenness, while DLBs in isolated permafrost regions display lower greenness (Fig. 5d). In the tenth year after lake drainage, the median NDVI values for DLBs in Alaska, Canada, and Russia are 0.73 (25%–75% range: 0.68–0.78), 0.47 (0.28–0.70), and 0.69 (0.56–0.75), respectively (Fig. 5e). The NDVI values for DLBs in Alaska and Russia are approximately 0.03 (−0.03–0.10) and 0.02 (−0.06–0.08) higher than the surrounding vegetation, while the difference between the NDVI of DLBs in Canada and the surrounding vegetation is −0.07 (−0.17– −0.01). The NDVI of DLBs was noticeably lower in Canada, which may be attributed to the thin, rocky soils and barren complex vegetation in the Canadian Shield region[57]. These soil conditions could pose drainage challenges[58], which may lead to increased waterlogging and periodic inundation of vegetation, ultimately resulting in a lower NDVI within DLBs compared to the surrounding areas. Additionally, the proportion of small drained lakes is higher in Canada compared to other regions, potentially contributing to less lush vegetation in Canadian DLBs (Supplementary Table S2). We examined the NDVI of DLBs in flood-prone and non-flood-prone areas and did not find significant differences (Fig. 5f). Regions with higher soil carbon and nitrogen content support more vigorous vegetation growth[59], making vegetation in DLBs appear greener than the surrounding areas (Fig. 5g, h). This could partially explain why thermokarst DLBs[33] and the Yedoma region[39,60] exhibit greener vegetation compared to the surrounding areas (Fig. 5a, i).

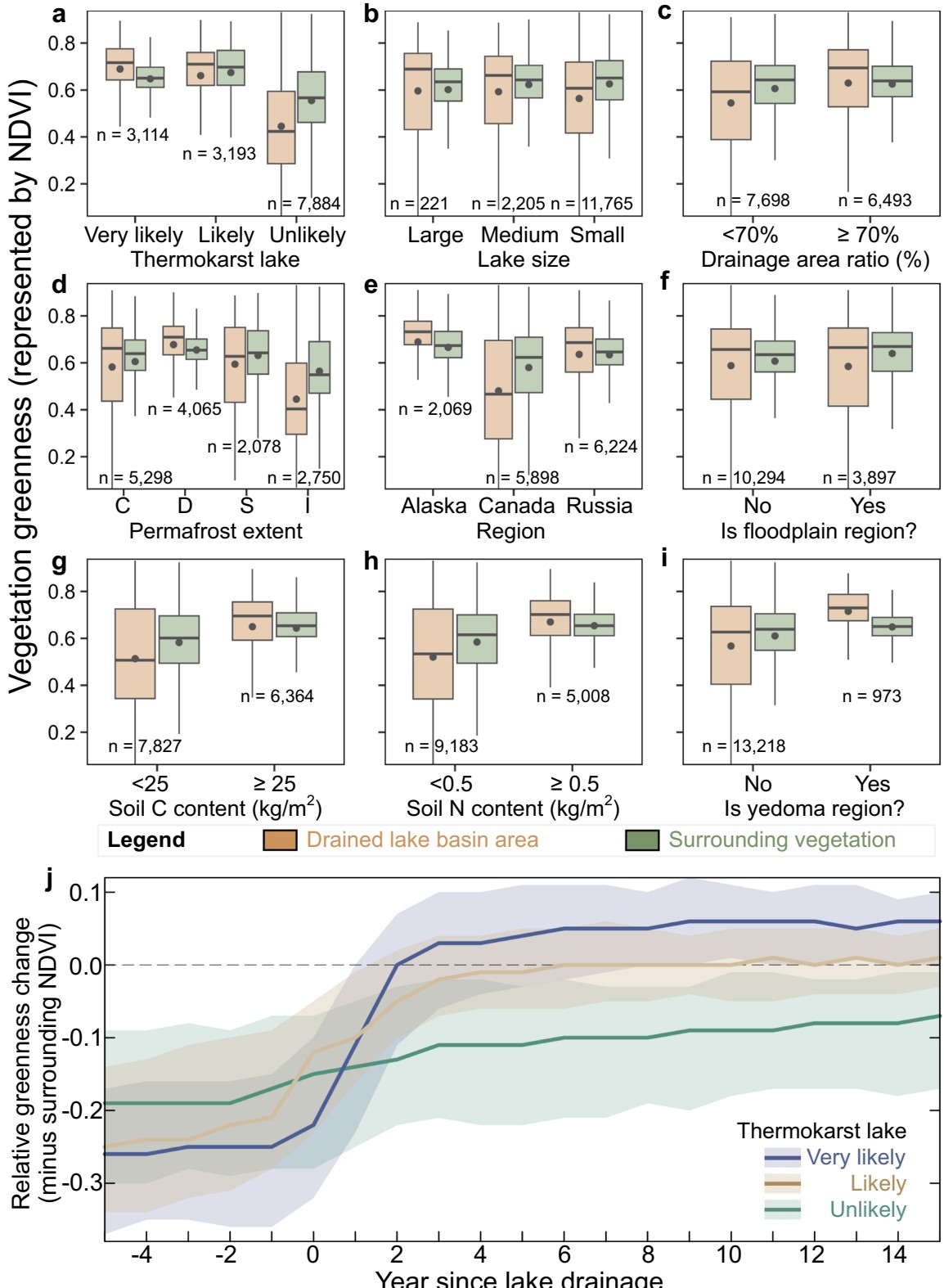

**Fig. 5 | Differences in vegetation greenness between DLBs and surrounding areas.** NDVI measured in the tenth year after lake drainage for various classifications of **a** thermokarst lake likelihoods, **b** lake sizes, **c** drainage area ratios, **d** permafrost extents, **e** regions, **f** floodplain status, **g** soil carbon contents, **h** soil nitrogen contents, and **i** Yedoma region. Boxplots show the statistics – horizontal lines: median; dots: mean; boxes: interquartile range; whiskers: 1.5 times the interquartile range. Sample sizes are indicated below each plot. **j** Time series of changes in relative greenness of very likely, likely and unlikely thermokarst DLBs, represented by NDVI differences compared to surrounding vegetation. Solid lines show median values, while shaded areas indicate upper and lower quartile ranges.

### Environmental influences on NDVI in DLBs

We used machine learning models to analyze the drivers of NDVI change in DLBs, and evaluated the relative importance of explanatory variables (Supplementary Table S4) on NDVI predictions. The diagnostic evaluation (Supplementary Fig. S10) indicate that our trained model is capable of capturing a substantial portion of NDVI variations in DLBs ($R^2 = 0.83$). The assessment of feature importance reveals that ecological zoning and the years since lake drainage are the most important influencing factors for NDVI predictions (Fig. 6a). Different ecoregions host distinct local species pools[50], leading to variations in vegetation greenness levels and successional trajectories. High spatial variability in vegetation dynamics of DLBs has been observed across

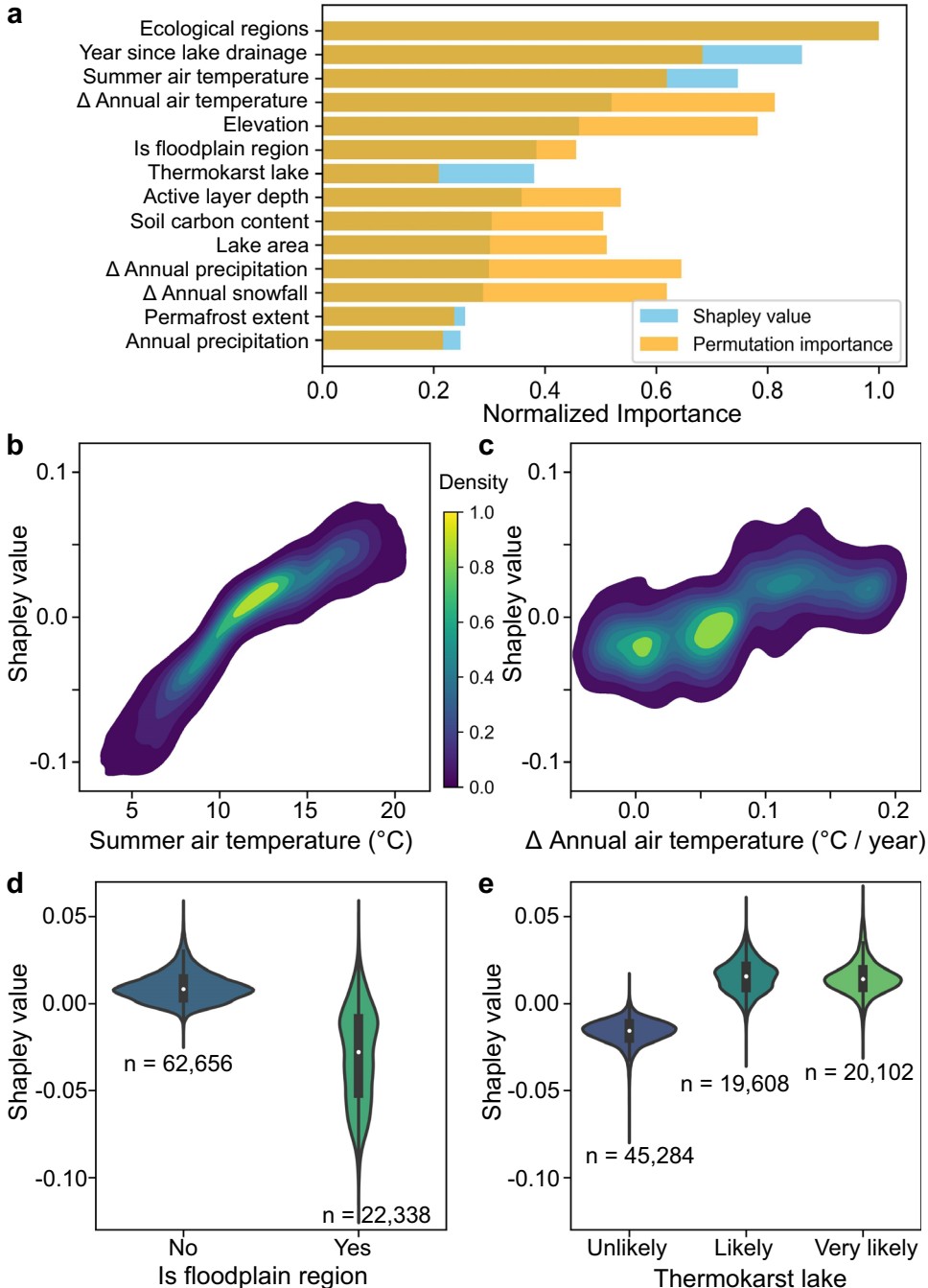

**Fig. 6 | Identification of key environmental factors driving NDVI changes in DLBs. a** The relative importance of predictive variables in explaining NDVI changes in DLBs, assessed using permutation importance and Shapley values. These two methods are superimposed; deep orange represents overlap. Kernel density estimations of Shapley value distributions for **b** summer air temperature and **c** slope of annual air temperature. The density of sample distribution is normalized, with yellow indicating the most concentrated regions of data points. The magnitude of Shapley values reflects their impact on NDVI predictions, with negative values representing lower NDVI. Violin plots displaying the distribution of Shapley values for **d** floodplain status and **e** thermokarst lake likelihoods, each incorporating internal boxplots to show the statistics – dots: mean; boxes: interquartile range; whiskers: 1.5 times the interquartile range. Sample sizes are indicated below each plot. Importantly, Shapley values can aid in model interpretation and variable selection, but they do not directly reflect the model's accuracy. The range of Shapley values differs between classification and regression models, making them non-comparable. The emphasis should be placed on analyzing the importance of variables within each model. Refer to Supplementary Fig. S10 for model diagnostics.

ecoregions, with NDVI values ranging from around 0.5 to 1.2 times the surrounding average (Supplementary Table S2). The specific years since lake drainage emerged as a key driver of NDVI predictions, as they potentially reflect the progression of ecosystem succession. This underscores the importance of accurately identifying the specific years of lake drainage events for analyzing the vegetation dynamics within DLBs.

Among the climatic variables, summer air temperature and annual air temperature trend stand out as the most important influencing factors for NDVI predictions (Fig. 6a). Their increase has an overall positive impact on NDVI within DLBs, primarily due to the extension of the growing season and the enhancement of photosynthetic efficiency[50,61]. Temperature serves as a primary constraint for the greenness of Arctic ecosystems, with vegetation greening most frequently occurring in regions where summer air temperatures and annual soil temperatures are increasing[53]. The relationship between summer air temperature and NDVI predictions displays an almost monotonically increasing trend, suggesting that warmer locations tend to show enhanced plant growth. When summer air temperatures fall below 10 °C, vegetation greenness is suppressed, evident by negative Sharpe values (Fig. 6b). The impact of the annual air temperature trend is more intricate, with two major clusters at 0.01 and 0.06 °C/year exerting negative effects on NDVI, while two minor clusters at 0.12 and 0.18 °C/year promote greening of vegetation (Fig. 6c).

Floodplain and thermokarst lake stand out as the most important categorical features for predicting NDVI (Fig. 6a). Floodplain areas and non-thermokarst lakes have an overall negative impact on predicting NDVI in DLBs, while non-floodplain areas and thermokarst lakes have a positive influence (Fig. 6d, e). These findings reveal how vegetation dynamics in DLBs are jointly influenced by species pools, vegetation succession patterns, regional climate and hydrological conditions, as well as landscape attributes. Additionally, due to permafrost aggradation and ice wedge growth, DLBs exhibit a highly heterogeneous geomorphic mosaic characterized by fine-scale variations in topography[22]. The micro-topography of DLBs effectively directs run-off, facilitates nutrient accumulation, and supports vegetation growth within DLBs[17,22]. Simultaneously, it fosters diverse tundra ecosystems within DLBs, making them ecological hotspots in the permafrost regions[4,16].

## Implications of lake drainage and vegetation growth in DLBs
In summary, our study leveraged remote sensing data to capture spatial and temporal distribution patterns of lake drainage events in the extensive northern permafrost regions, extending our comprehension of the post-drainage vegetation dynamics within DLBs. This technological advancement holds valuable implications for predicting the scale of lake drainage in permafrost regions in the 21st century. Annual air temperature and its trend, active layer thickness, and permafrost extent are key environmental drivers of lake drainage events (Fig. 3c), suggesting that with Arctic warming, deeper active layers and increased permafrost discontinuity are expected to lead to more frequent lake drainage events. Our statistical analysis of spatial and lake-wise density shows that smaller lakes, thermokarst lakes, and lakes in discontinuous permafrost areas are more prone to drainage than larger lakes, non-thermokarst lakes, and those in continuous permafrost regions (Fig. 4).

Our dataset of 35,337 detected lake drainage events can serve as a starting point for further research, such as investigating potential catastrophic flooding in DLBs[62,63] and improving simulations of permafrost hydrology linkages in Earth system models[20,64]. Analyses of mega-lake drainage events and drained lake clusters allow for more specific investigation of the role of climate change in triggering lake drainage. For example, approximately 60% of lake drainage events observed on St. Lawrence Island (Fig. 1g) have occurred since 2018, a

period characterized by historically low sea ice coverage[65] and widespread seabird mortality[66], indicating that the local climatic conditions may have reached a tipping point.

Lake drainage is essentially the abrupt thawing of permafrost intensified by climate change, reflecting permafrost degradation and instability[11,20]. Drainage events reduce the water storage capacity of lakes, impacting local hydrological conditions[67–69]. Drainage of large lakes often leads to catastrophic flooding due to the peak of snowmelt promoting the formation of ephemeral lakes and resulting in rapid and sustained flood peaks[4,63,70]. Such hydrological events can adversely affect infrastructure like Arctic roads and pipelines. Species dependent on lake habitats for survival, such as migratory birds and aquatic life, are threatened[71]. As more lakes are drained, access to clean freshwater may become even more challenging for many Arctic communities and indigenous populations[72]. Lake drainage impacts regional hydrology and ecology while also generating intricate feedbacks on carbon balance. After the lake drains, previously submerged organic matter is exposed to the atmosphere where it can be oxidized rather than consumed through microbial methanogenesis[11,12]. Climate change may exacerbate the loss of Arctic lakes, reduce lake methane emissions, and expose areas for tundra colonization and permafrost aggradation. While the overall impact of these processes on climate change is uncertain, vegetation is expected to play a significant role in many feedback mechanisms that may arise[17,73–75]. Vegetation growth in DLBs directly increases carbon sequestration, while also promotes permafrost aggradation by providing thermal insulation[23]. As a result, the carbon fluxes in DLBs are lower by 1–3 orders of magnitude compared to pre-drainage[20], and may even exhibit net carbon sinks for certain periods[19,34].

Based on a recent study[59], the average increase in NDVI in the high Arctic region was approximately 3.9% between 2000 and 2020. While in newly drained DLBs, the slope of NDVI was 1–2 orders of magnitude higher than the average level of arctic greening (Fig. 5j). The greening events observed within DLBs can be embedded within the overall greening trend of the Arctic, although they may not be the primary driver of this trend[54]. It's important to note that while NDVI typically corresponds well with vegetation characteristics, surface water can impact NDVI values[54]. Specifically, in partially drained basin areas, standing water can lead to underestimated NDVI values, despite vigorous sedge growth in developing aquatic environments. In this study, we used the 90th percentile of NDVI values from annual image collections to generate a stable vegetation greenness assessment. Subsequently, we conducted median extraction of the annual 90th percentile NDVI time series after masking water bodies within DLBs and their surrounding areas. A comparison with the results of the TCG index (Supplementary Fig. S9), which is less affected by surface water, indicates that our method effectively mitigates the impact of standing water on NDVI estimates within DLBs.

DLBs, as hotspots of Arctic greening, have profound implications for vegetation composition of tundra ecosystems and permafrost carbon feedbacks[4,43]. The clustered DLBs may yield habitats at various stages of succession, supporting ecosystems ranging from aquatic to dry tundra within basins over time, thus enhancing ecosystem diversity[4,76]. Vegetation in DLB ecosystems can alter species composition, abundance, and distribution in Arctic vegetation, thereby driving interspecies competition within tundra vegetation[17,59]. Environmental benefits from vegetation also include enhanced sediment stability, erosion prevention, improved water quality, habitat creation for wildlife, increased biodiversity, and support for indigenous pastoral activities[50]. By tracking vegetation dynamics following lake drainage, we have found that vegetation greenness in DLBs exhibited high spatial variability, and analyzed the differences in greenness between DLBs and surrounding vegetation under various environmental conditions. Our findings indicate that DLBs in thermokarst lakes, larger lakes, and lakes with higher drainage area ratios

exhibit higher vegetation greenness compared to DLBs in non-thermokarst lakes, smaller lakes, and lakes with lower drainage area ratios (Fig. 5). We quantified the influence of environmental factors on predicting NDVI in DLBs and analyzed temperature and flooding as key environmental constraints. These findings can help with the management of DLBs as shifting habitat mosaics[77], and facilitate future conservation efforts for Arctic biodiversity.

## Methods

### Surface water products

In this study, we utilized two of the most comprehensive surface water products available to assist in identifying drained lakes in the Arctic permafrost zone. These two data products, both based on archived Landsat imagery at 30 m resolution, are published by the Joint Research Centre (JRC) of the European Commission[36] and the Global Land Analysis and Discovery (GLAD) team at the University of Maryland[37], respectively. We identified drained lakes in permafrost regions by selecting pixels labeled as "lost permanent" in the JRC water transition map and as "water loss" in the GLAD water dynamics map, which both represent the transition from lakes to DLBs. However, there are notable differences in the identification methods used by the two datasets.

The JRC map identifies the start year of transition for each water pixel as the first year between 1984 and 2000 that has sufficient observations to characterize the presence of water, while the GLAD map uses 1999 as the start year. The JRC map identifies water pixel changes based on the initial and final states of the time period, with intervening years considered for the presence of ephemeral permanent (year-round) or seasonal water only if the initial and final years represent land. Consequently, the JRC map can classify cyclical water inundation as ephemeral water, permanent/seasonal water, water loss or water gain. In contrast, the GLAD map evaluates seasonal and interannual fluctuations of water pixels over the entire time series, classifying only pixels with a stable trajectory of loss as "water loss". To mitigate short-term annual anomalies and inter-annual observation variability, the GLAD dataset employed a 3-year mean moving window to smooth the annual open water percentage time series.

Both the GLAD and JRC maps offer valuable insights for this study. It is worth noting that while the JRC and GLAD products mark the spatial distribution of these water loss pixels, they do not provide information on the timing of transition from permanent water to land for each pixel. In addition, we used the global floodplain dataset[78] based on 250 m resolution MODIS satellite images to analyze the impact of flood inundation on vegetation dynamics in DLBs.

### Object-based lake analysis

The detection of lake drainage events requires calculating the proportion of lake drainage at the lake object level, but existing water products are in raster format and only provide pixel-by-pixel water loss information. We used the object-based analysis function provided by the Google Earth Engine (GEE) cloud platform[79] to create lake objects and identify lake drainage events. First, we used the JRC product's thematic map of water extent as a mask to identify persistent water. Morphological operations, involving erosion followed by dilation with a 1-pixel radius and 2 iterations, were conducted on the water extent raster image to separate finely connected water bodies and eliminate isolated pixels. We converted the set of connected pixels into water objects with unique identifiers via GEE and calculated the number of water pixels for each lake object.

To avoid missing drained lakes, we performed spatial union of water loss pixels from JRC and GLAD products and calculated the proportion of lake drainage within the scope of each lake object. After screening, we found approximately $2.3 \times 10^5$ lake objects larger than 1 hectare in the northern permafrost region with varying degrees of water loss during 1984–2020. We identified lake objects as having

undergone a drainage event only if their proportion of surface area loss exceeded 50%, which is a more stringent criterion than that used in previous studies[31,34,42,43], given focus on investigating drastic alterations in Arctic lake systems. Here, we have not imposed constraints on the time span of lake drainage events, as there is no universally accepted quantitative definition for the specific duration of lake drainage events in existing literature[25]. In some cases, lakes can gradually drain over multiple years (Fig. S11).

### Circum-Arctic maps of permafrost, soil and ground ice conditions

We utilized the most up-to-date data products (Supplementary Fig. S5) available on thermokarst lake coverage (released in 2016)[33], permafrost extent (2019)[40], Yedoma region (2021)[39], ground ice content (2002)[41], soil carbon content (2013)[80] and soil nitrogen content (2020)[81] to investigate the influence of permafrost-related properties on lake drainage and post-drainage vegetation growth. Olefeldt et al. (2016)[33] provided a thermokarst lake distribution map based on expert judgment, categorized into five levels: very high (60–100%), high (30–60%), medium (10–30%), low (1–10%) and none (0–1%). For analytical convenience, we reclassified thermokarst lake likelihoods into three categories: very likely (60–100%), likely (1–60%), and unlikely (0–1%). This classification is map-based and not an assessment of the origin for individual drained lakes.

We used the global permafrost map[40] with 1 km resolution to classify permafrost extent into continuous (>90%), discontinuous (50–90%), sporadic (10–50%) and isolated (<10%) permafrost zones based on area percent. The circumpolar ground ice product[41] classifies ground ice content into three classes based on volume percentage: high (>20%), medium (10–20%), and low (0–10%). Yedoma is an organic-rich (~2% carbon by mass), ice-rich (>50% ice content by volume) permafrost formed during the Pleistocene, mainly distributed in eastern Siberia, Alaska and the Yukon, and the landscape is characterized by glacial plains and hills with sparse vegetation cover[39]. While it is worth noting that these datasets do not provide temporal information, we would like to emphasize that the overall impact on our analysis is generally limited. This can be attributed to the relatively modest changes in permafrost-related properties over the course of a decade, with the exception of ground ice content, which represents the only dataset currently available.

In this study, the spatial density of drained lakes (Fig. 1h) was calculated by overlaying the spatial layers delineated from collected data products with the detected drained lakes, representing the number of drained lakes per unit area in different regional units. The lake-wise density of drained lake (Fig. 4) is calculated based on the ratio of drained lakes to all lakes within a specific region unit or attribute type. These methods enable us to visually depict the spatial distribution pattern and occurrence probability of lake drainage events across various regions. In addition, we analyzed the spatial heterogeneity of drained lakes and vegetation dynamics in the circum-Arctic permafrost region using the ecoregion map[38]. Ecoregions are defined as vast terrestrial or aquatic domains characterized by unique ensemble of species, communities, and environmental conditions that are geographically distinct. The circumpolar permafrost region was partitioned into 50 ecoregions (see Supplementary Fig. S4 and Supplementary Table S1), primarily composed of tundra and boreal forest biomes.

### Acquisition and preprocessing of Landsat data

This study utilized long-term time series of orthorectified Landsat surface reflectance imagery archives, provided by the GEE platform[79], to detect the occurrence years of lake drainage events and track vegetation growth dynamics in DLBs. The complete coverage data spanned from 2000 to 2020 and was derived from Landsat-5 Thematic Mapper (TM), Landsat-7 Enhanced Thematic Mapper-plus (ETM+) and

Landsat-8 Operational Land Imager (OLI). Regions like Alaska have had enough observation density to analyze the seasonal and interannual dynamics of surface water since 1984, but the first observation in Siberia was not until around 2000[36]. Therefore, despite the JRC product providing valuable isolated observations of surface water dynamics prior to 2000, our evaluation of lake drainage and vegetation growth dynamics is limited to the period 2001–2020.

We utilized an open-source toolkit (https://jdbcode.github.io/EE-LCB/) called Earth Engine Landsat Collection Builder for preprocessing the satellite image archive. This toolkit is deeply integrated into the GEE platform and is specifically designed for Landsat image pre-processing, offering analysis-ready images through customizable preprocessing chains. The preprocessing workflow employed in this study encompassed image filtering, cloud and shadow masking, sensor harmonization, multispectral index calculation, and image compositing.

Landsat images were first screened for acceptable data quality, with cloud coverage less than 50% to reduce cloud and shadow interference, and acquired during June to September each year to maintain phenological consistency and minimize the impact of ice and snow. The quality assessment band (pixel_qa) of Landsat images was then used to mask out observation noise such as clouds, ice and snow, and shadows in the image collection to improve image quality. Landsat images from different sensor sources (TM, ETM+, and OLI) were processed based on the statistical transformation function to correct spectral differences and improve spectral continuity of the images. Three multispectral indices were calculated at pixel-level to detect long-term changes in lakes and vegetation, including the Tasseled Cap Greenness index (TCG), the Normalized Difference Vegetation Index (NDVI), and the Automated Water Extraction Index (AWEI). TCG and NDVI provide insights into vegetation status and greenness[55], while AWEI is useful for accurately distinguishing between water and non-water pixels[82]. Finally, for each pixel and each index of the processed Landsat image collection, annual composite images are generated using the percentile composite method (ee.Reducer.percentile). We employed the annual time series of TCG and NDVI to analyze post-drainage vegetation dynamics and utilized AWEI time series as the basis for lake drainage year detection. Note that multispectral indices are calculated prior to image compositing to ensure the reliability of analysis based on individual observations. The processed data forms the foundation for consistent monitoring of lake dynamics and quantification of vegetation changes over a 20-year period.

## Detection of lake drainage year

Based on the JRC and GLAD maps and object-based lake analysis method, we identified lakes that underwent drainage events. We then employed the Landsat-based Detection of Trends in Disturbance and Recovery (LandTrendr) algorithm[45] to detect the main occurrence year of lake drainage events. The LandTrendr algorithm was integrated into the GEE cloud platform in 2018, and has been demonstrated to accurately detect the timing of thermokarst lake drainage events in northern Alaska[46,83]. The algorithm effectively leverages the high-frequency multi-temporal analysis capabilities of Landsat imagery to capture change trends and disturbance dynamics reflected in spectral trajectories, thereby accurately identifying and recording the occurrence year, disturbance intensity, and duration of disturbance events.

The core of the LandTrendr algorithm is the pixel-level spectral-temporal segmentation, which simplifies the spectral trajectory into a series of breakpoints and linear segments through iterative piecewise linear fitting, to describe change trends and disturbance events more concisely. We constrained the LandTrendr algorithm by adjusting the fitting parameters to focus on capturing the spectral variation characteristics of the drained lakes and carried out pixel-by-pixel disturbance year detection (refer to Supplementary Table S5 for the control parameters of the Landtrendr algorithm). For lakes experiencing multi-year drainage, LandTrendr accurately detected the annual process of lake drainage (Supplementary Fig. S11). We identified the year with the most drained pixels as the occurrence year of the lake drainage event.

## Validation of detected lake drainage events

Remote sensing detection of lake extent and identification of drainage events in the circum-Arctic permafrost region is a difficult challenge because the spectral properties of water can vary due to factors such as chlorophyll concentration, total suspended solids, colored dissolved organic matter, water depth, and observation conditions[84,85]. The JRC and GLAD products reported the following accuracies on a pixel-by-pixel basis: JRC with a user's accuracy of 49.8% (±19.3%) and a producer's accuracy of 65.5% (±11.4%), and GLAD with a user's accuracy of 30.0% (±6.5%) and a producer's accuracy of 86.2% (±7.4%). The GLAD product exhibits a higher producer's accuracy (corresponding to a lower omission error) compared to the JRC product in capturing the transition from water to land, although this comes at the cost of a lower user's accuracy (corresponding to a higher commission error). In this study, we improved the detection rate by combining water loss pixels from the JRC and GLAD products, and filtered out false lake drainage information by setting the criteria of lake area (>1 ha) and drainage area ratio (>50%).

Validation of detected lake drainage events was performed on an individual lake basis through visual interpretation using TimeSync[86], a visualization tool based on Landsat images and with a similar framework as the LandTrendr algorithm. The accuracy evaluated through the TimeSync tool only reflects the false positive rate of the drained lake's spatial location and occurrence time. The false negative rate of lake drainage events is challenging to accurately assess due to the relatively small proportion of drained lakes (less than 1% of all lakes) and the absence of a comprehensive reference dataset of drained lakes with the same temporal coverage. We randomly selected and validated 10% of the 35,337 detected lake drainage events, specifically distinguishing the accuracy of detection for small and medium-large lakes. This distinction was made considering the inherent difficulty in classifying transitions in small water bodies. The results of spatial validation for drained lakes showed that the accuracy of detection (i.e., the percentage of lakes that actually drained versus all lakes mapped as drained) for small and medium-large drained lakes was approximately 82.6% and 96.1% respectively. The temporal accuracy (i.e., the percentage of mapped drained lakes where the major drainage year was correctly identified) of the LandTrendr algorithm's detection year for small and medium-large drained lakes was found to be 63.8% and 89.2%, respectively.

The detection errors of drainage years were mainly due to the anomalous fluctuations of the spectral features of remote sensing images that caused the LandTrendr algorithm to fail to correctly identify the disturbance years. In some cases, the LandTrendr algorithm may have incorrectly mapped the disturbance years to the first and last years of the study period, resulting in larger errors in the detected lake drainage events in 2001 and 2020 than in other years[46]. Additionally, the issue of underdetection in the spatial recognition of drained lakes primarily occurs in dense lake regions along the coastal lowlands. When dividing lake objects, multiple lakes may be combined into a single entity due to pixel adhesion, resulting in an underestimation of the drainage area ratio and underdetection of drained lakes.

## Track vegetation dynamics in DLBs

To track the dynamics of vegetation growth following lake drainage, within the preprocessing of Landsat image archives, we utilized the 90th percentile of NDVI values from annual image collections to generate a robust NDVI time series. This approach yields a value resembling the second highest NDVI value, which might perform

better than the maximum and median NDVI values considering the potential influence of shadow masking and standing water in the lake basin.

Based on the generated 90th percentile NDVI annual time series, we pixel-wise acquired NDVI values within the lake basin area for each drained lake, masked out values below 0, and computed the median NDVI for the region. This step further mitigated the impact of standing water on NDVI assessment in DLBs. We also calculated the median NDVI of the surrounding vegetation to compare its greenness difference with DLBs. The surrounding vegetation was defined by a circular buffer of twice the lake diameter centered at the lake center, and masked to exclude the lake and other water bodies. Lastly, based on the identified drainage year for each lake, we transformed the time series of NDVI values from 2000 to 2020 into vegetation dynamics recorded by years after drainage.

Additionally, we applied the same approach to extract the TCG index from both DLBs and the surrounding areas for comparison (Supplementary Fig. S9). The results indicate that despite being based on different image bands and algorithms, the vegetation greening trends and patterns reflected by TCG are basically consistent with those of NDVI, underscoring the reliability of our method in capturing the vegetation dynamics in DLBs.

## ERA5-Land reanalysis data

Due to the lack of fully covered high spatial resolution meteorological products in the northern permafrost regions, we used ERA5-Land monthly reanalysis dataset[87] for the period 2000–2020 to analyze the climate factors affecting lake drainage and vegetation growth. ERA5-Land is a land-enhanced product from the fifth generation of European ReAnalysis (ERA5), with a spatial resolution of 0.1° (resampled to ~11 km in the GEE platform). It is considered a state-of-the-art global reanalysis dataset for land applications that efficiently reconstructs surface states and process parameters through advanced data assimilation techniques[87]. A recent study noted that ERA5-Land showed a high degree of global consistency with MODIS satellite products in terms of surface temperature[88].

Despite the relatively coarse spatial resolution of ERA5-Land, the analysis based on a 0.1° × 0.1° grid reveals that 35,337 drained lakes are distributed in 20,132 grid cells, most of which contained 1–2 drained lakes (Supplementary Fig. S12a). Over 99% of the grid cells exhibit fewer than 10 drained lakes (Supplementary Fig. S12b). The highest count of drained lakes in a grid cell is 33, occurring at St. Lawrence Island (Fig. 1g). Therefore, for lake drainage events occurring between 2000 and 2020, ERA5-Land can provide sufficiently differentiated climatic information for lake drainage prediction and post-drainage vegetation dynamics analysis. It should be noted that utilizing coarse-resolution ERA5-Land data introduces uncertainties into the modeling and relationships established between climate drivers and observed changes for individual lakes.

Furthermore, we evaluated the temperature and precipitation of ERA5-Land dataset using the 1 km resolution Daymet dataset[89] covering North America (Supplementary Fig. S13). This analysis compares climate data of Daymet and ERA5-Land for the year in which the lakes were drained using 16,104 drained lakes detected in North America as sample points. The results indicate overall consistency and relatively low variability between Daymet and ERA5-Land reanalysis data in simulating temperature and precipitation (Supplementary Fig. S13).

The parameters extracted from the ERA5-Land monthly reanalysis dataset in this study include: air temperature, soil temperature, soil moisture, precipitation, evaporation, snowfall, snowmelt, wind speed, and solar radiation. Specifically, the air temperature was taken at a height of two meters above the ground, and the soil temperature and soil moisture were taken at the first soil layer (0–7 cm). Additionally, we used "total_precipitation" minus snowfall to obtain rainfall, and used "total_evaporation" to obtain evaporation with the effect of transpiration

included. We calculated annual averages, summer averages (June, July, August), and Sen's slope values for these climatic parameters.

## Machine learning models

In this study, we trained a binary classification model for predicting lake drainage (Fig. 3) and a regression model to predict NDVI within DLBs (Fig. 6). We utilized CatBoost[90], a categorical-feature-based gradient boosting method, for model training. This choice was motivated by several advantages: (a) CatBoost can effectively handle heterogeneous data comprised of both continuous and categorical variables, (b) it demonstrates resilience against outliers and noise data, and (c) it generates enhanced models with reduced bias and variance through ensemble learning, as opposed to individual tree-based models[90].

The training and testing samples for the binary classification model were derived from all drained lakes (~28,000) between 2001 and 2020 and a randomly selected 1% subset of undrained lakes (~58,000). The candidate explanatory variables for the model are detailed in Supplementary Table S3. For drained lakes, the climate parameters were extracted using values corresponding to their identified primary drainage year, while for undrained lakes, climate parameters were randomly sampled between the years 2001 and 2020, aiming to maximize coverage across different climate conditions. We used 70% of the samples for model training and the remaining 30% for independent testing of predictive performance. We utilized the Python-based Scikit-learn library[91] to fit CatBoost models and determined the optimal model hyperparameters through random search and ten-fold cross-validation. The initial model includes all candidate explanatory variables, which might exhibit collinearity, meaning that introducing certain variables may lead to considerably changes in the importance estimates of other variables. To address this issue, during the iterative training process, we removed features that had negligible contributions to model performance and features highly correlated (Pearson correlation coefficient $r > 0.5$) with the most important variables for model performance. The important variables were identified through preliminary feature importance evaluations through the Shapley additive explanations model interpretability approach[92]. We tested training separate models for very likely and unlikely thermokarst lakes for drainage prediction but did not observe an improvement in model accuracy. Therefore, we included thermokarst lake likelihoods as a categorical feature in the final model. The final model achieved an AUC score of 0.92, indicating its effectiveness in distinguishing drained lakes from undrained lakes. The model's precision is 0.84, recall is 0.72, and average precision is 0.88, demonstrating its ability to maintain high accuracy while effectively capturing drained lake samples. The remaining explanatory variables of the final model are shown in Fig. 3a, and the optimal model hyperparameters are listed in Supplementary Table S6.

The training process of the regression model for predicting NDVI in DLBs is similar to the binary classification model described earlier, with the distinction that the predicted variable is a continuous value between 0 and 1 (NDVI), rather than a binary value (0 or 1). Additionally, the loss function used is Root Mean Square Error (RMSE) instead of Logarithmic Loss. The training and testing samples of the model are based on the annual NDVI values of all drained lakes and the climate data of the corresponding years, recorded with reference to the year of lake drainage. In other words, the model predicts the NDVI values in DLBs for the 0–15 years after lake drainage. Just like the binary classification model, we followed a similar approach for training and testing the regression model, utilizing the CatBoost algorithm and conducting a 70–30 split of the dataset for training and independent testing. The initial model included all candidate explanatory variables (Supplementary Table S4), and the feature selection process was also applied to remove negligible or highly correlated variables based on Shapley values and Pearson correlation coefficients. The optimal hyperparameters for the model were determined through methods like random

search and ten-fold cross-validation (Supplementary Table S7). The final model's RMSE is 0.08, $R^2$ is 0.83, mean absolute error is 0.06, and mean bias is 0.00, indicating that the regression model can effectively predict NDVI values in DLBs based on relevant climate and environmental variables, supporting a deeper understanding of post-drainage vegetation dynamics.

We employed two importance evaluation methods, namely Shapley values and Permutation importance, to interpret the machine learning model and quantify the contribution of each explanatory variable. The Shapley value is a contribution score for each feature, indicating its expected marginal contribution on the prediction task. The calculation method of Shapley value is to average the feature contribution of all possible feature alliances and consider all possible interaction combinations between features. It's important to note that the range of Shapley values differs between classification and regression models, making them non-comparable. Therefore, the focus should be on analyzing the importance ranking of variables within each model. The calculation of Permutation importance involves randomly altering the value of a single feature while keeping the values of the other features unchanged. The subsequent observation of how this affects the final model performance metric indicates the feature's importance. The greater the reduction in the metric after permutation, the more crucial the feature is for the model's predictive ability. Together, these two approaches provide a fair and comprehensive assessment of feature importance, taking into account both main effects and interactions.

### Reporting summary

Further information on research design is available in the Nature Portfolio Reporting Summary linked to this article.

## Data availability

The drained lakes data generated in this study have been deposited in the Zenodo database under accession code https://zenodo.org/record/7632013#.Y-d7iHZBxD8. The United States Geologic Survey Landsat 5, 7, and 8 Surface Reflectance data used in this study are available from Google Earth Engine at https://developers.google.com/earth-engine/datasets/catalog/LANDSAT_LT05_C02_T1_L2, https://developers.google.com/earth-engine/datasets/catalog/LANDSAT_LE07_C02_T1_L2, https://developers.google.com/earth-engine/datasets/catalog/LANDSAT_LC08_C02_T1_L2. The surface water products published by the Joint Research Centre (JRC) of the European Commission[36] and the Global Land Analysis and Discovery (GLAD) team[37] are available at https://global-surface-water.appspot.com/download, https://www.glad.umd.edu/dataset/global-surface-water-dynamics. The global floodplain dataset[78] based on MODIS are available at https://developers.google.com/earth-engine/datasets/catalog/GLOBAL_FLOOD_DB_MODIS_EVENTS_V1. The maps of permafrost extent[40], lake thermokarst landscape[33], Yedoma[39], ground ice content[41], ecoregion map[38] are available at https://doi.pangaea.de/10.1594/PANGAEA.888600, https://doi.org/10.3334/ORNLDAAC/1332, https://maps.awi.de/awimaps/projects/public/?cu=ice_rich_yedoma_permafrost, https://doi.org/10.7265/skbg-kf16, https://www.worldwildlife.org/publications/terrestrial-ecoregions-of-the-world. The storage of soil organic carbon[80] is available at http://bolin.su.se/data/ncscd/, and the nitrogen content[81] at https://bolin.su.se/data/hugelius-2020. The ECMWF Climate reanalysis ERA5-Land[87] Monthly data is available at https://developers.google.com/earth-engine/datasets/catalog/ECMWF_ERA5_LAND_MONTHLY_AGGR. Source data are provided with this paper.

## Code availability

Google Earth Engine code used to detect lake drainage events and Python code used to perform machine learning analysis[93] are available at https://doi.org/10.5281/zenodo.10043131.

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

## Acknowledgements
We thank the JRC and GLAD teams for providing invaluable surface water products. We also thank the GEE platform for its support in processing large-scale remote sensing data. We also thank Dr. Zhiqiang Yang for providing access to the TimeSync platform. This research was funded by the National Outstanding Youth Foundation of China (Grant No. 41925027 to X.C.), the National Natural Science Foundation of China (Grant No. 42301148 to Y.C., 42306254 to A.L., and 42077051 to Q.C.), the Major Project of Key Research Bases for Humanities and Social Sciences Funded by the Ministry of Education of China (Grant No. 22JJD790015 to C.W.), and the Natural Science Foundation of Shandong Province, China (Grant No. ZR2023QD022 to Y.C.).

## Author contributions
Y.C. contributed to the conceptualization, methodology, formal analysis, and writing of the manuscript. X.C. contributed to the conceptualization, review and editing of the manuscript as well as supervised the work. A.L. contributed to the conceptualization, methodology, formal analysis, and writing of the manuscript as well as supervised the work. Q.C. and C.W. contributed to the review and editing of the manuscript.

## Competing interests
The authors declare no competing interests.
