## [Peer Review File · Nature Communications]

Tracking lake drainage events and drained lake basin vegetation dynamics across the ArcticREVIEWER COMMENTS

Reviewer #1 (Remarks to the Author):

This study is an important step forward and very well presented. This study provides a holistic analysis of drained lakes in the Arctic, providing a database of drained lakes, correlation of draining with climate and environmental variables, and analysis of subsequent vegetation growth. The authors found marked differences between thermokarst and non-thermokarst lakes in the correlating variables with a greater dependence on temperature and water, respectively, and in the subsequent vegetation with thermokarst lakes having a greater intensity of vegetation. By analyzing thermokarst and non-thermokarst with identical methods, the variations can be effectively compared.

Below are specific comments with most of them just requesting greater clarity. The one methodological change I would request is to calculate the NDVI using the reflectances from a single image rather than likely two different images. Great work on an awesome study.

9: "previously overlooked" meaning all non-thermokarst not a subset of them. Can you make this more clear?

10-12: The second two clauses are difficult to pair with the first before reading the paper. Also "upward trends" of what? Perhaps: increasing rates of drainage.

25: "net gain" of what?

53: The lake drainage events were detected by the existing products (JRC/GLAD). Here you identify them in those products and then detect the drainage year by satellite remote sensing.

85: "high spatial variability" of what?

Fig 1: It is hard to glean from D all that is there and I would love to be able to see it better. The colors seem to bleed together. Can you add more contrast? Perhaps yellow instead of purple.

Fig S6: Are these lines the median over all the drained lakes per year?

Fig 2 and S7: These figures would be more valuable with the points colored by year.

200: "approximately 15 years" makes me think it goes up after that. Is that true? Perhaps for at least 15 years? Or for the 15 years we were able to analyze?

299: "analyzed cases where..." which means those cases existed but doesn't tell what you found or how prevalent they were. Engage more deeply with references 23 and 27 that made this case (per line 49). Your results seem to say that there is high variability and for a little more than half of thermokarst lakes the vegetation was more luxuriant, but this did not hold for the majority of non-thermokarst lakes. There are many intricacies playing into the vegetation cover rates as you describe elsewhere. State the conclusions we can and cannot draw from this study on the question of whether vegetation is more luxuriant.

312-316: The water transition map of JRC and the water dynamics map of GLAD are quite different. The JRC map only looks at the first and last year and ignores all the intervening years except when the start and end years were land and then the intervening years are checked for ephemeral permanent (year-round) or seasonal water. Thus cyclical water inundation could be captured as permanent/seasonal water, water loss, water gain, or ephemeral water. Whereas, the GLAD layer evaluates all the years for all pixels, and only pixels with a stable trajectory of loss are included in the "water loss" class. For this study, both products provide valuable insights.

However, the difference between the products should be made more clear.

331: Please provide more details on the "morphological opening operations". I'm assuming this is an erosion followed by a dilation. What was the scale of these operations?

345: LandTrendr was only applied to the lakes with drainage as delineated by the JRC/GLAD maps, correct? Make this more explicit. Together with lines 63-67 this is not completely clear.

352-360: Revise this paragraph. Very important information, but as written the sentences do not connect well with one another. Also, here you state you use the entire archive whereas later you describe how you filtered the images and pixels for the NDVI assessment. Did you do any of those same steps here or every single image was used as is for LandTrendr?

377: You have two sets of pairs (users/producers, JRC/GLAD) which makes it difficult to interpret the respectively. I initially interpreted the first two as JRC and the second two as GLAD, but the next sentence seems that it is paired by accuracy. Also, perhaps add "per pixel" (The reported per pixel...) to clarify this vs the validation of lakes as an object that is next.

383: Is this event/lake based or pixel based?

389: 10% randomly selected? State that.

391: "accuracy of detection" means the percent of lakes that actually drained vs all those mapped as drained? And the temporal accuracy means the percent of mapped drained lakes where the majority drain year was correctly identified?

406: What years are these datasets from? Does this impact your results at all?

451: NDVI should be calculated from the NIR and Red reflectance of a single observation. You are taking the median red value and the median NIR value and calculating NDVI to get a figure for the year. However, there is a good chance that the median NIR and the median red come from different dates. Thus it could be combining different vegetative states as well as any differences in atmospheric contamination/correction. Instead, you should calculate the NDVI for all clear observations within the summer and then take the median. Or perhaps even better take something like the second highest NDVI (may perform better than the max to account for missed shadows)

494: many of these links gave 404 responses.

The manuscript “Tracing Arctic lake drainage events and emergent vegetation in drained lake basins” identifies lake drained lakes across the northern permafrost zone and tracks NDVI trends in these basins following drainage. The authors then calculate trends in the frequency of drainage events and identify the environmental variables that covary with drainage event frequency. Lastly, the authors compare NDVI within drained lake basins to the surrounding vegetation and identify the environmental variables that explain variation in NDVI.

Mapping drainage events at such a large scale is an important forward step in our understanding of Arctic lake hydrology. In general, the methods are sound, but there are a few analyses that need improvement. However, the presentation, interpretation, and discussion of the results is under-developed, the text is woefully under-referenced, and I am left wondering what the study adds to our understanding of Arctic ecosystems.

Below, I detail the under-development of the results, interpretation, and discussion of each subsection.

Spatial patterns of drained lakes in the Arctic. Here, the authors describe the size and area of DLBs over the entire study domain as well as provide some eco-region specific results. However, without describing the size distribution of existing lakes over the domain (or specific ecoregion), it is impossible to interpret the meaning of the size distribution of DLBs. One question that comes to mind but was not answered in the text is: are large or small lakes more likely to drain? A similar argument can be made for comparisons between regions; if all regions exhibited the same lake size distribution, it would be notable that some regions have smaller DLBs and some have larger. However, not all regions have the same lake size distribution. It is therefore unclear how to interpret the region-specific results; is the fact that some regions exhibit smaller DLBs simply a reflection of the fact that the lakes in that region are smaller, or is there something about the landscape that makes smaller lakes in this region more likely to drain? Furthermore, only three ecoregions across the entire study domain are highlighted with no explanation as to why they were chosen or what their significance is.

Trends of lake drainage events for 2001-2000. The authors report an increase in thermokarst DLB occurrence over the study period, with a slope of 19. However, this slope is not interpreted; does this mean that, on average, each year sees an increase of 19 DLBs across the domain? Because that seems like a very small number and, even if statistically significant, is not ecologically significant. But the authors do not guide the reader here and we are left to wonder that question as well as others like: does the trend in DLB occurrence vary by region, permafrost extent, or ground ice extent?

One major result (highlighted in the abstract) is that summer soil temperature is a major driver of the trend in thermokarst DLB occurrence frequency. This claim is based on soil temperature data from the ERA-5, which, according to a previously published evaluation of this dataset, “... are not well suited for informing permafrost research” (Cao et al., 2020). Given that the data informing their results may not be valid, I question the reliability of the results presented here. The authors do not provide any justification for using this dataset. Furthermore, this finding seems to be based on correlation coefficients between climate variables and DLB occurrence frequency. Given that multiple environmental variables, many of which are likely correlated,

likely lead to DLB occurrence frequency, analysis of variables individually is not an appropriate approach. One option would be multiple regression analysis, where all environmental variables would be considered together, residuals would be checked, etc.

Post-drainage vegetation dynamics in DLBs. This section reports the results of the authors' comparison of NDVI within and next to DLBs. A major confounding factor in this analysis is the presence of surface water; since the authors use a cutoff of 50% drainage, many of the DLBs still contain standing water. Surface water is known to influence NDVI values (Myers-Smith et al., 2020), and it would follow that any comparison of NDVI within incompletely drained DLBs to nearby land with presumably less standing water would also be affected by the presence of water. This issue must, at minimum, be discussed and, if possible, mitigated.

With respect to the results of this section, there is no attempt to put them into a larger context, even the context the authors lay out at the beginning of the paper (e.g., what is the role of DLBs in promoting Arctic greening?). One novel finding seems to be that, in general, non-thermokarst DLBs exhibit lower NDVI than the surrounding vegetation. But the reasons for and implications of this are not discussed.

Environmental constraints of vegetation growth in DLBs. This section is a litany of results without interpretation or discussion and I am unsure how the results enhance our understanding of the ecosystem.

Other comments:

The authors describe their study region as the 'entire circum-Arctic permafrost region' and refer to it as the Arctic throughout the text. However, much of the study region is in fact not part of the Arctic such as subarctic/boreal regions and the Tibetan plateau. The term 'Arctic' is inappropriate for the study region. An alternative could be: 'northern permafrost zone,' but there are likely others.

The authors divide DLBs into two types: thermokarst and non-thermokarst. I cannot find in the methods how these two lake types were differentiated, which is problematic. But assuming this was done based on the map from Olefeldt et al., 2016, the authors are defining lakes by whether or not they fall within a region of high thermokarst lake coverage, NOT whether each individual lake is a thermokarst lake or not. Non-thermokarst lakes exist within regions of high thermokarst lake coverage. See for example Lara and Chipman (2021). Presumably, thermokarst lakes can also exist within regions of low thermokarst lake coverage. Therefore, what the authors are referring to as a 'thermokarst DLB' may actually be of non-thermokarst origin (and same for non-thermokarst DLB being of thermokarst origin). This point needs to be highlighted in the text and different language needs to be used to describe lake types.

Notably, the authors do not reference or discuss other studies related to DLB mapping in the Arctic. See for example: Lara and Chipman, 2021; Lara et al., 2021; Bergstedt et al., 2021. Other studies that track DLB frequency over time (also not cited): Marsh et al., 2009; Lantz and Turner, 2015; Lara et al., 2021.

Line-by-line comments:

Figure 1D: The color choice for the lake size is difficult to differentiate.

Figure 4: I am not sure what NDVI means here. According to the methods you calculated NDVI for multiple years after drainage, but there is no mention of the timeframe for NDVI in this figure.

Line	Comments
5	delete 'the'
21	change 'its fragile ecosystem' to 'ecosystems.' Additionally, what is a challenge to an ecosystem? Consider re-wording.
22	The Arctic is a region, not an ecosystem. You could simply say 'Arctic ecosystems'
25	what do you mean by 'net gain?' Earlier in the sentence you say there is a net loss of lake area. What is gaining?
29	Delete critical. (How are methane emissions critical?)
31	change 'hydrological' to 'water.' Also, this is a long sentence. Consider breaking up into two.
35	The point is not that DLBs have long intrigued researchers, but rather DLBs are important for ecosystem permafrost characteristics, vegetation dynamics, and carbon storage.
42	There are studies that have looked at this; it has not been ignored. Both Webb et al., (2022) and Nitze et al., (2018) included ground-ice poor regions. Other examples include: Smol and Douglas (2007), Carroll et al., (2011), Finger Higgins et al., (2019), Law et al. (2018) Carroll and Loboda (2018)
45	This paper looked at drainage event timing and associated landscape characteristics and environmental drivers across NW Alaska: Lara et al., (2021)
52	Include a ref for Arctic greening.
53	Entire circum is redundant. Choose one.
54	Change 'four decades or so' to 'three and a half decades'; change 'by' to 'using'
69	Over what time period? If a lake lost 50% of its area over the 35 years, would that be a DLB? Or does the drainage need to occur on a shorter timescale?
73	Delete 'spatially.' Unless it is important to include, in which case you will need to re-word the sentence.
74	Are you saying that the thermokarst lakes are located in coastal plains and river deltas, or are you saying that the majority of DLBs occurred here?
78	Did you show that here? If so, refer to table/figure. If you don't show that here, you will need a reference.
79	The first part of this paragraph is on thermokarst lakes. The second part is on lake size. Are these lake sizes for thermokarst lakes or for all lake types? Please clarify and if it is all lakes combined, do not include in the same paragraph as the discussion of thermokarst lakes.
86	But perhaps all the lakes in this region are small. How does this compare to the size distribution of existing lakes?
87	Again, how does this compare to the size distribution of all lakes in the region? You may be showing that the lakes in this region are larger than average, and so the drained lakes are larger than average.

- 88 How do you know they are previously unnoticed? The people who live there may have noticed.
- 91 Are you referring to the events on St. Lawrence Island, or across the study region?
- 101 Should be 'the continuous permafrost zone' rather than 'continuous permafrost zones'
- 102 With climate change, some continuous permafrost at the southern extent will become discontinuous permafrost. But the entire continuous permafrost zone is not vulnerable to becoming discontinuous. This sentence is misleading and the references do not support the claim.
- 104 This is the first time you mention ice wedge degradation and drainage channels. I suggest you explain this mechanism a bit more.
- 107 I am not sure why this is implied. More development of this idea is needed, including references.
- 125 How can you know that there were undetected events if you cannot detect them? A simple rewording might suffice.
- 130 What is the p-value for this statement? Is there a figure or table where this is showed?
- 131 I am having trouble interpreting the slope here. This is an increase of 19 drainage events per year? For the whole region?
- 133 I am not sure what the percentages are referring to. The numerator is drained thermokarst lakes but what is the denominator?
- 148 remove 'the'
- 152 This sentence needs a citation
- 157 These references are for thermokarst lakes, yet the sentence is about non-thermokarst lakes
- 159 This statement needs a citation
- 183 This statement needs a citation - your data is just NDVI. What field studies show that this is actually what is happening on the ground?
- 185 This statement needs a citation. What field studies show that this is actually what is happening on the ground?
- 186 need p-value
- 196 Since the Chen et al. numbers are in %, can you also put your numbers in % so the reader can compare?
- 197 need p-value
- 205 This statement needs a citation
- 207 This isn't just because there is more disturbed area, so more opportunity for veg growth?
- 207 Be careful here and elsewhere; you measured NDVI, not vegetation growth. NDVI is only a proxy and you do not have direct measurements of vegetation growth.
- 209 Need citation for "...favorable conditions for vegetation growth"
- 210 Explain this more. Shouldn't thin and rocky soils be characteristic of both the DLB and the surrounding areas? Why would this cause DLBs to have lower NDVI relative to the surrounding?
- 213 This statement needs a citation
- 215-16 I am having a hard time understanding what this sentence means. What does it mean to "facilitate nutrient concentration"? What does 'them' refer to?
- 245 How do you know this? At the very least, this statement needs a citation. But it should probably also be re-worded as speculative rather than declarative.

- 253 This makes it sound like an increase in temp at given location is beneficial for plant growth. But what I think the figure is showing is that, in general, warmer locations tend to have enhanced plant growth. There is a difference between the two.
- 268 Remote sensing studies do not ‘overcome’ sampling bias in field studies. The two approaches measure different things on different scales and give us different types of information.
- 270 entire circumpolar is redundant. Pick one.
- 291 emissions of what?
- 316 delete ‘directly’
- 329 Earth Engine should be capitalized and cited
- 330 ‘firstly’ should be ‘first’
- 340 Which other studies are you referring to? What is the justification for 50%?
- 375 Citation needed
- 388 Other drained lake datasets DO exist. See for example:
<https://arcticdata.io/catalog/view/doi:10.18739/A2BV79W8S>
<https://tc.copernicus.org/articles/14/4279/2020/#section7>
- 400 ‘densely’ should be ‘dense’
- 417 The Brown et al. ground ice map only has three categories: high, medium, and low ground ice. Please explain how you derive very high.
- 458 change ‘drainage lake ecosystems’ to ‘drained lake object’
- 473 citation needed

References cited

- Bergstedt, H., Jones, B. M., Hinkel, K., Farquharson, L., Gaglioti, B. V., Parsekian, A. D., Kanevskiy, M., Ohara, N., Breen, A. L., Rangel, R. C., Grosse, G., and Nitze, I.: Remote Sensing-Based Statistical Approach for Defining Drained Lake Basins in a Continuous Permafrost Region, North Slope of Alaska, *Remote Sensing*, 13, 2539, <https://doi.org/10.3390/rs13132539>, 2021.
- Cao, B., Gruber, S., Zheng, D., and Li, X.: The ERA5-Land soil temperature bias in permafrost regions, *The Cryosphere*, 14, 2581–2595, <https://doi.org/10.5194/tc-14-2581-2020>, 2020.
- Carroll, M. L. and Loboda, T. V.: The sign, magnitude and potential drivers of change in surface water extent in Canadian tundra, *Environmental Research Letters*, 13, 045009, <https://doi.org/10.1088/1748-9326/aab794>, 2018.
- Carroll, M. L., Townshend, J. R. G., Dimiceli, C. M., Loboda, T., and Sohlberg, R. A.: Shrinking lakes of the Arctic: Spatial relationships and trajectory of change, *Geophysical Research Letters*, 38, 1–5, <https://doi.org/10.1029/2011GL049427>, 2011.
- Finger Higgs, R. A., Chipman, J. W., Lutz, D. A., Culler, L. E., Virginia, R. A., and Ogden, L. A.: Changing Lake Dynamics Indicate a Drier Arctic in Western Greenland, *Journal of Geophysical Research: Biogeosciences*, 124, 870–883, <https://doi.org/10.1029/2018JG004879>, 2019.

Lantz, T. C. and Turner, K. W.: Changes in lake area in response to thermokarst processes and climate in Old Crow Flats, Yukon, *Journal of Geophysical Research: Biogeosciences*, 120, 513–524, <https://doi.org/10.1002/2014JG002744>, 2015.

Lara, M. J. and Chipman, M. L.: Periglacial Lake Origin Influences the Likelihood of Lake Drainage in Northern Alaska, *Remote Sensing*, 13, 852, <https://doi.org/10.3390/rs13050852>, 2021.

Lara, M. J., Chen, Y., and Jones, B. M.: Recent warming reverses forty-year decline in catastrophic lake drainage and hastens gradual lake drainage across northern Alaska, *Environmental Research Letters*, 16, <https://doi.org/10.1088/1748-9326/ac3602>, 2021.

Law, A. C., Nobajas, A., and Sangonzalo, R.: Heterogeneous changes in the surface area of lakes in the Kangerlussuaq area of southwestern Greenland between 1995 and 2017, *Arctic, Antarctic, and Alpine Research*, 50, e1487744, <https://doi.org/10.1080/15230430.2018.1487744>, 2018.

Marsh, P., Russell, M., Pohl, S., Haywood, H., and Onclin, C.: Changes in thaw lake drainage in the Western Canadian Arctic from 1950 to 2000, *Hydrological Processes*, 23, 145–158, <https://doi.org/10.1002/hyp.7179>, 2009.

Myers-Smith, I. H., Kerby, J. T., Phoenix, G. K., Bjerke, J. W., Epstein, H. E., Assmann, J. J., John, C., Andreu-Hayles, L., Angers-Blondin, S., Beck, P. S. A., Berner, L. T., Bhatt, U. S., Bjorkman, A. D., Blok, D., Bryn, A., Christiansen, C. T., Cornelissen, J. H. C., Cunliffe, A. M., Elmendorf, S. C., Forbes, B. C., Goetz, S. J., Hollister, R. D., de Jong, R., Loranty, M. M., Macias-Fauria, M., Maseyk, K., Normand, S., Olofsson, J., Parker, T. C., Parmentier, F. J. W., Post, E., Schaepman-Strub, G., Stordal, F., Sullivan, P. F., Thomas, H. J. D., Tømmervik, H., Treharne, R., Tweedie, C. E., Walker, D. A., Wilmking, M., and Wipf, S.: Complexity revealed in the greening of the Arctic, *Nature Climate Change*, 10, 106–117, <https://doi.org/10.1038/s41558-019-0688-1>, 2020.

Nitze, I., Grosse, G., Jones, B. M., Romanovsky, V. E., and Boike, J.: Remote sensing quantifies widespread abundance of permafrost region disturbances across the Arctic and Subarctic, *Nature Communications*, 9, 1–11, <https://doi.org/10.1038/s41467-018-07663-3>, 2018.

Smol, J. P. and Douglas, M. S. V.: Crossing the final ecological threshold in high Arctic ponds, *Proceedings of the National Academy of Sciences of the United States of America*, 104, 12395–12397, <https://doi.org/10.1073/pnas.0702777104>, 2007.

Webb, E. E., Liljedahl, A. K., Cordeiro, J. A., Loranty, M. M., Witharana, C., and Lichstein, J. W.: Permafrost thaw drives surface water decline across lake-rich regions of the Arctic, *Nature Climate Change*, <https://doi.org/10.1038/s41558-022-01455-w>, 2022.

Reviewer #3 (Remarks to the Author):

This study looks at Landsat images from 1984-2020 and identifies 35000 lake drainage events in Arctic regions, including about half that are previously unreported in non-thermokarst lakes. The analysis identifies summer soil temperature and net annual precipitation as the primary drivers of drainage in thermokarst and non-thermokarst lakes respectively and demonstrates growth in vegetation following the drainage event.

The findings presented are interesting and builds on prior studies (cited in the paper) that demonstrate decreased lake area due to permafrost thaw in the Arctic, and detection of lake drainage events in permafrost regions. That being said, most of the findings and some of the methodology essentially seem to be an extension of the work of Chen et al. 2022 (<https://doi.org/10.1016/j.scitotenv.2021.150828>) & Chen et al. 2021 (10.1111/gcb.15853), but applied to a global dataset. Even the figures seem remarkably similar (e.g. Figure 3 in Chen et al. 2021 and Fig 3A/3B in this paper). The most noteworthy, previously unreported findings are the distinctions between thermokarst and non-thermokarst lakes, but the implications of these differences can be better explained in the introduction and discussion sections.

I have some questions and concerns about the methodology. Overall some sections are not described in enough detail to evaluate and reproduce the methodology:

1. One of the primary findings is the relationship between climactic variables and the drainage lake events. This analysis seems to be problematic for many reasons. First the choice of the ERA-5 dataset seems strange for this purpose given its spatial resolution of ~9km, and that most of the lakes (85%) are very small 1-10 ha & 98% are <100 ha (1 sq km). I understand that there aren't meteorological products available at high resolution globally, but this enormous difference in resolution needs to be handled, by either upscaling the lakes to a comparable area (see for e.g. Webb et al. 2023), downscaling the ERA-5 data, or at least comparing with a finer-scale product for regions where they are available (e.g. Daymet at 1 km resolution). Secondly there are significant collinearities between the variables chosen, so it is not surprising that net annual precipitation (P-ET) and annual precipitation or summer soil temperature/annual soil temperature/air temperature have similar correlations. Redundancies in climate variables should be handled when interpreting the correlation analysis. I would also have liked to have seen a stronger approach (beyond correlations) to determining the drivers for the extent of drained lakes, and it seems like a boosted tree ML approach might also work for this purpose.
2. Another finding is with respect to the increased vegetation in drained lakes based on the changes in the NDVI. The ML approach overlaps with prior publications, but with less rigor (e.g. Webb et al. 2023: <https://www.nature.com/articles/s41558-022-01455-w#Sec8>). The description of the machine learning is not at all adequate. For e.g. the input variables, type of cross validation are not specified, nor are the hyperparameters included in the appendix. Why were 2 models fit separately for the thermokarst and non-thermokarst lakes, and not a single model considered, with thermokarst/non-thermokarst specified as a categorical variable? Were the inputs different for each model to differentiate the drivers of lake drainage for the two lake types (which is what it appears like from Fig 4)? A comprehensive list of inputs must be provided in the SI. The methods also specify that variables that didn't contribute much were removed based on the feature importance during the iterative training process. This can be problematic with redundant variables, as sometimes these can all have greater feature importance scores due to collinearity. Overall the models seem to have moderate skill ($R^2 = 0.71$). The Shapley values seem low, perhaps indicating that the variables don't have that much contribution to the NDVI, and needs to be accounted for explaining the results in 249-256 (e.g. there is only ~0.1 difference between the ranges of volumetric SWC in the partial dependence plot). A permutation feature importance could also be done to validate the findings.
3. Another section where I found the methodology inadequate is the one about the permafrost extent, ice content, soil carbon etc. It is unclear how the authors got to following finding "Regions with very high thermokarst lake coverage, discontinuous permafrost extent, and very high ground ice content are more susceptible to lake drainage, with densities approximately 3, 2.5, and 2 times higher than average, respectively (Fig. 1H)." It seems like this was done by overlaying many spatial layers with the drained lake areas, but that was not described in the methods. Also would it be possible to link these variables with the drainage events using the same ML model with the climate inputs (point 1 above)? There are examples where the gradient boosted trees can be used

for time series regression with dynamic (meteorological) inputs and static attributes (such as soil C %, permafrost extent etc).

In general, given the overlap in the findings/methodology with prior literature, I felt further analysis based on some hypotheses would help augment the novelty of the manuscript. For e.g. further time series analysis of the meteorological data may explain what triggered the sudden drainage event in 2017 shown in figure S9, and more broadly the periods of maximum loss in other lakes. Similarly, some exploration into the spatial variability, for e.g. why eastern Canada is different from other regions (lines 243-245, S8 or other regions in Table S1) would be useful. How was it determined that this was "due to differences in the physiological traits of plant species adapted to these regions". Lines 288-303 came across as suggesting that increased vegetation in drained lakes was a positive outcome in terms of acting as a net carbon sink, mitigating emissions and stabilizing sediments, but the negative consequences of such dramatic land cover change could also be discussed.

Finally, I would strongly recommend that the code be made public for this work to be evaluated and reproduced.

Minor comments

Introduction Line 40-43 – the general reader would benefit from an explanation of the difference between thermokarst and non-thermokarst lakes, and why the distinction is necessary. You could consider moving lines 75-79 up into the introduction.

Fig S1 – if Olsen et al. 2001 is the source for the ecoregions, it must be cited.

Fig S6 It would be useful to have the figure labels (A, B, etc.) mentioned in the caption

This is an example where I wondered whether the ERA5 resolution would allow clear delineation of thermokarst vs. non thermokarst lakes.

I. Response to comments and suggestions of Reviewer #1

This study is an important step forward and very well presented. This study provides a holistic analysis of drained lakes in the Arctic, providing a database of drained lakes, correlation of draining with climate and environmental variables, and analysis of subsequent vegetation growth. The authors found marked differences between thermokarst and non-thermokarst lakes in the correlating variables with a greater dependence on temperature and water, respectively, and in the subsequent vegetation with thermokarst lakes having a greater intensity of vegetation. By analyzing thermokarst and non-thermokarst with identical methods, the variations can be effectively compared.

We sincerely appreciate the time and effort you have dedicated to reviewing our work. We would like to express our gratitude for recognizing the quality and significance of our study. Your positive feedback motivates us to continue our efforts in advancing research in this field. Thank you very much.

Below are specific comments with most of them just requesting greater clarity. The one methodological change I would request is to calculate the NDVI using the reflectances from a single image rather than likely two different images. Great work on an awesome study.

We sincerely appreciate the specific comments and positive feedback provided by the reviewer. We have carefully considered the suggestions provided and have taken steps to address each comment in order to enhance the clarity and quality of our manuscript.

Regarding the methodological change request, we have recalculated the NDVI using reflectance values from a single image (using the 90th percentile to obtain a value similar to the second highest NDVI), which will enhance the accuracy and reliability of our vegetation analysis.

Your valuable feedback has played a crucial role in refining our work, and we sincerely appreciate the thoroughness with which you reviewed our study. Thank you for helping us improve the overall quality of our research.

9: “previously overlooked” meaning all non-thermokarst not a subset of them. Can you make this more clear? Appreciate the feedback.

10-12: The second two clauses are difficult to pair with the first before reading the paper. Also “upward trends” of what? Perhaps: increasing rates of drainage.

Thanks to your suggestion. In the new manuscript we have updated the abstract (revisions bolded) and this sentence has been removed (we have classified the drained lakes as very likely, likely, and unlikely thermokarst lakes).

Abstract: “Widespread lake drainage can lead to large-scale drying in Arctic lake-rich areas, affecting hydrology, ecosystems and permafrost carbon dynamics. To date, the spatio-temporal distribution, driving factors, and post-drainage dynamics of lake drainage events across the Arctic remain unclear. Here, we have identified over 35,000 (**~0.6% of all lakes**) lake drainage events in the northern permafrost zone between 1984 and 2020, with approximately half being **relatively understudied** non-thermokarst lakes. **Smaller, thermokarst, and discontinuous permafrost area lakes are more susceptible to drainage compared to their larger, non-thermokarst, and continuous permafrost area counterparts. Over time, discontinuous permafrost areas contribute more drained lakes annually than continuous permafrost areas.** Following drainage, vegetation rapidly colonizes drained lake basins (DLBs), with thermokarst DLBs showing significantly higher vegetation growth rates and greenness levels than non-thermokarst DLBs. **With warming, DLBs are likely to become more prevalent and serve as greening hotspots,** playing an important role in shaping Arctic ecosystems.”

25: “net gain” of what?

Sorry for the confusion, we have modified the sentence. Lines 25–27: "Satellite observations^{8–10} indicate that Arctic lake-rich areas have experienced **a decline in lake area** over the last 20 years, **contrary to** model projections^{11,12} that suggested **an increase** due to widespread permafrost thaw."

53: The lake drainage events were detected by the existing products (JRC/GLAD). Here you identify them in those products and then detect the drainage year by satellite remote sensing.

Yes, the existing products (JRC/GLAD) detected water loss **pixels**, we identified the presence of drained lakes through GEE's object-based analysis method and then identified the year of lake drainage through LandTrendr algorithm.

We have modified in Lines 65–69: "In this study, we conducted a comprehensive analysis of lake drainage events across the northern permafrost zone over a span of three and a half decades (1984–2020). **By leveraging existing surface water products^{36,37} and satellite remote sensing**, we accurately detected these events, allowing us to delineate their spatial distribution and determine the corresponding drainage years."

85: "high spatial variability" of what?

Thanks for the correction. We have revised in Lines 97–99: "Based on the ecoregion delineation³⁸ (fig. S4 and table S1), we categorized the region and performed a statistical analysis of both the count and drainage probabilities of lakes, **revealing notable spatial variability in the distribution of lake drainage events** (table S2)."

Fig 1: It is hard to glean from D all that is there and I would love to be able to see it better. The colors seem to bleed together. Can you add more contrast? Perhaps yellow instead of purple.

Thank you for bringing this to our attention. We appreciate your feedback and acknowledge the need for improved clarity in Fig 1. We have made color adjustments to enhance the contrast and improve visibility.

Fig. 1. Spatial distribution of lake drainage events detected by remote sensing in the northern permafrost zone. (A–C) Satellite images of an Arctic lake at 164°45' W, 66°27' N showing the occurrence of a lake drainage event and subsequent vegetation growth. (D) Map of lake drainage events ($n = 35,337$) in the northern permafrost region during 1984–2020, categorized by initial lake size (small, medium, and large). The bottom map illustrates the permafrost zonation, with thermokarst lake landscape highlighted. (E–G) Enlarged maps illustrating the diversity of drained lake distribution in different regions: southern permafrost margin at the Russia-Mongolia border, Northeast Siberian coastal tundra, and St. Lawrence Island. (H) Spatial density of drained lakes in relation to permafrost extent, thermokarst lake likelihoods, ground ice content, and the Yedoma region. Yedoma is an organic-rich, ice-rich Pleistocene-age permafrost found primarily in eastern Siberia, Alaska, and the Yukon.

The dashed line shows the average reference for the entire study area. C: continuous, D: discontinuous, S: sporadic, I: isolated. Refer to fig. S5 for classification distribution patterns and fig. S6 for the proportion of drained lakes of different sizes in each region.

Fig S6: Are these lines the median over all the drained lakes per year?

Fig 2 and S7: These figures would be more valuable with the points colored by year.

In the revised manuscript, we have delved deeper into the environmental driving factors of lake drainage, thus these figures have been replaced. Please refer to the sections "Temporal patterns in lake drainage events for 2001–2020," "Environmental drivers of Lake drainage events," and "Statistical analysis of lake drainage probabilities" from lines 145 to 294 of the manuscript.

200: "approximately 15 years" makes me think it goes up after that. Is that true? Perhaps for at least 15 years? Or for the 15 years we were able to analyze?

Appreciate the feedback, we have modified in Lines 318–323: " In contrast, the NDVI of likely thermokarst DLBs reached a stable state similar to the surrounding vegetation around the 6th year after drainage, while the NDVI of unlikely thermokarst DLBs exhibited very slow growth, remaining significantly lower than the surrounding levels even 15 years after drainage (Fig. 5J). **Note that the vegetation dynamics beyond our analysis period remain uncertain due to the limitations of observations.** "

299: "analyzed cases where..." which means those cases existed but doesn't tell what you found or how prevalent they were. Engage more deeply with references 23 and 27 that made this case (per line 49). Your results seem to say that there is high variability and for a little more than half of thermokarst lakes the vegetation was more luxuriant, but this did not hold for the majority of non-thermokarst lakes. There are many intricacies playing into the vegetation cover rates as you describe elsewhere. State the conclusions we can and cannot draw from this study on the question of whether vegetation is more luxuriant.

We appreciate the suggestions from the reviewers. In the revised manuscript, we

have introduced a new section titled "Impacts of Lake Drainage and Vegetation Growth in DLBs" on lines 425–503 to provide a summary of our main findings and implications for further research.

The amendments corresponding to this comment are in Lines 495–500: "By tracing vegetation dynamics following lake drainage, we have found that vegetation greenness in DLBs exhibited high spatial variability, **and analyzed the differences in greenness between DLBs and surrounding vegetation under various environmental conditions. Our findings indicate that DLBs in thermokarst lakes, larger lakes, and lakes with higher drainage ratios exhibit higher vegetation greenness compared to DLBs in non-thermokarst lakes, smaller lakes, and lakes with lower drainage ratios (Fig. 5).**"

312-316: The water transition map of JRC and the water dynamics map of GLAD are quite different. The JRC map only looks at the first and last year and ignores all the intervening years except when the start and end years were land and then the intervening years are checked for ephemeral permanent (year-round) or seasonal water. Thus cyclical water inundation could be captured as permanent/seasonal water, water loss, water gain, or ephemeral water. Whereas, the GLAD layer evaluates all the years for all pixels, and only pixels with a stable trajectory of loss are included in the "water loss" class. For this study, both products provide valuable insights. However, the difference between the products should be made more clear.

We appreciate your insightful comments regarding the differences between the JRC water transition map and the GLAD water dynamics map. We acknowledge that clarifying the distinctions between the two products is important, and we have revised our manuscript to provide a more explicit explanation of these differences.

Lines 510–530: "We identified drained lakes in permafrost regions by selecting pixels labeled as "lost permanent" in the JRC water transition map and as "water loss" in the GLAD water dynamics map, which both represent the transition from lakes to DLBs. **However, there are notable differences in the identification methods used by the two datasets.**

The JRC map identifies the start year of transition for each water pixel as the first year between 1984 and 2000 that has sufficient observations to characterize the presence of water, while the GLAD map uses 1999 as the start year. The JRC map identifies water pixel changes based on the initial and final states of the time period, with intervening years considered for the presence of ephemeral permanent (year-round) or seasonal water only if the initial and final years represent land. Consequently, the JRC map can classify cyclical water inundation as ephemeral water, permanent/seasonal water, water loss or water gain. In contrast, the GLAD map evaluates seasonal and interannual fluctuations of water pixels over the entire time series, classifying only pixels with a stable trajectory of loss as “water loss”. To mitigate short-term annual anomalies and inter-annual observation variability, the GLAD dataset employed a 3-year mean moving window to smooth the annual open water percentage time-series. This smoothing technique introduces the possibility of misclassifying "water loss" as a "wet period" (land-water-land).

Both the GLAD and JRC maps offer valuable insights for this study, with the GLAD map demonstrating greater accuracy in detecting water losses compared to the JRC map. It is worth noting that while the JRC and GLAD products mark the spatial distribution of these water loss pixels, they do not provide information on the timing of transition from permanent water to land for each pixel."

331: Please provide more details on the “morphological opening operations”. I’m assuming this is an erosion followed by a dilation. What was the scale of these operations?

Yes. We have modified in Lines 539–541: "Morphological operations, **involving erosion followed by dilation with a 1-pixel radius and 2 iterations**, were conducted on the water extent raster image to separate finely connected water bodies and eliminate isolated pixels."

345: LandTrendr was only applied to the lakes with drainage as delineated by the JRC/GLAD maps, correct? Make this more explicit. Together with lines 63-67 this is not completely clear.

Yes. We have made the suggested revisions.

Lines 622–625: "**Based on the JRC/GLAD maps and object-based lake analysis method, we identified lakes that underwent drainage events. We then employed the Landsat-based Detection of Trends in Disturbance and Recovery (LandTrendr) algorithm⁴⁵ to detect the main occurrence year of lake drainage events.**"

Lines 77–83: "Using an object-based image analysis approach with well-established surface water products^{36,37}, we **delineated** lake objects in the northern permafrost region **and identified those that** experienced varying degrees of water loss during 1984–2020. Unlike the pixel-based analysis of surface water dynamics, our lake-object-based analysis **allows for the identification** of specific lake drainage events, **enabling us to calculate regional lake drainage probabilities and provide more detailed insights into the primary drainage years** and post-drainage vegetation dynamics for individual lakes."

Lines 142–143: "**Based on the identified lake drainage events**, we employed a temporal segmentation and change detection algorithm^{45,46} to **determine the primary years of drainage for individual lakes.**"

352-360: Revise this paragraph. Very important information, but as written the sentences do not connect well with one another. Also, here you state you use the entire archive whereas later you describe how you filtered the images and pixels for the NDVI assessment. Did you do any of those same steps here or every single image was used as is for LandTrendr?

Thank you for your feedback. We apologize for any confusion caused by this paragraph. We employed the same data acquisition and preprocessing procedures for the Landsat imagery archives used in both the LandTrendr algorithm and NDVI-based assessment. We have added a section titled "Acquisition and preprocessing of Landsat data" before the sections "Detection of lake drainage year" and "Trace vegetation dynamics in DLBs". We hope that this revision will enhance the clarity and coherence of our description.

Lines 587–619: "

Acquisition and preprocessing of Landsat data

This study utilized long-term time series of orthorectified Landsat surface reflectance imagery archives, provided by the GEE platform⁷⁷, to detect the occurrence years of lake drainage events and track vegetation growth dynamics in DLBs. The complete coverage data spanned from 2000 to 2020 and was derived from Landsat-5 Thematic Mapper (TM), Landsat-7 Enhanced Thematic Mapper-plus (ETM+) and Landsat-8 Operational Land Imager (OLI). Regions like Alaska have had enough observation density to analyze the seasonal and interannual dynamics of surface water since 1984, but the first observation in Siberia was not until around 2000³⁶. Therefore, despite the JRC product providing valuable isolated observations of surface water dynamics prior to 2000, our evaluation of lake drainage and vegetation growth dynamics is limited to the period 2001–2020.

We utilized an open-source toolkit (<https://jdbcode.github.io/EE-LCB/>) called Earth Engine Landsat Collection Builder for preprocessing the satellite image archive. This toolkit is deeply integrated into the GEE platform and is specifically designed for Landsat image preprocessing, offering analysis-ready images through customizable preprocessing chains. The preprocessing workflow employed in this study encompassed image filtering, cloud and shadow masking, sensor harmonization, multispectral index calculation, and image compositing.

Landsat images were first screened for acceptable data quality, with cloud coverage less than 50% to reduce cloud and shadow interference, and acquired during June to September each year to maintain phenological consistency and minimize the impact of ice and snow. The quality assessment band (pixel_qa) of Landsat images was then used to mask out observation noise such as clouds, ice and snow, and shadows in the image collection to improve image quality. Landsat images from different sensor sources (TM, ETM+, and OLI) were processed based on the statistical transformation function to correct spectral differences and improve spectral continuity of the images. Three multispectral indices were calculated at pixel-level to detect long-term changes in lakes and vegetation, including the Tasseled Cap Greenness index (TCG), the Normalized Difference Vegetation Index (NDVI), and the Automated Water Extraction Index (AWEI). TCG and NDVI provide insights into

vegetation status and greenness⁵⁴, while AWEI is useful for accurately distinguishing between water and non-water pixels⁸⁰. Finally, for each pixel and each index of the processed Landsat image collection, annual composite images are generated using the percentile composite method (ee.Reducer.percentile). Note that multispectral indices are calculated prior to image compositing to ensure the reliability of analysis based on individual observations. The processed data forms the foundation for consistent monitoring of lake dynamics and quantification of vegetation changes over a 20-year period."

377: You have two sets of pairs (users/producers, JRC/GLAD) which makes it difficult to interpret the respectively. I initially interpreted the first two as JRC and the second two as GLAD, but the next sentence seems that it is paired by accuracy. Also, perhaps add "per pixel" (The reported per pixel...) to clarify this vs the validation of lakes as an object that is next.

Thanks for pointing this out, the previous descriptions can indeed be confusing.

We have clarified in Lines 645–651: "The JRC and GLAD products reported the following accuracies **on a pixel-by-pixel basis: JRC with a user's accuracy of 49.8% ($\pm 19.3\%$) and a producer's accuracy of 65.5% ($\pm 11.4\%$), and GLAD with a user's accuracy of 30.0% ($\pm 6.5\%$) and a producer's accuracy of 86.2% ($\pm 7.4\%$)**. The GLAD product exhibits a higher producer's accuracy (corresponding to a lower omission error) compared to the JRC product in capturing the transition from water to land, although this comes at the cost of a lower user's accuracy (corresponding to a higher commission error)."

383: Is this event/lake based or pixel based?

We have revised in Line 654: "Validation of detected lake drainage events was performed **on an individual lake basis** through..."

389: 10% randomly selected? State that.

Yes, we have revised in Line 667: "we **randomly selected and** validated 10% of the 35,337 detected lake drainage events..."

391: "accuracy of detection" means the percent of lakes that actually drained vs all those mapped as drained? And the temporal accuracy means the percent of mapped drained lakes where the majority drain year was correctly identified?

Yes, thanks for your clarification. We have made the necessary revisions based on your comments.

Lines 670–675: "The results of spatial validation for drained lakes showed that the accuracy of detection (**i.e., the percentage of lakes that actually drained versus all lakes mapped as drained**) for small and medium-large drained lakes was approximately 82.6% and 96.1% respectively. The temporal accuracy (**i.e., the percentage of mapped drained lakes where the major drainage year was correctly identified**) of the LandTrendr algorithm's detection year for small and medium-large drained lakes was found to be 63.8% and 89.2%, respectively"

406: What years are these datasets from? Does this impact your results at all?

Thank you for bringing it to our attention. We have carefully considered this aspect and made revisions to the manuscript to emphasise this issue.

Lines 555–558: "We utilized the most up-to-date data products (fig. S5) available on thermokarst lake coverage (released in 2016)³³, permafrost extent (2019)⁴⁰, Yedoma region (2021)³⁹, ground ice content (2002)⁴¹, soil carbon content (2013)⁷⁸ and soil nitrogen content (2020)⁷⁹ to investigate the influence of permafrost-related properties on lake drainage and post-drainage vegetation growth."

Lines 571–574: "While it is worth noting that these datasets do not provide temporal information, we would like to emphasize that the overall impact on our analysis is generally limited. This can be attributed to the relatively modest changes in permafrost-related properties over the course of a decade, with the exception of ground ice content, which represents the only dataset currently available."

Lines 267–271: "Considering that ground ice content dataset covering the northern permafrost region have not been updated for over 20 years, we need to be mindful of the timeliness of the data and its impact on the current research findings. In future

studies, updating ground ice content data will contribute to a more accurate assessment of the relationship between ground ice content and lake drainage probabilities⁴."

451: NDVI should be calculated from the NIR and Red reflectance of a single observation. You are taking the median red value and the median NIR value and calculating NDVI to get a figure for the year. However, there is a good chance that the median NIR and the median red come from different dates. Thus it could be combining different vegetative states as well as any differences in atmospheric contamination/correction. Instead, you should calculate the NDVI for all clear observations within the summer and then take the median. Or perhaps even better take something like the second highest NDVI (may perform better than the max to account for missed shadows)

We sincerely appreciate your valuable suggestion regarding the calculation of NDVI. You are correct in pointing out the potential issue with using median values of NIR and Red reflectance, as they might come from different dates.

We have updated the NDVI calculations as suggested.

Lines 614–619: "Finally, for each pixel and each index of the processed Landsat image collection, annual composite images are generated using the percentile composite method (ee.Reducer.percentile). **Note that multispectral indices are calculated prior to image compositing to ensure the reliability of analysis based on individual observations.** The processed data forms the foundation for consistent monitoring of lake dynamics and quantification of vegetation changes over a 20-year period."

Lines 686–704: "Trace vegetation dynamics in DLBs

To trace the dynamics of vegetation growth following lake drainage, within the preprocessing of Landsat image archives, **we utilized the 90th percentile of NDVI values from annual image collections to generate a robust NDVI time series. This approach yields a value resembling the second highest NDVI value, which might perform better than the maximum and median NDVI values considering the potential influence of shadow masking and standing water in the lake basin.**

For each drained lake, we pixel-wise acquired NDVI values within the lake basin area, masked out values below 0, and computed the median NDVI for the region. This step further mitigated the impact of standing water on NDVI assessment in DLBs. We also calculated the median NDVI of the surrounding vegetation to compare its greenness difference with DLBs. The surrounding vegetation was defined by a circular buffer of twice the lake diameter centered at the lake center, and masked to exclude the lake and other water bodies. Lastly, based on the identified drainage year for each lake, we transformed the time series of NDVI values from 2000 to 2020 into vegetation dynamics recorded by years after drainage.

Additionally, we applied the same approach to extract the TCG index from both DLBs and the surrounding areas for comparison (fig. S8). The results indicate that despite being based on different image bands and algorithms, the vegetation greening trends and patterns reflected by TCG are basically consistent with those of NDVI, underscoring the reliability of our method in capturing the vegetation dynamics in DLBs."

494: many of these links gave 404 responses.

Thank you for notifying us about the broken links (which was caused by line breaks in the PDF), and we apologize for any inconvenience this may have caused. We have rectified the problem by checking and updating the links in the manuscript to ensure that they are functional and accessible.

We would like to express our sincere gratitude once again for your valuable feedback and insights.

II. Response to comments and suggestions of Reviewer #2

The manuscript “Tracing Arctic lake drainage events and emergent vegetation in drained lake basins” identifies lake drained lakes across the northern permafrost zone and tracks NDVI trends in these basins following drainage. The authors then calculate trends in the frequency of drainage events and identify the environmental variables that covary with drainage event frequency. Lastly, the authors compare NDVI within drained lake basins to the surrounding vegetation and identify the environmental variables that explain variation in NDVI.

We sincerely appreciate your thoughtful review of our manuscript. Your valuable feedback and insights have provided us with important perspectives for further enhancing the quality and impact of our research. We acknowledge your comments regarding the presentation, interpretation, and discussion of results, as well as the need for additional references to strengthen the manuscript's context and contribution.

Mapping drainage events at such a large scale is an important forward step in our understanding of Arctic lake hydrology. In general, the methods are sound, but there are a few analyses that need improvement. However, the presentation, interpretation, and discussion of the results is under-developed, the text is woefully under-referenced, and I am left wondering what the study adds to our understanding of Arctic ecosystems.

We are committed to addressing your suggestions and making the necessary improvements to ensure that our study provides meaningful insights into Arctic lake hydrology and ecosystem dynamics. Your input is highly valuable to us, and we are grateful for your time and effort in reviewing our work.

Below, I detail the under-development of the results, interpretation, and discussion of each subsection.

Spatial patterns of drained lakes in the Arctic. Here, the authors describe the size and area of DLBs over the entire study domain as well as provide some eco-region specific results. However, without describing the size distribution of existing lakes over the domain (or

specific ecoregion), it is impossible to interpret the meaning of the size distribution of DLBs. One question that comes to mind but was not answered in the text is: are large or small lakes more likely to drain? A similar argument can be made for comparisons between regions; if all regions exhibited the same lake size distribution, it would be notable that some regions have smaller DLBs and some have larger. However, not all regions have the same lake size distribution. It is therefore unclear how to interpret the region-specific results; is the fact that some regions exhibit smaller DLBs simply a reflection of the fact that the lakes in that region are smaller, or is there something about the landscape that makes smaller lakes in this region more likely to drain? Furthermore, only three ecoregions across the entire study domain are highlighted with no explanation as to why they were chosen or what their significance is.

We sincerely appreciate your insightful comments. Your questions about whether larger or smaller lakes are more prone to drainage and your inquiries into the interpretation of region-specific results are important considerations that we have comprehensively addressed in the revised manuscript.

Lines 10–12: "Smaller, thermokarst, and discontinuous permafrost area lakes are more susceptible to drainage compared to their larger, non-thermokarst, and continuous permafrost area counterparts."

The specific analysis is in Lines 224–286: "

Statistical analysis of lake drainage probabilities

Since attributes such as permafrost extent, ground ice content, thermokarst lake likelihoods, and whether Yedoma region are considered time-invariant categorical features in the machine learning model, their relative importance may be underestimated. Given the known importance of these landscape variables in driving surface water changes associated with permafrost degradation, we conducted an enumeration of all lake objects within our study area and calculated lake drainage probabilities for various sizes of lakes and geographical regions (Fig. 4 and table S2). A total of approximately 5.83 million lakes (with an area larger than 1 ha) were identified across the northern permafrost region. The overall average drainage probability for lakes during the study period was 0.61%, while the drainage

probabilities were 0.64% for small lakes, 0.50% for medium lakes, and 0.38% for large lakes, indicating a relatively lower likelihood of drainage for larger lakes. One potential explanation for this is that smaller lakes, due to their shallower water columns and lower water storage capacity²⁹, are more susceptible to lateral drainage events triggered by extreme precipitation or rapid snowmelt. In contrast, larger lakes have the capacity to accumulate more heat and water, thus showing greater resilience against hydrological disturbances²⁸. This provides a thermal inertia and water storage buffer when temperatures and precipitation fluctuate, enabling larger lakes to better maintain thermal equilibrium and shoreline stability compared to smaller water bodies.

Significant spatial heterogeneity in lake drainage probabilities has been observed across ecoregions (table S2), highlighting the complexity of lake drainage dynamics. For instance, within the Canadian Aspen forests and parklands (spanning 1.6×10^4 km²), 16.33% of lakes experienced drainage, while in the Central Canadian Shield forests (spanning 16.5×10^4 km²), only 0.11% of lakes underwent drainage. This diverse distribution of lake drainage probabilities underscores the intricate interplay of local environmental factors. Further analysis indicates that lake drainage probabilities are associated with thermokarst lake likelihoods, permafrost extent, and ground ice content (Fig. 4). Notably, for very likely thermokarst lakes and the Yedoma region, the drainage probabilities for lakes of all sizes exceed the regional averages, with large lakes anomalously having higher drainage probabilities than small and medium-sized lakes. Large drained lakes are disproportionately concentrated in these areas, with 6–8 times the average density (Fig. 1H) and 3–4 times the average drainage possibility (Fig. 4B&D), suggesting well-developed erosional drainage systems may have formed. The prevalent landscape in these areas is a mosaic of lakes and streams, interconnected through an underground network of ice^{15,42}. When expanding lakes encounter the ice network, thermal erosion along the network results in the melting of ice wedges, creating drainage channels through which the lake water drains^{7,34}.

The drainage probabilities for likely thermokarst lakes fall between those for very

likely and unlikely thermokarst lakes (Fig. 4D), consistent with existing knowledge that thermokarst lakes are more susceptible to drainage⁴. The impact of ground ice content on lake drainage probabilities is relatively intricate (Fig. 4F). In regions with medium ground ice content, the overall lake drainage probability is the highest, approximately 1.4 times the average level. In regions with high and low ground ice content, the overall lake drainage probabilities are slightly below average, while the drainage probabilities for large lakes in these regions are 1.6 and 0.5 times the average, respectively. Variations in the probability of lake drainage under different ground ice content levels may be associated with the thermal inertia of latent heat fusion and the formation of subsurface drainage channels^{28,50}. Considering that ground ice content dataset covering the northern permafrost region have not been updated for over 20 years, we need to be mindful of the timeliness of the data and its impact on the current research findings. In future studies, updating ground ice content data will contribute to a more accurate assessment of the relationship between ground ice content and lake drainage probabilities⁴.

Furthermore, the discontinuous permafrost zone exhibits the highest overall lake drainage probabilities (more than double the average; Fig. 4H), with drainage probabilities for small, medium, and large lakes at 1.33%, 1.28%, and 0.78% respectively. There are over 13,000 drained lakes in the continuous permafrost zone (Fig. 4G), but the likelihood of lake drainage in this area is lower than the average level (Fig. 4H). However, the susceptibility is rising in certain southern areas where permafrost is becoming more discontinuous due to climate warming^{9,20}.

Fig. 4. Drainage probability analysis of lakes across various lake sizes and regions. Analysis of drained lake area distribution and drainage probabilities for lakes of different sizes in (A, B) the Yedoma region and the entire study area, along with different classifications of (C, D) thermokarst lake likelihoods, (E, F) ground ice contents, and (G, H) permafrost extents. The left-hand panels depict kernel density estimation plots with a logarithmic x-axis to display lake areas, along with upper

boxplots presenting statistical information and numerical annotations denoting the number of drained lakes. The right-hand panels present radar charts illustrating the percentage of drained lakes relative to the total number of lakes within the respective regions."

The method description is on Lines 534–542: "The detection of lake drainage events requires calculating the proportion of lake drainage at the lake object level, but existing water products are in raster format and only provide pixel-by-pixel water loss information. We used the object-based analysis function provided by the Google Earth Engine (GEE) cloud platform⁷⁷ to create lake objects and identify lake drainage events. First, we used the JRC product's thematic map of water extent as a mask to identify persistent water. Morphological operations, involving erosion followed by dilation with a 1-pixel radius and 2 iterations, were conducted on the water extent raster image to separate finely connected water bodies and eliminate isolated pixels. We converted the set of connected pixels into water objects with unique identifiers via GEE and calculated the number of water pixels for each lake object."

Lines 575–580: "In this study, the spatial density of drained lakes (Fig. 1H) was calculated by overlaying the spatial layers delineated from Collected data products with the detected drained lakes, representing the number of drained lakes per unit area in different regional units. The probability of lake drainage (Fig. 4) is calculated based on the ratio of drained lakes to all lakes within a specific region unit. These methods enable us to visually depict the spatial distribution pattern and occurrence probability of lake drainage events across various regions."

In addition, regarding the selection of ecoregions in Figure 1, Lines 97–109:

"Based on the ecoregion delineation³⁸ (fig. S4 and table S1), we categorized the region and performed a statistical analysis of both the count and drainage probabilities of lakes, revealing notable spatial variability in the distribution of lake drainage events (table S2). For instance, along the southern permafrost margin at the Russia-Mongolia border, this area is densely populated with thousands of drained

lakes (Fig. 1E), even though it is not situated within the lake-rich zone. Here, the lake drainage probability reaches nearly 8%, which is more than ten times higher than the average level across the entire permafrost zone. In contrast, in the Northeast Siberian coastal tundra ecoregion (Fig. 1F), although the overall lake drainage probability is lower at one-third of the regional average, large lakes have a drainage probability twice the average, leading to more than half of the drained lakes being medium to large in size. In addition, we have discovered previously underreported spatial clusters of drained lakes, such as St. Lawrence Island in the Bering Sea, where the density of drained lakes is approximately 80 times that of the entire permafrost zone, with 655 drained lakes densely distributed over an area of 4,640 km² (Fig. 1G)."

Trends of lake drainage events for 2001-2000. The authors report an increase in thermokarst DLB occurrence over the study period, with a slope of 19. However, this slope is not interpreted; does this mean that, on average, each year sees an increase of 19 DLBs across the domain? Because that seems like a very small number and, even if statistically significant, is not ecologically significant. But the authors do not guide the reader here and we are left to wonder that question as well as others like: does the trend in DLB occurrence vary by region, permafrost extent, or ground ice extent?

Thank you for your thoughtful feedback. We conducted an analysis, as suggested, to examine whether trends in lake drainage occurrence vary by permafrost extent, thermokarst lake likelihood, or ground ice content.

Lines 141–180: "

Temporal patterns in lake drainage events for 2001–2020

Based on the identified lake drainage events, we employed a temporal segmentation and change detection algorithm^{45,46} to determine the primary years of drainage for individual lakes. We identified 6,858 and 28,479 lake drainage events for the periods 1984–2000 and 2001–2020, respectively. However, the fractured nature of Landsat observations across space and time³⁶ prior to 2000 limited the detection of lake drainage events that occurred during 1984–2000. Therefore, in this study we only analyzed changes in the frequency of lake drainage events over the period 2001–

2020 (Fig. 2). The results showed that the mean annual count of drained lakes in the northern permafrost region was 1424 (range: 767–2073), with a standard deviation of 332. Despite annual fluctuations, the lake drainage frequency exhibited a non-significant ($p=0.08$), slight upward trend (with a slope of 23) throughout the study period.

Regional statistics show that temporal peaks in lake drainage frequency are not synchronized across landscape attributes (Fig. 2). For example, in 2020, the number of drained lakes peaked in areas with low ground ice content (1,329) and for unlikely thermokarst lakes (1,305), while drained lakes in other categorized regions remained close to the annual average level. In contrast, in 2016, likely (495) and very likely (630) thermokarst lakes had a peak drainage frequency, while the number of unlikely thermokarst drained lakes was below the annual average. The elevated count of drained lakes in these specific years suggests the potential prevalence of triggering factors for lake drainage. Thermokarst and non-thermokarst lakes making distinct contributions to total drainage in various years. Despite comprising over half of all drained lakes (Fig. 2C), non-thermokarst lakes have garnered relatively less research attention⁴. The prevailing emphasis on thermokarst-related phenomena^{7,11,33} has led to a comparative scarcity of studies exploring the characteristics, dynamics, and ecological implications of these non-thermokarst lakes. Given their substantial number and potential contributions to our understanding of diverse ecosystems, there exists a compelling need for future investigations to delve into the unique dynamics and roles of these less explored lake types.

From 2001 to 2020, lake drainage events exhibited significant upward trends ($p < 0.001$) in discontinuous permafrost zones and very likely thermokarst lakes, with slopes of 21 and 14, respectively. This indicates that over the 20-year period, the mean annual number of drained lakes increased by 420 in discontinuous permafrost zones and 280 for very likely thermokarst lakes. Over time, the contribution of the discontinuous permafrost zone to the annual count of drained lakes gradually surpasses that of the continuous permafrost zone (Fig. 2F), suggesting that the impacts of climate change and permafrost degradation on the stability of lakes in

these sensitive regions appear to be intensifying^{24,34,42}.

Fig. 2. Annual variation in the frequency of lake drainage event during 2001–2020. Stacked and percentage stacked bar charts illustrating the frequencies of lake drainage events over time under different classifications of (A, B) thermokarst lake likelihoods, (C, D) ground ice contents, and (E, F) permafrost extents. The occurrence year of a lake drainage event was determined as the year with the maximum drainage proportion. "

One major result (highlighted in the abstract) is that summer soil temperature is a major driver of the trend in thermokarst DLB occurrence frequency. This claim is based on soil temperature data from the ERA-5, which, according to a previously published evaluation of this dataset, "... are not well suited for informing permafrost research" (Cao et al., 2020). Given that the data informing their results may not be valid, I question the reliability of the results presented here. The authors do not provide any justification for using this dataset. Furthermore, this finding seems to be based on correlation coefficients between

climate variables and DLB occurrence frequency. Given that multiple environmental variables, many of which are likely correlated, likely lead to DLB occurrence frequency, analysis of variables individually is not an appropriate approach. One option would be multiple regression analysis, where all environmental variables would be considered together, residuals would be checked, etc.

We sincerely thank you for pointing out these important issues.

Regarding the validity of ERA5-Land data, we carefully reviewed the literature provided (Cao et al., 2020) and contacted the corresponding author, Dr. Donghai Zheng. In our communication, Dr. Zheng explained that their study solely assessed the soil temperature data from ERA5-Land and identified deviations primarily during the winter season. For the study across the northern permafrost region, Dr. Zheng acknowledged that ERA5-Land remains the best available reanalysis dataset.

Furthermore, in accordance with the recommendation of Reviewer 3, we conducted a comparison between ERA5-Land data and the 1km resolution Daymet data for the North American region, revealing a high degree of consistency. In the Methods section, we have provided justifications for utilizing the ERA5-Land dataset.

Lines 706–727: "

ERA5-Land reanalysis data

Due to the lack of fully covered high spatial resolution meteorological products in the northern permafrost regions, we used ERA5-Land monthly reanalysis dataset⁸⁵ for the period 2000-2020 to analyze the climate factors affecting lake drainage and vegetation growth. ERA5-Land is a land-enhanced product from the fifth generation of European ReAnalysis (ERA5), with a spatial resolution of 0.1° (or 9 km). It is considered a state-of-the-art global reanalysis dataset for land applications that efficiently reconstructs surface states and process parameters through advanced data assimilation techniques⁸⁵. A recent study noted that ERA5-Land showed a high degree of global consistency with MODIS satellite products in terms of surface temperature⁸⁶.

Despite the relatively coarse spatial resolution of ERA5-Land, the analysis based on a 0.1° x 0.1° grid reveals that 35,337 drained lakes are distributed in 20,132 grid

cells, most of which contained 1–2 drained lakes (fig. S10A). Over 99% of the grid cells exhibit fewer than 10 drained lakes (fig. S10B). The highest count of drained lakes in a grid cell is 33, occurring at St. Lawrence Island (Fig. 1G). Therefore, for lake drainage events occurring between 2000 and 2020, ERA5-Land can provide sufficiently differentiated climatic information for lake drainage prediction and post-drainage vegetation dynamics analysis.

Furthermore, we evaluated the temperature and precipitation of ERA5-Land dataset using the 1 km resolution Daymet dataset⁸⁷ covering North America (fig. S11). This analysis compares climate data of Daymet and ERA5-Land for the year in which the lakes were drained using 16,104 drained lakes detected in North America as sample points. The results indicate strong overall consistency and low variability between Daymet and ERA5-Land reanalysis data in simulating temperature and precipitation (fig. S11)."

Fig. S10. Grid-based statistics of drained lakes. (A) Distribution map of drained lakes based on a 0.1°x0.1° grid. (B) Pie chart statistics of grid counts with drained lakes. A total of 35,337 drained lakes are distributed across 20,132 grid cells, with the majority of cells containing 1–2 drained lakes. More than 99% of grid cells have fewer than 10 drained lakes.

Fig. S11. Bland-Altman plots for assessing the consistency of ERA5-Land and Daymet reanalysis data. Evaluation of (A) mean annual air temperature, (B) mean summer air temperature, (C) annual precipitation, and (D) summer precipitation. The analysis is based on a total of 16,104 drained lakes detected in North America, using climate data for the year of drainage for comparison. As Daymet dataset does not provide mean air temperature, a simple estimation was made using the average of maximum and minimum air temperature. Bland-Altman plots visually display the average difference and variability between the two datasets. The x-axis represents the average values of the two datasets, and the y-axis shows the differences. The solid red line represents the mean difference, while the dashed red line indicates the 95% limits of agreement. Scatter density is shown in different colours, with yellow representing areas of high concentration of sample points. Overall, ERA5-Land and Daymet reanalysis data exhibit strong consistency in simulating temperature and precipitation, with low variability.

To address the issue of collinearity among variables, as suggested by Reviewer 3, we have employed a more advanced machine learning classification model to predict lake drainage and analyze the environmental driving factors.

Lines 181–223: "

Environmental drivers of Lake drainage events

To reveal the key environmental drivers behind lake drainage events from a variety of candidate explanatory variables such as climate, topography, and permafrost characteristics (table S3), we developed a binary classification model to predict lake drainage. We conducted a comprehensive evaluation of model performance using multiple metrics, which showed the model demonstrates strong classification capabilities (Fig. 3A&B), effectively utilizing explanatory variables to predict whether an individual lake will experience drainage. Further feature importance assessment indicates that trend slope of annual air temperature, annual air temperature and elevation are the most important drivers of lake drainage events (Fig. 3C). The effect of annual air temperature trend on lake drainage events is nonlinear (Fig. 3D), with the primary clusters between 0–0.04 °C/year having an overall negative impact, and secondary clusters near 0.1 °C/year promoting lake drainage. The relationship between annual air temperature and the prediction of lake drainage demonstrates an almost monotonically increasing trend (Fig. 3E). As the annual air temperature rises, the risk of lakes experiencing drainage events also increases. This can be attributed to various mechanisms, including the thermal erosion of permafrost around lakes and the formation of drainage channels due to melting of ground ice^{27,31,47}.

Increasing elevation has an overall negative impact on the probability of lake drainage (Fig. 3F), with the primary cluster located in the range of 0–150 meters above sea level. We found that this lowland area covers approximately 29.6% of the northern permafrost zone, yet contributes to about 57.1% of the drained lakes, with drainage probabilities exceeding the average level (Fig. S7). We examined the impact of active layer depth on lake drainage (Fig. 3G) and observed a consistent rise in lake drainage probability as the active layer depth increases. Specifically, when the active layer depth in the area exceeds 0.6 meters, the probability of lake drainage tends to be higher than the average level, which might be associated with the formation of taliks^{48,49} at the lake bottom. The mechanisms and factors driving lake drainage are highly diverse, and with future Arctic warming and permafrost degradation, the frequency of lake drainage will likely increase^{20,25,43}.

Fig. 3. Model predictions of lake drainage and identification of key

environmental drivers. (A) Receiver operating characteristic (ROC) curve and (B) precision-recall (PR) curve plots for the accuracy assessment of the machine learning binary classification model used in predicting lake drainage events. The blue dashed lines represent a random guess. Recall: ability to find actual positives; Precision: accuracy of positive predictions; AUC: area under the curve; AP: average precision. A high AUC value indicates that the model has a strong ability to discriminate between positive (drained lakes) and negative (undrained lakes) samples, while a high AP value signifies high predictive quality for positive samples. (C) The relative importance of predictive variables in predicting the occurrence of lake drainage events, assessed using permutation importance and Shapley values⁸. The two methods are superimposed, with dark orange representing overlap. Both indicators identify trend slope of annual air temperature, annual air temperature and elevation as the most important predictors of lake drainage events. Kernel density estimations of Shapley value distributions for (D) trend slope of annual air temperature, (E) annual air temperature, (F) elevation, and (G) active layer depth. The density of sample distribution is normalized, with yellow indicating the most concentrated regions of data points. The magnitude of Shapley values reflects their impact on lake drainage predictions, with negative values representing lower likelihood. "

Lines 745–768: "

The training and testing samples for the binary classification model were derived from all drained lakes (~28,000) between 2001 and 2020 and a randomly selected 1% subset of undrained lakes (~58,000). The candidate explanatory variables for the model are detailed in table S3. For drained lakes, the climate parameters were extracted using values corresponding to their identified primary drainage year, while for undrained lakes, climate parameters were randomly sampled between the years 2001 and 2020, aiming to maximize coverage across different climate conditions. We used 70% of the samples for model training and the remaining 30% for independent testing of predictive performance. We utilized the Python-based Scikit-learn library⁸⁹ to fit CatBoost models and determined the optimal model hyperparameters through random search and ten-fold cross-validation. **The initial model includes all candidate explanatory variables, which might exhibit collinearity, meaning that introducing certain variables may lead to considerably changes in the importance estimates of other variables. To address this issue, during the**

iterative training process, we removed features that had negligible contributions to model performance and features highly correlated (Pearson correlation coefficient $r > 0.5$) with the most important variables for model performance. The important variables were identified through preliminary feature importance evaluations through the Shapley additive explanations model interpretability approach⁹⁰. We tested training separate models for very likely and unlikely thermokarst lakes for drainage prediction but did not observe an improvement in model accuracy. Therefore, we included thermokarst lake likelihoods as a categorical feature in the final model. The final model achieved an AUC score of 0.92 (Fig. 3A), indicating its effectiveness in distinguishing drained lakes from undrained lakes. The model's precision is 0.84, recall is 0.72, and average precision is 0.88 (Fig. 3B), demonstrating its ability to maintain high accuracy while effectively capturing drained lake samples. The remaining explanatory variables of the final model are shown in Fig. 3C, and the optimal model hyperparameters are listed in table S6."

Post-drainage vegetation dynamics in DLBs. This section reports the results of the authors' comparison of NDVI within and next to DLBs. A major confounding factor in this analysis is the presence of surface water; since the authors use a cutoff of 50% drainage, many of the DLBs still contain standing water. Surface water is known to influence NDVI values (Myers-Smith et al., 2020), and it would follow that any comparison of NDVI within incompletely drained DLBs to nearby land with presumably less standing water would also be affected by the presence of water. This issue must, at minimum, be discussed and, if possible, mitigated. Thank you for bringing up this crucial point. The potential influence of surface water on the NDVI comparison within and next to incompletely drained DLBs is indeed an important consideration. This issue was carefully analyzed and addressed in the revised manuscript.

Lines 686–704: "

Trace vegetation dynamics in DLBs

To trace the dynamics of vegetation growth following lake drainage, within the

preprocessing of Landsat image archives, we utilized the 90th percentile of NDVI values from annual image collections to generate a robust NDVI time series. This approach yields a value resembling the second highest NDVI value, which might perform better than the maximum and median NDVI values considering the potential influence of shadow masking and standing water in the lake basin.

For each drained lake, we pixel-wise acquired NDVI values within the lake basin area, masked out values below 0, and computed the median NDVI for the region. This step further mitigated the impact of standing water on NDVI assessment in DLBs. We also calculated the median NDVI of the surrounding vegetation to compare its greenness difference with DLBs. The surrounding vegetation was defined by a circular buffer of twice the lake diameter centered at the lake center, and masked to exclude the lake and other water bodies. Lastly, based on the identified drainage year for each lake, we transformed the time series of NDVI values from 2000 to 2020 into vegetation dynamics recorded by years after drainage.

Additionally, we applied the same approach to extract the TCG index from both DLBs and the surrounding areas for comparison (fig. S8). The results indicate that despite being based on different image bands and algorithms, the vegetation greening trends and patterns reflected by TCG are basically consistent with those of NDVI, underscoring the reliability of our method in capturing the vegetation dynamics in DLBs."

Lines 297–299: "Taking into account potential standing water in DLBs, we examined the Tasseled Cap Greenness (TCG) Index, which is less affected by surface water than NDVI⁵⁴, finding similar patterns (fig. S8)."

Lines 478–486: "It's important to note that while NDVI typically corresponds well with vegetation characteristics, surface water can impact NDVI values⁵³. Specifically, in partially drained basin areas, standing water can lead to underestimated NDVI values, despite vigorous sedge growth in developing aquatic environments. In this study, we used the 90th percentile of NDVI values from annual image collections to generate a

stable vegetation greenness assessment. We performed median extraction of NDVI after masking water bodies within DLBs and surrounding areas. A comparison with the results of the TCG index (fig. S8), which is less affected by surface water, indicates that our method effectively mitigates the impact of standing water on NDVI estimates within DLBs."

Fig. 5. Differences in vegetation greenness between DLBs and surrounding areas. NDVI measured in the tenth year after lake drainage for various classifications of (A) thermokarst lake likelihoods, (B) lake sizes, (C) drainage ratios, (D) permafrost extents, (E) regions, (F) floodplain status, (G) soil carbon contents, (H) soil nitrogen

contents, and (I) Yedoma region. Boxplots show the statistics – horizontal lines: median; dots: mean; boxes: interquartile range; whiskers: 1.5 times the interquartile range. (J) Time series of changes in relative greenness of very likely, likely and unlikely thermokarst DLBs, represented by NDVI differences compared to surrounding vegetation. Solid lines show median values, while shaded areas indicate upper and lower quartile ranges.

Fig. S8. Differences in vegetation greenness between DLBs and surrounding areas. TCG measured in the tenth year after lake drainage for various classifications of (A) thermokarst lake likelihoods, (B) lake sizes, (C) drainage ratios, (D) permafrost extents, (E) regions, (F) floodplain status, (G) soil carbon contents, (H) soil nitrogen contents, and (I) Yedoma region. Boxplots show the statistics – horizontal lines:

median; dots: mean; boxes: interquartile range; whiskers: 1.5 times the interquartile range. (J) Time series of changes in relative greenness of very likely, likely and unlikely thermokarst DLBs, represented by TCG differences compared to surrounding vegetation. Solid lines show median values, while shaded areas indicate upper and lower quartile ranges.

With respect to the results of this section, there is no attempt to put them into a larger context, even the context the authors lay out at the beginning of the paper (e.g., what is the role of DLBs in promoting Arctic greening?). One novel finding seems to be that, in general, non-thermokarst DLBs exhibit lower NDVI than the surrounding vegetation. But the reasons for and implications of this are not discussed.

In response, we have endeavored to provide a more comprehensive discussion that places our findings in the larger context.

Lines 474–478: "Based on a recent study⁵⁷, the average increase in NDVI in the high Arctic region was approximately 3.9% between 2000 and 2020. While in newly drained DLBs, the trend slope of NDVI was 1–2 orders of magnitude higher than the average level of arctic greening (Fig. 5J). The greening events observed within DLBs can be embedded within the overall greening trend of the Arctic, although they may not be the primary driver of this trend⁵³."

Lines 487–503: "DLBs, as hotspots of Arctic greening, have profound implications for vegetation composition of tundra ecosystems and permafrost carbon feedbacks^{4,43}. The clustered DLBs may yield habitats at various stages of succession, supporting ecosystems ranging from aquatic to dry tundra within basins over time, thus enhancing ecosystem diversity^{4,74}. Vegetation in DLB ecosystems can alter species composition, abundance, and distribution in Arctic vegetation, thereby driving interspecies competition within tundra vegetation^{17,57}. Environmental benefits from vegetation also include enhanced sediment stability, erosion prevention, improved water quality, habitat creation for wildlife, increased biodiversity, and support for indigenous pastoral activities⁵⁸. By tracing vegetation dynamics following lake drainage, we have found that vegetation greenness in DLBs exhibited high spatial

variability, and analyzed the differences in greenness between DLBs and surrounding vegetation under various environmental conditions. Our findings indicate that DLBs in thermokarst lakes, larger lakes, and lakes with higher drainage ratios exhibit higher vegetation greenness compared to DLBs in non-thermokarst lakes, smaller lakes, and lakes with lower drainage ratios (Fig. 5). We quantified the influence of environmental factors on predicting NDVI in DLBs and analyzed temperature and flooding as key environmental constraints. These findings can help with the management of DLBs as shifting habitat mosaics⁷⁵, and facilitate future conservation efforts for Arctic biodiversity. "

Regarding the analysis of NDVI differences between thermokarst and non-thermokarst DLBs, please refer to:

Lines 327–331: "The median areas of very likely, likely, and unlikely thermokarst drained lakes are 4.4 (25%–75% range: 2.0–12.7), 2.9 (1.6–7.1), and 2.1 (1.4–4.1) ha, respectively (Fig. 4C). Moreover, the drainage probability of large and medium-sized thermokarst lakes is markedly higher than that of non-thermokarst lakes (Fig. 4D), leading to proportions of small lakes within very likely, likely, and unlikely thermokarst drained lakes of 70%, 82%, and 90%, respectively (fig. S6)."

Lines 359–363: "Regions with higher soil carbon and nitrogen content support more vigorous vegetation growth⁵⁷, making vegetation in DLBs appear greener than the surrounding areas (Fig. 5G&H). This could partially explain why thermokarst DLBs and the Yedoma region exhibit greener vegetation compared to the surrounding areas (Fig. 5A&I)."

Environmental constraints of vegetation growth in DLBs. This section is a litany of results without interpretation or discussion and I am unsure how the results enhance our understanding of the ecosystem.

We have rewritten this section. And a new section was added later (Implications of lake drainage and vegetation growth in DLBs; lines 425–503)

Lines 373–424: "

Environmental influences on NDVI in DLBs

We further used machine learning models to analyze the drivers of NDVI change in DLBs, and evaluated the relative importance of explanatory variables (table S4) on NDVI predictions. The results (Fig. 6A&B) indicate that our trained model is capable of capturing a substantial portion of NDVI variations in DLBs ($R^2=0.83$). The assessment of feature importance reveals that ecological zoning and the years since lake drainage are the most important influencing factors for NDVI predictions (Fig. 6C). Different ecoregions indeed host distinct local species pools⁵⁸, leading to variations in vegetation greenness levels and successional trajectories. High spatial variability in vegetation dynamics of DLBs has been observed across ecoregions, with NDVI values ranging from around 0.5 to 1.2 times the surrounding average (table S2). The specific years since lake drainage emerged as a key driver of NDVI predictions, as they potentially reflect the progression of ecosystem succession. This underscores the importance of accurately identifying the specific years of lake drainage events for analyzing the vegetation dynamics within DLBs.

Among the climatic variables, summer air temperature and annual air temperature trend stand out as the most important influencing factors for NDVI predictions (Fig. 6C). Their increase has an overall positive impact on NDVI within DLBs, primarily due to the extension of the growing season and the enhancement of photosynthetic efficiency^{58,59}. Temperature serves as a primary constraint for the greenness of Arctic ecosystems, with vegetation greening most frequently occurring in regions experiencing elevated summer air temperatures and annual soil temperatures⁵². The relationship between summer air temperature and NDVI predictions displays an almost monotonically increasing trend. This suggests that warmer locations tend to show enhanced plant growth, rather than implying that an increase in temperature at any specific location universally benefits plant growth. When summer air temperatures fall below 10°C, vegetation greenness is suppressed, evident by negative Sharpe values (Fig. 6D). The impact of the annual air temperature trend is more intricate, with two major clusters at 0.01 and 0.06 °C/year exerting negative effects on NDVI, while two minor clusters at 0.12 and 0.18 °C/year promote

greening of vegetation (Fig. 6E).

Floodplain and thermokarst lake stand out as the most important categorical features for predicting NDVI (Fig. 6C). Floodplain areas and non-thermokarst lakes have an overall negative impact on predicting NDVI in DLBs, while non-floodplain areas and thermokarst lakes have a positive influence (Fig. 6F&G). The machine learning model uncovers the suppressive effect of flood on NDVI in DLBs, which is not easily observable through conventional statistical analysis (Fig. 5F), demonstrating the model's insights. These findings reveal how vegetation dynamics in DLBs are jointly influenced by species pools, vegetation succession patterns, regional climate and hydrological conditions, as well as landscape attributes.

Fig. 6. Model predictions of NDVI in DLBs and identification of key

environmental factors. (A) Scatter plot comparing predicted and actual values and (B) residual plot of the machine learning regression model for predicting post-drainage NDVI in DLBs. Scatter point density is color-coded, with a red dashed line representing the ideal scenario. RMSE: root mean square error; MAE: mean absolute error; MB: mean bias. (C) The relative importance of predictive variables in explaining NDVI in DLBs, assessed using permutation importance and Shapley values. These two methods are superimposed; deep orange represents overlap. Kernel density estimations of Shapley value distributions for (D) summer air temperature and (E) trend slope of annual air temperature. The density of sample distribution is normalized, with yellow indicating the most concentrated regions of data points. The magnitude of Shapley values reflects their impact on NDVI predictions, with negative values representing lower NDVI. Violin plots of Shapley value distributions for (F) floodplain status and (G) thermokarst lake likelihoods, containing internal boxplots. Importantly, Shapley values can aid in model interpretation and variable selection, but they do not directly reflect the model's accuracy. The range of Shapley values differs between classification and regression models, making them non-comparable. The emphasis should be placed on analyzing the importance of variables within each model."

Other comments:

The authors describe their study region as the 'entire circum-Arctic permafrost region' and refer to it as the Arctic throughout the text. However, much of the study region is in fact not part of the Arctic such as subarctic/boreal regions and the Tibetan plateau. The term 'Arctic' is inappropriate for the study region. An alternative could be: 'northern permafrost zone,' but there are likely others.

We appreciate the reviewer's assistance in ensuring the accuracy of our manuscript. Thank you for bringing this issue to our attention, and we apologize for any confusion. We have carefully reviewed the text and made the adjustments to accurately describe the geographical extent of our study. As suggested, we have referred to our study region consistently as the 'northern permafrost zone'.

The authors divide DLBs into two types: thermokarst and non-thermokarst. I cannot find in the methods how these two lake types were differentiated, which is problematic. But assuming this was done based on the map from Olefeldt et al., 2016, the authors are defining

lakes by whether or not they fall within a region of high thermokarst lake coverage, NOT whether each individual lake is a thermokarst lake or not. Non-thermokarst lakes exist within regions of high thermokarst lake coverage. See for example Lara and Chipman (2021). Presumably, thermokarst lakes can also exist within regions of low thermokarst lake coverage. Therefore, what the authors are referring to as a 'thermokarst DLB' may actually be of non-thermokarst origin (and same for non-thermokarst DLB being of thermokarst origin). This point needs to be highlighted in the text and different language needs to be used to describe lake types.

We appreciate the reviewer's comment and acknowledge the need for clarification. To address this concern, we have provided a more detailed explanation of thermokarst lake classification in the Methods section. Additionally, we have revised the language throughout the manuscript to ensure clarity and accuracy, avoiding any potential confusion or misdescription.

Lines 559–563: "Olefeldt et al. (2016)³³ provided a thermokarst lake distribution map based on expert judgment, categorized into five levels: very high (60–100%), high (30–60%), medium (10–30%), low (1–10%) and none (0–1%). For analytical convenience, we reclassified thermokarst lake likelihoods into three categories: very likely (60–100%), likely (1–60%), and unlikely (0–1%). This classification is map-based and not an assessment of the origin for individual drained lakes."

Lines 113–116: "The Yedoma region, discontinuous permafrost zone, and lakes classified as very likely to have a thermokarst origin (hereafter referred to as 'very likely thermokarst lakes') are densely drained, with densities approximately 2, 2.5, and 3 times higher than average, respectively (Fig. 1H)."

Lines 123–126: "Drained lakes are less dense in areas that are ice-poor and unlikely to form thermokarst lakes (hereafter referred to as 'unlikely thermokarst lakes'), where lake abundance is relatively low and limited ice wedge melting does not readily create drainage channels⁴."

Notably, the authors do not reference or discuss other studies related to DLB mapping in the

Arctic. See for example: Lara and Chipman, 2021; Lara et al., 2021; Bergstedt et al., 2021. Other studies that track DLB frequency over time (also not cited): Marsh et al., 2009; Lantz and Turner, 2015; Lara et al., 2021.

Thank you for bringing these important references to our attention. These references will undoubtedly strengthen our discussion and provide a more comprehensive perspective on the topic. We have thoroughly reviewed the suggested references and included these studies in our manuscript.

Line-by-line comments:

Figure 1D: The color choice for the lake size is difficult to differentiate.

Appreciate the feedback. We have made color adjustments to enhance the contrast and improve visibility in Fig 1D.

Figure 4: I am not sure what NDVI means here. According to the methods you calculated NDVI for multiple years after drainage, but there is no mention of the timeframe for NDVI in this figure.

We have clarified this in Lines 769–775: "The training process of the regression model for predicting NDVI in DLBs is similar to the binary classification model described earlier, with the distinction that the predicted variable is a continuous value between 0 and 1 (NDVI), rather than a binary value (0 or 1). Additionally, the loss function used is Root Mean Square Error (RMSE) instead of Logarithmic Loss. The training and testing samples of the model are based on the annual NDVI values of all drained lakes and the climate data of the corresponding years, recorded with reference to the year of lake drainage. In other words, the model predicts the NDVI values in DLBs for the 0–15 years after lake drainage."

Line Comments

5 delete 'the'

Done as suggested.

21 change 'its fragile ecosystem' to 'ecosystems.' Additionally, what is a challenge to an ecosystem? Consider re-wording.

Appreciate the feedback, this has been revised to "ecosystem stability".

22 The Arctic is a region, not an ecosystem. You could simply say 'Arctic ecosystems'

Done as suggested.

25 what do you mean by 'net gain?' Earlier in the sentence you say there is a net loss of lake area. What is gaining?

We have modified this sentence: " Satellite observations^{8–10} indicate that Arctic lake-rich areas have experienced **a decline in lake area** over the last 20 years, **contrary to** model projections^{11,12} that suggested **an increase** due to widespread permafrost thaw."

29 Delete critical. (How are methane emissions critical?)

Done as suggested.

31 change 'hydrological' to 'water.' Also, this is a long sentence. Consider breaking up into two.

Thanks for the suggestion, we have made the suggested modification.

35 The point is not that DLBs have long intrigued researchers, but rather DLBs are important for ecosystem permafrost characteristics, vegetation dynamics, and carbon storage.

Appreciate the feedback, the suggested change has been incorporated. Lines 36–39: "Monitoring lake drainage and subsequent DLB evolution in Arctic and Boreal regions through detailed observations can provide valuable insights for studies of permafrost ecosystem characteristics, vegetation dynamics and carbon storage^{21–23}."

42 There are studies that have looked at this; it has not been ignored. Both Webb et al., (2022) and Nitze et al., (2018) included ground-ice poor regions. Other examples include: Smol and Douglas (2007), Carroll et al., (2011), Finger Higgs et al., (2019), Law et al. (2018) Carroll and Loboda (2018) **AND**

45 This paper looked at drainage event timing and associated landscape characteristics and environmental drivers across NW Alaska: Lara et al., (2021)

Appreciate the feedback, we have revised the statement. In addition, we have carefully checked the literature and cited it where appropriate.

Lines 42–50: "Existing remote sensing studies on lake drainage mostly focus on a

limited area within the northern permafrost zone^{24–30}, which hinders their capacity to provide a holistic understanding of the distribution patterns and underlying drivers of drained lakes due to the presence of spatial heterogeneity. Webb et al.⁸ conducted an analysis of surface water trends at a 12-km pixel resolution and found that the drying trend in lake area across Arctic lake-rich regions was correlated with increasing annual air temperatures and autumn rainfall. However, the spatio-temporal distribution of specific lake drainage events that contribute to the observed drying trend remains unclear, which is crucial for understanding the transition dynamics from lakes to DLBs and accurately identifying the climatic and landscape attributes associated with lake drainage."

52 Include a ref for Arctic greening.

Noted and added the necessary citations.

53 Entire circum is redundant. Choose one.

Done as suggested.

54 Change 'four decades or so' to 'three and a half decades'; change 'by' to 'using'

We have modified the text: "In this study, we conducted a comprehensive analysis of lake drainage events across the northern permafrost zone over a span of three and a half decades (1984–2020). By leveraging existing surface water products^{36,37} and satellite remote sensing, we accurately detected these events, allowing us to delineate their spatial distribution and determine the corresponding drainage years."

69 Over what time period? If a lake lost 50% of its area over the 35 years, would that be a DLB? Or does the drainage need to occur on a shorter timescale?

Thanks for the feedback. The time frame for a lake drainage event can vary, and there is no specific duration that universally defines such an event. The term "drainage event" typically implies a relatively rapid and substantial decrease in the lake's water volume or area. However, the exact rate and timeframe can depend on various factors, including the size of the lake, local climate conditions, and the underlying processes causing the drainage.

Lines 546–552: "We identified lake objects as having undergone a drainage event only if their proportion of surface area loss exceeded 50%, which is a more stringent

criterion than that used in previous studies, given our focus on investigating drastic alterations in Arctic lake systems. Here, we have not imposed constraints on the time span of lake drainage events, as there is no universally accepted quantitative definition for the specific duration of lake drainage events in existing literature²⁵. In some cases, lakes can gradually drain over multiple years (fig. S9)."

73 Delete 'spatially.' Unless it is important to include, in which case you will need to reword the sentence.

Done as suggested.

74 Are you saying that the thermokarst lakes are located in coastal plains and river deltas, or are you saying that the majority of DLBs occurred here?

We have modified the text: "The distribution of drained lakes exhibits spatial clustering, predominantly concentrated in coastal lowlands and river delta areas (figs. S1–S3). Among them, about half of the drained lakes are located within thermokarst lake landscapes (Fig.1D), which cover approximately 20% of the permafrost zone³³."

78 Did you show that here? If so, refer to table/figure. If you don't show that here, you will need a reference. AND

79 The first part of this paragraph is on thermokarst lakes. The second part is on lake size. Are these lake sizes for thermokarst lakes or for all lake types? Please clarify and if it is all lakes combined, do not include in the same paragraph as the discussion of thermokarst lakes. Thanks to the suggestion, we have moved the content about the thermokarst lakes to the Introduction. Lines 52–59: "Lakes across the northern permafrost region can be divided into thermokarst and non-thermokarst lakes, depending on their origin^{4,32}. Thermokarst lakes develop in ice-rich permafrost zones through the thawing of sediments, melting of ice wedges, and localized ground subsidence^{15,33,34}, while non-thermokarst lakes form by accumulating water in pre-existing topographic depressions without significant thaw-induced subsidence. Compared to non-thermokarst lakes surrounded by ice-poor permafrost, thermokarst lakes usually undergo faster lateral erosion and bottom talik (unfrozen ground) development^{31,35}, making them more likely to drain⁴."

86 But perhaps all the lakes in this region are small. How does this compare to the size distribution of existing lakes? AND

87 Again, how does this compare to the size distribution of all lakes in the region? You may be showing that the lakes in this region are larger than average, and so the drained lakes are larger than average.

Thanks to the suggestion, we have calculated the lake drainage probabilities for all ecoregions (table S2) and updated the main text.

Lines 99–106: "For instance, along the southern permafrost margin at the Russia-Mongolia border, this area is densely populated with thousands of drained lakes (Fig. 1E), even though it is not situated within the lake-rich zone. Here, the lake drainage probability reaches nearly 8%, which is more than ten times higher than the average level across the entire permafrost zone. In contrast, in the Northeast Siberian coastal tundra ecoregion (Fig. 1F), although the overall lake drainage probability is lower at one-third of the regional average, large lakes have a drainage probability twice the average, leading to more than half of the drained lakes being medium to large in size."

88 How do you know they are previously unnoticed? The people who live there may have noticed.

Sorry about the wording, we changed it as "underreported".

91 Are you referring to the events on St. Lawrence Island, or across the study region?

This part of the text is an analysis of St. Lawrence Island, which in the new manuscript we have moved to the section "Implications of lake drainage and vegetation growth in DLBs".

101 Should be 'the continuous permafrost zone' rather than 'continuous permafrost zones'
Done as suggested.

102 With climate change, some continuous permafrost at the southern extent will become discontinuous permafrost. But the entire continuous permafrost zone is not vulnerable to becoming discontinuous. This sentence is misleading and the references do not support the claim.

Thanks for the correction, we have modified in Lines 276–277: "However, the susceptibility is rising in certain southern areas where permafrost is becoming more

discontinuous due to climate warming^{9,20}."

104 This is the first time you mention ice wedge degradation and drainage channels. I suggest you explain this mechanism a bit more.

We have added in Lines 116–123: " Drained lakes in the northern permafrost zone can be divided into two primary categories: lateral drainage and internal drainage³⁴. In the continuous permafrost zone, lake drainage often results from lateral expansion of lakes into low-lying areas, driven by mechanisms such as thawing of ice barriers, headward stream erosion, coastal erosion, and bank overtopping due to rapid snowmelt, extreme precipitation, and flooding^{30,42,43}. While in the discontinuous permafrost zone, in addition to lateral drainage, internal drainage related to ice wedge degradation frequently occurs, where taliks or thawed zones beneath lakes penetrate the permafrost, allowing for drainage subterraneously^{24,34,44}."

107 I am not sure why this is implied. More development of this idea is needed, including references.

We have revised in Lines 251–257: "Large drained lakes are disproportionately concentrated in these areas, with 6–8 times the average density (Fig. 1H) and 3–4 times the average drainage possibility (Fig. 4B&D), suggesting well-developed erosional drainage systems may have formed. The prevalent landscape in these areas is a mosaic of lakes and streams, interconnected through an underground network of ice^{15,42}. When expanding lakes encounter the ice network, thermal erosion along the network results in the melting of ice wedges, creating drainage channels through which the lake water drains^{7,34}."

125 How can you know that there were undetected events if you cannot detect them? A simple rewording might suffice.

Thanks. We have modified the sentence: "However, the fractured nature of Landsat observations across space and time³⁶ prior to 2000 limited the detection of lake drainage events that occurred during 1984–2000."

130 What is the p-value for this statement? Is there a figure or table where this is showed?

We have added the p-values as requested.

131 I am having trouble interpreting the slope here. This is an increase of 19 drainage events

per year? For the whole region?

We have modified in Lines 167–174: "From 2001 to 2020, lake drainage events exhibited significant upward trends ($p < 0.001$) in discontinuous permafrost zones and very likely thermokarst lakes, with slopes of 21 and 14, respectively. This indicates that over the 20-year period, the mean annual number of drained lakes increased by 420 in discontinuous permafrost zones and 280 for very likely thermokarst lakes. Over time, the contribution of the discontinuous permafrost zone to the annual count of drained lakes gradually surpasses that of the continuous permafrost zone (Fig. 2F), suggesting that the impacts of climate change and permafrost degradation on the stability of lakes in these sensitive regions appear to be intensifying^{24,34,42}."

133 I am not sure what the percentages are referring to. The numerator is drained thermokarst lakes but what is the denominator?

We have deleted this expression.

148 remove 'the'

Done as suggested.

152 This sentence needs a citation

We have added citations as requested.

157 These references are for thermokarst lakes, yet the sentence is about non-thermokarst lakes

We have updated citations as requested.

159 This statement needs a citation

We have added citations as requested.

183 This statement needs a citation - your data is just NDVI. What field studies show that this is actually what is happening on the ground? **AND**

185 This statement needs a citation. What field studies show that this is actually what is happening on the ground?

We have added citations as requested.

186 need p-value

We have added the p-values as requested.

196 Since the Chen et al. numbers are in %, can you also put your numbers in % so the reader

can compare?

Thanks for the suggestion. We have modified in Lines 305–311: "Chen et al.¹⁷ analyzed vegetation dynamics in thermokarst DLBs in northern Alaska and found that tundra vegetation growing on wet and nutrient-rich lake sediments was more luxuriant (with 0.15 or 25% higher NDVI) than in surrounding areas. Here we find that across the northern permafrost region, the NDVI of very likely thermokarst DLBs is higher by 0.06 or 10% compared to the surrounding areas, while the NDVI of unlikely thermokarst DLBs is lower by 0.09 or 15% than the surrounding areas (Fig. 5A)."

197 need p-value

We have added the p-values as requested.

205 This statement needs a citation

We have added citations as requested.

207 This isn't just because there is more disturbed area, so more opportunity for veg growth?

207 Be careful here and elsewhere; you measured NDVI, not vegetation growth. NDVI is only a proxy and you do not have direct measurements of vegetation growth.

Thanks for the reminder. We have modified in Lines 339–341: "A higher drainage ratio implies a greater amount of exposed land for plant growth, and in these areas, earlier summer thaw and elevated concentrations of dissolved soil carbon and nitrogen create favorable conditions for promoting vegetation growth⁵⁵."

209 Need citation for "...favorable conditions for vegetation growth"

We have added citations as requested.

210 Explain this more. Shouldn't thin and rocky soils be characteristic of both the DLB and the surrounding areas? Why would this cause DLBs to have lower NDVI relative to the surrounding?

We have added explanations as requested in Lines 348–354: "The NDVI of DLBs was noticeably lower in Canada, which may be attributed to the thin and rocky soils in the Canadian Shield region⁵⁶. These soil conditions could pose drainage challenges, which may lead to increased waterlogging and periodic inundation of vegetation, ultimately resulting in a lower NDVI within DLBs compared to the surrounding areas. Additionally, the proportion of small drained lakes is higher in

Canada compared to other regions, potentially contributing to less lush vegetation in Canadian DLBs (table S2)."

213 This statement needs a citation

We have added the citations as requested.

215-16 I am having a hard time understanding what this sentence means. What does it mean to "facilitate nutrient concentration"? What does 'them' refer to?

Sorry for the lack of clarity. We have modified in Lines 355–359: "We examined the NDVI of DLBs in flood-prone and non-flood-prone areas and did not find significant differences (Fig. 5F). This may be due to the micro-topography of DLBs characterized by subtle variations in land surface elevation and slope. This micro-topography effectively directs runoff, facilitating nutrient accumulation to support vegetation growth in DLBs, while also rendering the vegetation more susceptible to waterlogging^{17,22}."

245 How do you know this? At the very least, this statement needs a citation. But it should probably also be re-worded as speculative rather than declarative.

We have deleted this expression.

253 This makes it sound like an increase in temp at given location is beneficial for plant growth. But what I think the figure is showing is that, in general, warmer locations tend to have enhanced plant growth. There is a difference between the two.

Thanks for bringing this to our attention. We have made the suggested changes in Lines 392–395: "The relationship between summer air temperature and NDVI predictions displays an almost monotonically increasing trend. This suggests that warmer locations tend to show enhanced plant growth, rather than implying that an increase in temperature at any specific location universally benefits plant growth."

268 Remote sensing studies do not 'overcome' sampling bias in field studies. The two approaches measure different things on different scales and give us different types of information.

Thanks for the correction, we have modified in Lines 426–428: "In summary, our study leveraged remote sensing data to capture spatial and temporal distribution patterns of lake drainage events in the extensive northern permafrost regions, extending our

comprehension of the post-drainage vegetation dynamics within DLBs."

270 entire circumpolar is redundant. Pick one.

Done as suggested.

291 emissions of what?

Appreciate the feedback, "emission fluxes" has been revised to "carbon fluxes".

316 delete 'directly'

Done as suggested.

329 Earth Engine should be capitalized and cited

Done as suggested.

330 'firstly' should be 'first'

Done as suggested.

340 Which other studies are you referring to? What is the justification for 50%?

We have modified Lines 546–549: "We identified lake objects as having undergone a drainage event only if their proportion of surface area loss exceeded 50%, which is a more stringent criterion than that used in previous studies^{31,34,42,43}, given our focus on investigating drastic alterations in Arctic lake systems."

375 Citation needed

We have added the citations as requested.

388 Other drained lake datasets DO exist. See for example:

<https://arcticdata.io/catalog/view/doi:10.18739/A2BV79W8S>

<https://tc.copernicus.org/articles/14/4279/2020/#section7>

Thanks for the information. We have added in Lines 658–667: "The false negative rate of lake drainage events is challenging to accurately assess due to the relatively small proportion of drained lakes (less than 1% of all lakes) and the absence of a comprehensive reference dataset of drained lakes with the same temporal coverage. Based on two existing regional drained lake datasets^{25,43}, we evaluated drained lake detection results for the Kotzebue Sound Lowlands in northwestern Alaska. We selected lakes with drainage proportions exceeding 50% and performed a visual assessment. The assessment revealed that our detection accuracy was 86.5%, while the datasets of Lara et al. (2021)²⁵ and Nitze et al. (2020)⁴³ had accuracies of 71.4%

and 58.3% respectively. However, it's important to note that our study has a distinct time range compared to these datasets and uses a different water extent map to calculate lake drainage proportions."

400 'densely' should be 'dense'

Done as suggested.

417 The Brown et al. ground ice map only has three categories: high, medium, and low ground ice. Please explain how you derive very high.

In the previous version of the manuscript, we integrated Yedoma data into ground ice data as "very high" ground ice content. Recognizing the need for a more rigorous classification, we have redefined this in the revised manuscript.

Lines 566–570: "The circumpolar ground ice product⁴¹ classifies ground ice content into three classes based on volume percentage: high (>20%), medium (10–20%), and low (0–10%). Yedoma is an organic-rich (~2% carbon by mass), ice-rich (>50% ice content by volume) permafrost formed during the Pleistocene, mainly distributed in eastern Siberia, Alaska and the Yukon, and the landscape is characterized by glacial plains and hills with sparse vegetation cover³⁹."

458 change 'drainage lake ecosystems' to 'drained lake object'

Done as suggested.

473 citation needed

We have added the citations as requested.

References cited

Bergstedt, H., Jones, B. M., Hinkel, K., Farquharson, L., Gaglioti, B. V., Parsekian, A. D., Kanevskiy, M., Ohara, N., Breen, A. L., Rangel, R. C., Grosse, G., and Nitze, I.: Remote Sensing-Based Statistical Approach for Defining Drained Lake Basins in a Continuous Permafrost Region, North Slope of Alaska, *Remote Sensing*, 13, 2539, <https://doi.org/10.3390/rs13132539>, 2021.

Cao, B., Gruber, S., Zheng, D., and Li, X.: The ERA5-Land soil temperature bias in permafrost regions, *The Cryosphere*, 14, 2581–2595, <https://doi.org/10.5194/tc-14-2581-2020>, 2020.

Carroll, M. L. and Loboda, T. V.: The sign, magnitude and potential drivers of change in surface water extent in Canadian tundra, *Environmental Research Letters*, 13, 045009, <https://doi.org/10.1088/1748-9326/aab794>, 2018.

Carroll, M. L., Townshend, J. R. G., Dimiceli, C. M., Loboda, T., and Sohlberg, R. A.: Shrinking lakes of the Arctic: Spatial relationships and trajectory of change, *Geophysical Research Letters*, 38, 1–

5, <https://doi.org/10.1029/2011GL049427>, 2011.

Finger Higgins, R. A., Chipman, J. W., Lutz, D. A., Culler, L. E., Virginia, R. A., and Ogden, L. A.: Changing Lake Dynamics Indicate a Drier Arctic in Western Greenland, *Journal of Geophysical Research: Biogeosciences*, 124, 870–883, <https://doi.org/10.1029/2018JG004879>, 2019.

Lantz, T. C. and Turner, K. W.: Changes in lake area in response to thermokarst processes and climate in Old Crow Flats, Yukon, *Journal of Geophysical Research: Biogeosciences*, 120, 513–524, <https://doi.org/10.1002/2014JG002744>, 2015.

Lara, M. J. and Chipman, M. L.: Periglacial Lake Origin Influences the Likelihood of Lake Drainage in Northern Alaska, *Remote Sensing*, 13, 852, <https://doi.org/10.3390/rs13050852>, 2021.

Lara, M. J., Chen, Y., and Jones, B. M.: Recent warming reverses forty-year decline in catastrophic lake drainage and hastens gradual lake drainage across northern Alaska, *Environmental Research Letters*, 16, <https://doi.org/10.1088/1748-9326/ac3602>, 2021.

Law, A. C., Nobajas, A., and Sangonzalo, R.: Heterogeneous changes in the surface area of lakes in the Kangerlussuaq area of southwestern Greenland between 1995 and 2017, *Arctic, Antarctic, and Alpine Research*, 50, e1487744, <https://doi.org/10.1080/15230430.2018.1487744>, 2018.

Marsh, P., Russell, M., Pohl, S., Haywood, H., and Onclin, C.: Changes in thaw lake drainage in the Western Canadian Arctic from 1950 to 2000, *Hydrological Processes*, 23, 145–158, <https://doi.org/10.1002/hyp.7179>, 2009.

Myers-Smith, I. H., Kerby, J. T., Phoenix, G. K., Bjerke, J. W., Epstein, H. E., Assmann, J. J., John, C., Andreu-Hayles, L., Angers-Blondin, S., Beck, P. S. A., Berner, L. T., Bhatt, U. S., Bjorkman, A. D., Blok, D., Bryn, A., Christiansen, C. T., Cornelissen, J. H. C., Cunliffe, A. M., Elmendorf, S. C., Forbes, B. C., Goetz, S. J., Hollister, R. D., de Jong, R., Loranty, M. M., Macias-Fauria, M., Maseyk, K., Normand, S., Olofsson, J., Parker, T. C., Parmentier, F. J. W., Post, E., Schaepman-Strub, G., Stordal, F., Sullivan, P. F., Thomas, H. J. D., Tømmervik, H., Treharne, R., Tweedie, C. E., Walker, D. A., Wilmsking, M., and Wipf, S.: Complexity revealed in the greening of the Arctic, *Nature Climate Change*, 10, 106–117, <https://doi.org/10.1038/s41558-019-0688-1>, 2020.

Nitze, I., Grosse, G., Jones, B. M., Romanovsky, V. E., and Boike, J.: Remote sensing quantifies widespread abundance of permafrost region disturbances across the Arctic and Subarctic, *Nature Communications*, 9, 1–11, <https://doi.org/10.1038/s41467-018-07663-3>, 2018.

Smol, J. P. and Douglas, M. S. V.: Crossing the final ecological threshold in high Arctic ponds, *Proceedings of the National Academy of Sciences of the United States of America*, 104, 12395–12397, <https://doi.org/10.1073/pnas.0702777104>, 2007.

Webb, E. E., Liljedahl, A. K., Cordeiro, J. A., Loranty, M. M., Witharana, C., and Lichstein, J. W.: Permafrost thaw drives surface water decline across lake-rich regions of the Arctic, *Nature Climate Change*, <https://doi.org/10.1038/s41558-022-01455-w>, 2022.

We would like to extend our sincere gratitude to the reviewer for the valuable contribution in suggesting relevant references. In the revised manuscript, we have followed the advice and enhanced the citation of references accordingly.

III. Response to comments and suggestions of Reviewer #3

This study looks at Landsat images from 1984-2020 and identifies 35000 lake drainage events in Arctic regions, including about half that are previously unreported in non-thermokarst lakes. The analysis identifies summer soil temperature and net annual precipitation as the primary drivers of drainage in thermokarst and non-thermokarst lakes respectively and demonstrates growth in vegetation following the drainage event.

The findings presented are interesting and builds on prior studies (cited in the paper) that demonstrate decreased lake area due to permafrost thaw in the Arctic, and detection of lake drainage events in permafrost regions. That being said, most of the findings and some of the methodology essentially seem to be an extension of the work of Chen et al. 2022 (<https://doi.org/10.1016/j.scitotenv.2021.150828>) & Chen et al. 2021 (10.1111/gcb.15853), but applied to a global dataset. Even the figures seem remarkably similar (e.g. Figure 3 in Chen et al. 2021 and Fig 3A/3B in this paper). The most noteworthy, previously unreported findings are the distinctions between thermokarst and non-thermokarst lakes, but the implications of these differences can be better explained in the introduction and discussion sections.

We sincerely appreciate your insightful observations and comments on our manuscript. Your recognition of the study's key findings and its extension of prior research is truly appreciated. We are honored to have been able to expand our previous work and deepen this research.

Thank you for your constructive feedback and for taking the time to review our manuscript. Your input has undoubtedly contributed to the refinement of our work. In our revision, we have fully taken your suggestion into consideration and enhanced the clarity to provide a more comprehensive context for the distinct findings.

I have some questions and concerns about the methodology. Overall some sections are not described in enough detail to evaluate and reproduce the methodology:

1. One of the primary findings is the relationship between climactic variables and the drainage lake events. This analysis seems to be problematic for many reasons. First the

choice of the ERA-5 dataset seems strange for this purpose given its spatial resolution of ~9km, and that most of the lakes (85%) are very small 1-10 ha & 98% are <100 ha (1 sq km). I understand that there aren't meteorological products available at high resolution globally, but this enormous difference in resolution needs to be handled, by either upscaling the lakes to a comparable area (see for e.g. Webb et al. 2023), downscaling the ERA-5 data, or at least comparing with a finer-scale product for regions where they are available (e.g. Daymet at 1 km resolution). Secondly there are significant collinearities between the variables chosen, so it is not surprising that net annual precipitation (P-ET) and annual precipitation or summer soil temperature/annual soil temperature/air temperature have similar correlations. Redundancies in climate variables should be handled when interpreting the correlation analysis. I would also have liked to have seen a stronger approach (beyond correlations) to determining the drivers for the extent of drained lakes, and it seems like a boosted tree ML approach might also work for this purpose.

We sincerely appreciate your pointing out these important issues. In the Methods section, we have provided justifications for utilizing the ERA5-Land dataset.

Lines 706–727: "

ERA5-Land reanalysis data

Due to the lack of fully covered high spatial resolution meteorological products in the northern permafrost regions, we used ERA5-Land monthly reanalysis dataset⁸⁵ for the period 2000-2020 to analyze the climate factors affecting lake drainage and vegetation growth. ERA5-Land is a land-enhanced product from the fifth generation of European ReAnalysis (ERA5), with a spatial resolution of 0.1° (or 9 km). It is considered a state-of-the-art global reanalysis dataset for land applications that efficiently reconstructs surface states and process parameters through advanced data assimilation techniques⁸⁵. A recent study noted that ERA5-Land showed a high degree of global consistency with MODIS satellite products in terms of surface temperature⁸⁶.

Despite the relatively coarse spatial resolution of ERA5-Land, the analysis based on a 0.1° x 0.1° grid reveals that 35,337 drained lakes are distributed in 20,132 grid cells, most of which contained 1–2 drained lakes (fig. S10A). Over 99% of the grid

cells exhibit fewer than 10 drained lakes (fig. S10B). The highest count of drained lakes in a grid cell is 33, occurring at St. Lawrence Island (Fig. 1G). Therefore, for lake drainage events occurring between 2000 and 2020, ERA5-Land can provide sufficiently differentiated climatic information for lake drainage prediction and post-drainage vegetation dynamics analysis.

Furthermore, we evaluated the temperature and precipitation of ERA5-Land dataset using the 1 km resolution Daymet dataset⁸⁷ covering North America (fig. S11). This analysis compares climate data of Daymet and ERA5-Land for the year in which the lakes were drained using 16,104 drained lakes detected in North America as sample points. The results indicate strong overall consistency and low variability between Daymet and ERA5-Land reanalysis data in simulating temperature and precipitation (fig. S11)."

Fig. S10. Grid-based statistics of drained lakes. (A) Distribution map of drained lakes based on a 0.1°x0.1° grid. (B) Pie chart statistics of grid counts with drained lakes. A total of 35,337 drained lakes are distributed across 20,132 grid cells, with the majority of cells containing 1–2 drained lakes. More than 99% of grid cells have fewer than 10 drained lakes.

Fig. S11. Bland-Altman plots for assessing the consistency of ERA5-Land and Daymet reanalysis data. Evaluation of (A) mean annual air temperature, (B) mean summer air temperature, (C) annual precipitation, and (D) summer precipitation. The analysis is based on a total of 16,104 drained lakes detected in North America, using climate data for the year of drainage for comparison. As Daymet dataset does not provide mean air temperature, a simple estimation was made using the average of maximum and minimum air temperature. Bland-Altman plots visually display the average difference and variability between the two datasets. The x-axis represents the average values of the two datasets, and the y-axis shows the differences. The solid red line represents the mean difference, while the dashed red line indicates the 95% limits of agreement. Scatter density is shown in different colours, with yellow representing areas of high concentration of sample points. Overall, ERA5-Land and Daymet reanalysis data exhibit strong consistency in simulating temperature and precipitation, with low variability.

We have employed the machine learning classification model to predict lake drainage and analyze the environmental driving factors.

Lines 181–223: "

Environmental drivers of Lake drainage events

To reveal the key environmental drivers behind lake drainage events from a variety

of candidate explanatory variables such as climate, topography, and permafrost characteristics (table S3), we developed a binary classification model to predict lake drainage. We conducted a comprehensive evaluation of model performance using multiple metrics, which showed the model demonstrates strong classification capabilities (Fig. 3A&B), effectively utilizing explanatory variables to predict whether an individual lake will experience drainage. Further feature importance assessment indicates that trend slope of annual air temperature, annual air temperature and elevation are the most important drivers of lake drainage events (Fig. 3C). The effect of annual air temperature trend on lake drainage events is nonlinear (Fig. 3D), with the primary clusters between 0–0.04 °C/year having an overall negative impact, and secondary clusters near 0.1 °C/year promoting lake drainage. The relationship between annual air temperature and the prediction of lake drainage demonstrates an almost monotonically increasing trend (Fig. 3E). As the annual air temperature rises, the risk of lakes experiencing drainage events also increases. This can be attributed to various mechanisms, including the thermal erosion of permafrost around lakes and the formation of drainage channels due to melting of ground ice^{27,31,47}.

Increasing elevation has an overall negative impact on the probability of lake drainage (Fig. 3F), with the primary cluster located in the range of 0–150 meters above sea level. We found that this lowland area covers approximately 29.6% of the northern permafrost zone, yet contributes to about 57.1% of the drained lakes, with drainage probabilities exceeding the average level (Fig. S7). We examined the impact of active layer depth on lake drainage (Fig. 3G) and observed a consistent rise in lake drainage probability as the active layer depth increases. Specifically, when the active layer depth in the area exceeds 0.6 meters, the probability of lake drainage tends to be higher than the average level, which might be associated with the formation of taliks^{48,49} at the lake bottom. The mechanisms and factors driving lake drainage are highly diverse, and with future Arctic warming and permafrost degradation, the frequency of lake drainage will likely increase^{20,25,43}.

Fig. 3. Model predictions of lake drainage and identification of key

environmental drivers. (A) Receiver operating characteristic (ROC) curve and (B) precision-recall (PR) curve plots for the accuracy assessment of the machine learning binary classification model used in predicting lake drainage events. The blue dashed lines represent a random guess. Recall: ability to find actual positives; Precision: accuracy of positive predictions; AUC: area under the curve; AP: average precision. A high AUC value indicates that the model has a strong ability to discriminate between positive (drained lakes) and negative (undrained lakes) samples, while a high AP value signifies high predictive quality for positive samples. (C) The relative importance of predictive variables in predicting the occurrence of lake drainage events, assessed using permutation importance and Shapley values⁸. The two methods are superimposed, with dark orange representing overlap. Both indicators identify trend slope of annual air temperature, annual air temperature and elevation as the most important predictors of lake drainage events. Kernel density estimations of Shapley value distributions for (D) trend slope of annual air temperature, (E) annual air temperature, (F) elevation, and (G) active layer depth. The density of sample distribution is normalized, with yellow indicating the most concentrated regions of data points. The magnitude of Shapley values reflects their impact on lake drainage predictions, with negative values representing lower likelihood. "

2. Another finding is with respect to the increased vegetation in drained lakes based on the changes in the NDVI. The ML approach overlaps with prior publications, but with less rigor (e.g. Webb et al. 2023: <https://www.nature.com/articles/s41558-022-01455-w#Sec8>). The description of the machine learning is not at all adequate. For e.g. the input variables, type of cross validation are not specified, nor are the hyperparameters included in the appendix. Why were 2 models fit separately for the thermokarst and non-thermokarst lakes, and not a single model considered, with thermokarst/non-thermokarst specified as a categorical variable? Were the inputs different for each model to differentiate the drivers of lake drainage for the two lake types (which is what it appears like from Fig 4)? A comprehensive list of inputs must be provided in the SI. The methods also specify that variables that didn't contribute much were removed based on the feature importance during the iterative training process. This can be problematic with redundant variables, as sometimes these can all have greater feature importance scores due to collinearity. Overall the models seem to have moderate skill ($R^2 = 0.71$). The Shapley values seem low, perhaps indicating that the variables don't have that much contribution to the NDVI, and needs to be accounted for

explaining the results in 249-256 (e.g. there is only ~0.1 difference between the ranges of volumetric SWC in the partial dependence plot). A permutation feature importance could also be done to validate the findings.

We truly appreciate your valuable suggestions and comments. In response to your feedback, we have modified the training and method description of the machine learning models. We have also provided the input list and optimal parameters in the supplementary material.

Lines 737–799: "

Machine learning models

In this study, we trained a binary classification model for predicting lake drainage (Fig. 3) and a regression model to predict NDVI within DLBs (Fig. 6). We utilized CatBoost⁸⁸, a categorical-feature-based gradient boosting method, for model training. This choice was motivated by several advantages: (a) CatBoost can effectively handle heterogeneous data comprised of both continuous and categorical variables, (b) it demonstrates resilience against outliers and noise data, and (c) it generates enhanced models with reduced bias and variance through ensemble learning, as opposed to individual tree-based models⁸⁸.

The training and testing samples for the binary classification model were derived from all drained lakes (~28,000) between 2001 and 2020 and a randomly selected 1% subset of undrained lakes (~58,000). The candidate explanatory variables for the model are detailed in table S3. For drained lakes, the climate parameters were extracted using values corresponding to their identified primary drainage year, while for undrained lakes, climate parameters were randomly sampled between the years 2001 and 2020, aiming to maximize coverage across different climate conditions. We used 70% of the samples for model training and the remaining 30% for independent testing of predictive performance. We utilized the Python-based Scikit-learn library⁸⁹ to fit CatBoost models and determined the optimal model hyperparameters through random search and ten-fold cross-validation. The initial model includes all candidate explanatory variables, which might exhibit collinearity, meaning that introducing certain variables may lead to considerably changes in the importance estimates of

other variables. To address this issue, during the iterative training process, we removed features that had negligible contributions to model performance and features highly correlated (Pearson correlation coefficient $r > 0.5$) with the most important variables for model performance. The important variables were identified through preliminary feature importance evaluations through the Shapley additive explanations model interpretability approach⁹⁰. We tested training separate models for very likely and unlikely thermokarst lakes for drainage prediction but did not observe an improvement in model accuracy. Therefore, we included thermokarst lake likelihoods as a categorical feature in the final model. The final model achieved an AUC score of 0.92 (Fig. 3A), indicating its effectiveness in distinguishing drained lakes from undrained lakes. The model's precision is 0.84, recall is 0.72, and average precision is 0.88 (Fig. 3B), demonstrating its ability to maintain high accuracy while effectively capturing drained lake samples. The remaining explanatory variables of the final model are shown in Fig. 3C, and the optimal model hyperparameters are listed in table S6.

The training process of the regression model for predicting NDVI in DLBs is similar to the binary classification model described earlier, with the distinction that the predicted variable is a continuous value between 0 and 1 (NDVI), rather than a binary value (0 or 1). Additionally, the loss function used is Root Mean Square Error (RMSE) instead of Logarithmic Loss. The training and testing samples of the model are based on the annual NDVI values of all drained lakes and the climate data of the corresponding years, recorded with reference to the year of lake drainage. In other words, the model predicts the NDVI values in DLBs for the 0–15 years after lake drainage. Just like the binary classification model, we followed a similar approach for training and testing the regression model, utilizing the CatBoost algorithm and conducting a 70-30 split of the dataset for training and independent testing. The initial model included all candidate explanatory variables (table S4), and the feature selection process was also applied to remove negligible or highly correlated variables based on Shapley values and Pearson correlation coefficients. The optimal hyperparameters for the model were determined through methods like random search

and ten-fold cross-validation (table S7). The final model's RMSE is 0.08, R^2 is 0.83, mean absolute error is 0.06, and mean bias is 0.00 (Fig. 6C), indicating that the regression model can effectively predict NDVI values in DLBs based on relevant climate and environmental variables, supporting a deeper understanding of post-drainage vegetation dynamics.

We employed two importance evaluation methods, namely Shapley values and Permutation importance, to interpret the machine learning model and quantify the contribution of each explanatory variable. The Shapley value is a contribution score for each feature, indicating its expected marginal contribution on the prediction task. The calculation method of Shapley value is to average the feature contribution of all possible feature alliances and consider all possible interaction combinations between features. It's important to note that the range of Shapley values differs between classification and regression models, making them non-comparable. Therefore, the focus should be on analyzing the importance ranking of variables within each model. The calculation of Permutation importance involves randomly altering the value of a single feature while keeping the values of the other features unchanged. The subsequent observation of how this affects the final model performance metric indicates the feature's importance. The greater the reduction in the metric after permutation, the more crucial the feature is for the model's predictive ability. Together, these two approaches provide a fair and comprehensive assessment of feature importance, taking into account both main effects and interactions."

3. Another section where I found the methodology inadequate is the one about the permafrost extent, ice content, soil carbon etc. It is unclear how the authors got to following finding "Regions with very high thermokarst lake coverage, discontinuous permafrost extent, and very high ground ice content are more susceptible to lake drainage, with densities approximately 3, 2.5, and 2 times higher than average, respectively (Fig. 1H)." It seems like this was done by overlaying many spatial layers with the drained lake areas, but that was not described in the methods. Also would it be possible to link these variables with the drainage events using the same ML model with the climate inputs (point 1 above)? There are examples

where the gradient boosted trees can be used for time series regression with dynamic (meteorological) inputs and static attributes (such as soil C %, permafrost extent etc).

We have described the methodology for calculating spatial density and drainage probability in Lines 575–580: " In this study, the spatial density of drained lakes (Fig. 1H) was calculated by overlaying the spatial layers delineated from Collected data products with the detected drained lakes, representing the number of drained lakes per unit area in different regional units. The probability of lake drainage (Fig. 4) is calculated based on the ratio of drained lakes to all lakes within a specific region unit. These methods enable us to visually depict the spatial distribution pattern and occurrence probability of lake drainage events across various regions."

In addition to modelling lake drainage with the ML method as suggested, we also calculated the probability of lake drainage.

Lines 224–286: "

Statistical analysis of lake drainage probabilities

Since attributes such as permafrost extent, ground ice content, thermokarst lake likelihoods, and whether Yedoma region are considered time-invariant categorical features in the machine learning model, their relative importance may be underestimated. Given the known importance of these landscape variables in driving surface water changes associated with permafrost degradation, we conducted an enumeration of all lake objects within our study area and calculated lake drainage probabilities for various sizes of lakes and geographical regions (Fig. 4 and table S2). A total of approximately 5.83 million lakes (with an area larger than 1 ha) were identified across the northern permafrost region. The overall average drainage probability for lakes during the study period was 0.61%, while the drainage probabilities were 0.64% for small lakes, 0.50% for medium lakes, and 0.38% for large lakes, indicating a relatively lower likelihood of drainage for larger lakes. One potential explanation for this is that smaller lakes, due to their shallower water columns and lower water storage capacity²⁹, are more susceptible to lateral drainage events triggered by extreme precipitation or rapid snowmelt. In contrast, larger lakes

have the capacity to accumulate more heat and water, thus showing greater resilience against hydrological disturbances²⁸. This provides a thermal inertia and water storage buffer when temperatures and precipitation fluctuate, enabling larger lakes to better maintain thermal equilibrium and shoreline stability compared to smaller water bodies.

Significant spatial heterogeneity in lake drainage probabilities has been observed across ecoregions (table S2), highlighting the complexity of lake drainage dynamics. For instance, within the Canadian Aspen forests and parklands (spanning 1.6×10^4 km²), 16.33% of lakes experienced drainage, while in the Central Canadian Shield forests (spanning 16.5×10^4 km²), only 0.11% of lakes underwent drainage. This diverse distribution of lake drainage probabilities underscores the intricate interplay of local environmental factors. Further analysis indicates that lake drainage probabilities are associated with thermokarst lake likelihoods, permafrost extent, and ground ice content (Fig. 4). Notably, for very likely thermokarst lakes and the Yedoma region, the drainage probabilities for lakes of all sizes exceed the regional averages, with large lakes anomalously having higher drainage probabilities than small and medium-sized lakes. Large drained lakes are disproportionately concentrated in these areas, with 6–8 times the average density (Fig. 1H) and 3–4 times the average drainage possibility (Fig. 4B&D), suggesting well-developed erosional drainage systems may have formed. The prevalent landscape in these areas is a mosaic of lakes and streams, interconnected through an underground network of ice^{15,42}. When expanding lakes encounter the ice network, thermal erosion along the network results in the melting of ice wedges, creating drainage channels through which the lake water drains^{7,34}.

The drainage probabilities for likely thermokarst lakes fall between those for very likely and unlikely thermokarst lakes (Fig. 4D), consistent with existing knowledge that thermokarst lakes are more susceptible to drainage⁴. The impact of ground ice content on lake drainage probabilities is relatively intricate (Fig. 4F). In regions with medium ground ice content, the overall lake drainage probability is the highest, approximately 1.4 times the average level. In regions with high and low ground ice

content, the overall lake drainage probabilities are slightly below average, while the drainage probabilities for large lakes in these regions are 1.6 and 0.5 times the average, respectively. Variations in the probability of lake drainage under different ground ice content levels may be associated with the thermal inertia of latent heat fusion and the formation of subsurface drainage channels^{28,50}. Considering that ground ice content dataset covering the northern permafrost region have not been updated for over 20 years, we need to be mindful of the timeliness of the data and its impact on the current research findings. In future studies, updating ground ice content data will contribute to a more accurate assessment of the relationship between ground ice content and lake drainage probabilities⁴.

Furthermore, the discontinuous permafrost zone exhibits the highest overall lake drainage probabilities (more than double the average; Fig. 4H), with drainage probabilities for small, medium, and large lakes at 1.33%, 1.28%, and 0.78% respectively. There are over 13,000 drained lakes in the continuous permafrost zone (Fig. 4G), but the likelihood of lake drainage in this area is lower than the average level (Fig. 4H). However, the susceptibility is rising in certain southern areas where permafrost is becoming more discontinuous due to climate warming^{9,20}.

Fig. 4. Drainage probability analysis of lakes across various lake sizes and regions. Analysis of drained lake area distribution and drainage probabilities for lakes of different sizes in (A, B) the Yedoma region and the entire study area, along with different classifications of (C, D) thermokarst lake likelihoods, (E, F) ground ice contents, and (G, H) permafrost extents. The left-hand panels depict kernel density estimation plots with a logarithmic x-axis to display lake areas, along with upper

boxplots presenting statistical information and numerical annotations denoting the number of drained lakes. The right-hand panels present radar charts illustrating the percentage of drained lakes relative to the total number of lakes within the respective regions."

In general, given the overlap in the findings/methodology with prior literature, I felt further analysis based on some hypotheses would help augment the novelty of the manuscript. For e.g. further time series analysis of the meteorological data may explain what triggered the sudden drainage event in 2017 shown in figure S9, and more broadly the periods of maximum loss in other lakes. Similarly, some exploration into the spatial variability, for e.g. why eastern Canada is different from other regions (lines 243-245, S8 or other regions in Table S1) would be useful. How was it determined that this was "due to differences in the physiological traits of plant species adapted to these regions". Lines 288-303 came across as suggesting that increased vegetation in drained lakes was a positive outcome in terms of acting as a net carbon sink, mitigating emissions and stabilizing sediments, but the negative consequences of such dramatic land cover change could also be discussed.

We extend our gratitude for your invaluable suggestions, which have greatly contributed to enhancing the novelty of our manuscript.

In response to further time series analysis of the meteorological data may explain what triggered the sudden drainage event in 2017 shown in figure S9, and more broadly the periods of maximum loss in other lakes.

Lines 440–455: " Our dataset of 35,337 detected lake drainage events can serve as a starting point for further research, such as investigating potential catastrophic flooding in DLBs^{60,61} and improving simulations of permafrost hydrology linkages in Earth system models^{20,62}. Analyses of mega-lake drainage events and drained lake clusters allow for more specific investigation of the role of climate change in triggering lake drainage. For example, the largest drained lake identified in the study area, with an area of approximately 6,000 ha, is Ozero Maloye Yeravnoye Lake located in the discontinuous permafrost zone in southern Russia. This shallow lake, with an average depth of only 1.8 meters, experienced drainage between 2014 and 2017 (Fig. S9).

During the winter of 2013, there was an abnormally high amount of snowfall in the lake area, setting a record for the highest snowfall in nearly 20 years. An abnormally thick snowpack would not only hinder the refreezing of the active layer at the lake bottom, leading to talik development and potential drainage channel formation, but would also increase snowmelt, intensifying the possibility of bank overtopping^{34,43}. In addition, approximately 60% of lake drainage events observed on St. Lawrence Island (Fig. 1G) have occurred since 2018, a period characterized by historically low sea ice coverage⁶³ and widespread seabird mortality⁶⁴, indicating that the local climatic conditions may have reached a tipping point."

In response to *why eastern Canada is different from other regions (lines 243-245, S8 or other regions in Table S1) would be useful.*

Lines 348–354: "The NDVI of DLBs was noticeably lower in Canada, which may be attributed to the thin and rocky soils in the Canadian Shield region⁵⁶. These soil conditions could pose drainage challenges, which may lead to increased waterlogging and periodic inundation of vegetation, ultimately resulting in a lower NDVI within DLBs compared to the surrounding areas. Additionally, the proportion of small drained lakes is higher in Canada compared to other regions, potentially contributing to less lush vegetation in Canadian DLBs (table S2)."

In response to *but the negative consequences of such dramatic land cover change could also be discussed.*

Lines 456–464: " Lake drainage is essentially the abrupt thawing of permafrost intensified by climate change, reflecting permafrost degradation and instability. Drainage events reduce the water storage capacity of lakes, impacting local hydrological conditions^{65–67}. Drainage of large lakes often leads to catastrophic flooding due to the peak of snowmelt promoting the formation of ephemeral lakes and resulting in rapid and sustained flood peaks^{4,61,68}. Such hydrological disasters can adversely affect infrastructure like Arctic roads and pipelines. Species dependent on lake habitats for survival, such as migratory birds and aquatic life, are threatened⁶⁹.

As more lakes are drained, access to clean freshwater may become even more challenging for many Arctic communities and indigenous populations⁷⁰. "

Finally, I would strongly recommend that the code be made public for this work to be evaluated and reproduced.

Thank you for bringing this to our attention. We have shared the code to facilitate evaluation.

Minor comments

Introduction Line 40-43 – the general reader would benefit from an explanation of the difference between thermokarst and non-thermokarst lakes, and why the distinction is necessary. You could consider moving lines 75-79 up into the introduction.

Thank you for your valuable suggestion. We have implemented the recommended changes accordingly.

Lines 52–64: "Lakes across the northern permafrost region can be divided into thermokarst and non-thermokarst lakes, depending on their origin^{4,32}. Thermokarst lakes develop in ice-rich permafrost zones through the thawing of sediments, melting of ice wedges, and localized ground subsidence^{15,33,34}, while non-thermokarst lakes form by accumulating water in pre-existing topographic depressions without significant thaw-induced subsidence. Compared to non-thermokarst lakes surrounded by ice-poor permafrost, thermokarst lakes usually undergo faster lateral erosion and bottom talik (unfrozen ground) development^{31,35}, making them more likely to drain⁴. Regional studies have shown that after lake drainage, the tundra vegetation growing in the moist and nutrient-rich sediments of thermokarst DLBs may be more luxuriant than in surrounding areas^{16,17}. Hence, distinguishing between thermokarst and non-thermokarst lakes is essential to understand their distinct drainage processes, evolution trajectories and vegetation dynamics, as well as to investigate the differential impacts of climate change on them."

Fig S1 – if Olsen et al. 2001 is the source for the ecoregions, it must be cited.

Yes. Thank you for pointing that out. We apologize for the oversight. We have now included citations to acknowledge the sources of fig. S4 & S5.

Fig S6 It would be useful to have the figure labels (A, B, etc.) mentioned in the caption
This figure was removed due to a change in the analysis.

This is an example where I wondered whether the ERA5 resolution would allow clear delineation of thermokarst vs. non thermokarst lakes.

Yes, according to our Fig. S10, the resolution of ERA5-Land is sufficient to clearly delineate thermokarst and non-thermokarst lakes.

Once again, we express our heartfelt gratitude for your valuable feedback and insights.

REVIEWER COMMENTS

Reviewer #1 (Remarks to the Author):

I thank the authors for their careful revisions and responses to my comments. Everything was addressed. Below are just a few minor textual comments and one methodological point that unfortunately I missed in the first round.

65-74: State also where the thermokarst classification is from, follows on it being "essential" in the previous sentence.

483: At first I thought the median extraction was a leftover from the previous version. Make it more clear that this is from the annual 90th percentile time-series.

525: "This smoothing technique introduces the possibility of misclassifying "water loss" as a "wet period" (land-water-land)." How so?

613: Did you use AWEI for anything? It is only mentioned here. Provide how you used it or if you tried to use it for something and it didn't work, that would also be interesting. Else remove this reference.

622: "JRC and GLAD" instead.

692-99: suggest making it more explicit that these operations are done on the 90th percentile time-series and done annually.

696: A standard-sized buffer from the lake edge seems much more appropriate as lakes are irregularly shaped and for large lakes there can be significant land cover, even land use, transitions within such a large buffer. For instance, the largest drained lake was 6000ha, resulting in a buffer of 4.4km beyond the lake edge. A standard buffer of some distance, would reduce these transitions and make them more equivalent between different lake size classes. I do not know what would be appropriate, perhaps something in the 100-500m range. I also do not know to what extent the results of the different lake sizes would or would not be affected.

711: The length of degrees varies by latitude. 0.1° of longitude is equivalent to 7.2km at 50N and 2.9km at 75N with 0.1° of latitude equivalent to 11 km for both (<http://www.csgnetwork.com/degreenllavcalc.html>)

The revised manuscript “Tracing Arctic lake drainage events and emergent vegetation in drained lake basins” is much improved from the previous version, and it is clear that the authors spent a lot of effort addressing reviewer comments. Specifically, the authors added the analyses I suggested on drained lake density and improved their analysis of the environmental drivers of lake drainage. They have also addressed other important concerns I raised such as the differentiation between thermokarst and non-thermokarst lakes.

I still find the manuscript text to be heavy on the presentation of results and light on their interpretation. If this is a space issue, one suggestion to ameliorate is to move some of the results from the text into tables. Perhaps relatedly, while the authors have improved their citations, I still found ideas and sentences where previous work needs to be cited.

One issue that I found confusing is that the authors use two different metrics to calculate drained lake density, but this differentiation is not explained until the Methods (lines 575-579). I think if the difference between ‘spatial density’ and ‘probability of lake drainage’ is defined much earlier in the manuscript, this would clear up confusion. Similarly, since density could be calculated on an area basis (drained lakes/unit area) or lake basis (drained lakes/number of lakes in a region), I suggest the authors always specify ‘spatial density’ rather than simply ‘density’ to aid reader comprehension. Calling the other metric ‘lake-wise density’ or something similar rather than ‘probability of lake drainage’ might also be helpful.

The Zenodo repository with the drained lake dataset link is broken.

Line-by-line comments

- 1 Did the authors mean ‘tracking’ instead of ‘tracing’?
- 84 conversion info between ha and m² seems unnecessary
- 85 ‘in area’ should be ‘of original lake area.’
- 89 are there more drained lakes in coastal lowlands/river deltas, or just more lakes there in general? If the authors are going to make a statement about the spatial clustering of DLBs, they should also provide an analysis backing up this claim.
- 101 what is the ‘lake rich zone?’
- 105 what does ‘regional’ mean in this context?
- 150 $p=0.08$ could be considered significant in some contexts. I suggest rewording this sentence to: ‘the lake drainage frequency exhibited a slight upward trend (slope of 23; $p=0.08$) throughout the study period.’
- 164 Citing some studies that do track non-thermokarst lake dynamic would be appropriate here
- 176 Figure title should be ‘2001-2020’ rather than ‘2021-2020’
- 186 I suggest reporting model performance statistics here
- 188 Either use ‘trend’ or ‘slope’ but not both. Additionally, this sentence should be reworded, as it currently implies that the trend in elevation was an explanatory variable.
- 191 What is the difference between ‘promoting lake drainage’ and ‘having an overall negative impact.’ (what is ‘negative impact’ in this context?)
- 200 Perhaps this is just because there are more lakes in lowland regions
- 204 This could also be due to increased erosion and other thermal processes. Why would taliks be the only permafrost process affected by deeper active layer depth?

- 225 I do not understand this sentence. Why would the ML model underestimate the importance of categorical features?
- 236-237 This sentence needs a citation.
- 268 Just because a dataset is old does not make it wrong or outdated. Similarly, a new dataset is not good just because it is new.
- 297 suggested edit to this sentence: ‘taking into account the potential modulation of NDVI by standing water,...’
- 337 what is ‘drainage ratio’?
- 350 I would have thought rocky conditions would promote drainage. Please provide a citation for the idea that rocky soils would lead to more water logging.
- 356 I do not understand the point the authors are trying to make here. What does it mean for microtopography of DLBs to lead to non-significant differences between flood prone and non-flood prone regions?
- 362 Is the underlying assumption here that yedoma soils have more C and N? If so, a citation is needed. If not, clarification (and a citation) is needed.
- 374 delete ‘further’
- 379 delete ‘indeed’
- 391 consider re-wording to ‘.in regions where summer air temperatures and annual soil temperatures are increasing.’
- 394 yes, but the trend in annual air temperature also explains a good deal of the variation, and this implies that increasing temps benefit plant growth.
- 404 How do the authors know this is not easily observable through conventional statistical analyses? I suggest deleting this.
- 428 As a reader, I want to know about the implications of the study’s findings in the concluding section. These sentences instead reiterate methods and model performance. I suggest deleting.
- 444 Since this is not a study of one specific lake, it is out of place to discuss one lake in the conclusions, especially since this study contains no direct measurements of the lake other than its drainage date (i.e., everything else discussed here came from a different source - not cited - and is speculative). I suggest deleting.
- 457 citation needed
- 460 replace ‘disasters’ with ‘events’
- 576 collected need not be capitalized
- 662 I do not understand what the authors did here. From the text, it seems the authors are evaluating the accuracy of existing data sets, but I am not sure why that would be necessary. It would be more pertinent to use existing datasets to evaluate the false negative rate of the author’s data set.

Figures:

Figure 2: I suggest using a different color scheme for panels showing TK probabilities, ground ice content, and permafrost extent

Figure 3: Model diagnostic panels could be moved to the supplementary material.

Reviewer #3 (Remarks to the Author):

The authors have done a thorough job of addressing prior review comments. I only have some minor additional comments based on my prior review

Choice of ERA-5:

The authors justify the choice of ERA-5 as a dataset, by pointing out that primarily most lakes occur in one grid cell, and that the ERA-5 data are comparable to the high resolution Daymet dataset. There are a couple of caveats to this that need to be acknowledged

1) Even if there is only 1 lake per grid cell, the averaging that results from using a coarse 80 sq km dataset applied to an ~ 1 sq km (or less) area will result in uncertainty in the modeling/relations determined between the climate drivers and the changes seen for a particular lake.

2) While the ERA-5 and Daymet data are consistent, there is significant scatter. Particularly the differences in precipitation are notable (+/- 250 to 500 mm – the scales are hard to see in Fig S11), which is significant given that median precip is ~ 500 mm in the region.

This doesn't have to change the analysis or the results, but some statements regarding the uncertainties introduced by the choice of ERA-5 should be included. The terms "strong consistency and low variability" might be too strong given the scatter in Fig S11. If the authors wanted to make a definitive statement, a statistical test such as a Mann-Whitney or Kolmogorov-Smirnov test can be used to compare the two distributions instead of a visual comparison.

The description of the ML models are now sufficient, and the new classification model to determine between climate drivers and lake drainage will occur is interesting. I would still suggest being careful about overinterpreting highly correlated features (delta air temperature vs. air temperature) and trying to distinguish between the two as in the statement:

(e.g . Further feature importance assessment indicates that trend slope of annual air temperature, annual air temperature and elevation are the most important drivers of lake drainage events (Fig. 3C).)

It would suffice to say that air temperature and elevation are important drivers of drainage events.

I. Response to comments and suggestions of Reviewer #1

I thank the authors for their careful revisions and responses to my comments. Everything was addressed. Below are just a few minor textual comments and one methodological point that unfortunately I missed in the first round.

We appreciate your positive feedback regarding our revisions and responses to your comments. We're pleased to hear that we were able to address all your concerns. Thank you for your continued support.

65-74: State also where the thermokarst classification is from, follows on it being "essential" in the previous sentence.

Thanks to the suggestion, we have added lines 66–68: "We distinguished between thermokarst lakes and non-thermokarst lakes based on the published thermokarst lake coverage dataset³³."

483: At first I thought the median extraction was a leftover from the previous version. Make it more clear that this is from the annual 90th percentile time-series.

Yes. Thanks for pointing this out, we have changed it to: "Subsequently, we conducted median extraction of the annual 90th percentile NDVI time series after masking water bodies within DLBs and their surrounding areas."

525: "This smoothing technique introduces the possibility of misclassifying "water loss" as a "wet period" (land-water-land)." How so?

We have added explanations and a supplementary figure.

This smoothing technique introduces the possibility of misclassifying "water loss" as a "wet period" (land-water-land) due to its tendency to smooth out fluctuations in the open water percentage, potentially leading to the omission of detecting lake drainage events (fig. S10).

Fig. S11. Example of drainage pixels being misclassified in the GLAD dataset. (A–D) Landsat color-infrared images (NIR-R-G) captured at 161°16' W, 69°52' N in the years 2000, 2008, 2015 and 2020. (E) the GLAD water dynamics map and (F) the JRC water transition map. The smoothing technique used in the GLAD dataset resulted in the misclassification of "water loss" as "wet period" (land-water-land).

613: Did you use AWEI for anything? It is only mentioned here. Provide how you used it or if you tried to use it for something and it didn't work, that would also be interesting. Else remove this reference.

Thanks for pointing this out, we have added: "We employed the annual time series of TCG and NDVI to analyze post-drainage vegetation dynamics and utilized AWEI time series as the basis for lake drainage year detection."

622: "JRC and GLAD" instead.

Modified as suggested.

692-99: suggest making it more explicit that these operations are done on the 90th percentile time-series and done annually.

Thanks to the suggestion, we have revised it to: "Based on the generated 90th percentile NDVI annual time series, we pixel-wise acquired NDVI values within the lake basin area for each drained lake, masked out values below 0, and computed the median NDVI for the region."

696: A standard-sized buffer from the lake edge seems much more appropriate as lakes are irregularly shaped and for large lakes there can be significant land cover, even land use, transitions within such a large buffer. For instance, the largest drained lake was 6000ha, resulting in a buffer of 4.4km beyond the lake edge. A standard buffer of some distance, would reduce these transitions and make them more equivalent between different lake size classes. I do not know what would be appropriate, perhaps something in the 100-500m range. I also do not know to what extent the results of the different lake sizes would or would not be affected.

Thank you for your feedback. We tested the impact of buffer size on NDVI extraction from surrounding vegetation before the initial submission of the manuscript. We chose to use variable buffer zone instead of fixed sizes primarily due to concerns about potential inaccuracies in lake boundary identification. The use of small fixed buffer zones (e.g., 100 meters) may result in an underestimation of the greenness level of the surrounding vegetation. While larger buffer zones may include a wide range of land cover and transitions, the same large drained lakes might also exhibit diversity in land cover and transitions within them. In addition, we employed median extraction on the annual time series of the 90th percentile NDVI within both the drained lake basins and surrounding areas to obtain statistically stable results.

711: The length of degrees varies by latitude. 0.1° of longitude is equivalent to 7.2km at 50N and 2.9km at 75N with 0.1° of latitude equivalent to 11 km for both (<http://www.csgnetwork.com/degreenllavcalc.html>)

Yes, thanks for pointing that out. Muñoz-Sabater et al. (2021) used "9 km" and "0.1°" to describe the spatial resolution of the ERA5-Land, whereas in the GEE platform its resolution is labelled as "11132 m".

We have therefore modified it to: "with a spatial resolution of 0.1° (resampled to ~11km in the GEE platform)".

Thank you again for your valuable comments.

II. Response to comments and suggestions of Reviewer #2

The revised manuscript “Tracing Arctic lake drainage events and emergent vegetation in drained lake basins” is much improved from the previous version, and it is clear that the authors spent a lot of effort addressing reviewer comments. Specifically, the authors added the analyses I suggested on drained lake density and improved their analysis of the environmental drivers of lake drainage. They have also addressed other important concerns I raised such as the differentiation between thermokarst and non-thermokarst lakes.

Thank you for your valuable feedback on our manuscript. Your guidance is pivotal in elevating the quality of our research. We appreciate your acknowledgment of the improvements we've made in response to your suggestions.

I still find the manuscript text to be heavy on the presentation of results and light on their interpretation. If this is a space issue, one suggestion to ameliorate is to move some of the results from the text into tables. Perhaps relatedly, while the authors have improved their citations, I still found ideas and sentences where previous work needs to be cited.

We appreciate your feedback and have taken into consideration your comments regarding the interpretation of results and citation issues.

Regarding the comment about being "heavy on the presentation of results and light on their interpretation", it may be because our presentation and interpretation of results are intermixed. We have made revisions to enhance the overall balance between presentation and interpretation.

For example, we have added lines 89–95: "The distribution of drained lakes appears to exhibit spatial clustering, with a concentration in coastal lowlands and river delta areas (figs. S1–S3). These are generally lake-rich regions, as lakes and DLBs account for 21% of the northern permafrost regions and 49% of the lowland permafrost regions^{4,33}. Typically, lake drainage is thought to be dominated by thermokarst lake processes⁴, which cover about 20% of the continuous permafrost zone³³. However, we have found that only about half of the drained lakes are

situated within thermokarst lake landscapes (Fig. 1D), with the other half having non-thermokarst origins."

Lines 99–101: "Surface water in permafrost regions is critical to northern communities, with lakes and DLBs being a focus of local agriculture, livestock, and industrial activities⁴. Therefore, exploring the causes of drainage in large lakes..."

Lines 184–192: "Permafrost acts as a barrier to water exchange between surface and groundwater systems. In continuous permafrost regions, suprapermafrost groundwater often accumulates above the permafrost layer within the active layer, including closed subaerial and open subaqueous taliks⁴. In discontinuous permafrost regions, where permafrost continuity is lacking, direct connections between surface and groundwater systems can occur, influencing the hydrological dynamics of lakes^{24,44}. Therefore, lake drainage events are influenced by both the long-term legacy of permafrost characteristics, such as permafrost properties, distribution, and degradation extent, as well as short-term environmental features, including active layer dynamics, ground thermal conditions, and variations in water supply⁴."

Lines 215–219: "Development of terrestrial taliks is likely to result in widespread lake drainage when mean annual air temperature approaches 0 °C⁴³. In addition to promoting lake drainage, this warmer regime may adversely affect permafrost aggradation on newly exposed surfaces, inhibit ground ice accumulation, and bring about substantial changes in the landscape conditions of the DLB system⁴."

Lines 419–424: "Additionally, due to permafrost aggradation and ice wedge growth, DLBs exhibit a highly heterogeneous geomorphic mosaic characterized by fine-scale variations in topography²². The micro-topography of DLBs effectively directs runoff, facilitates nutrient accumulation, and supports vegetation growth within DLBs^{17,22}. Simultaneously, it fosters diverse tundra ecosystems within DLBs, making them ecological hotspots in the permafrost regions^{4,16}."

Additionally, we have supplemented the citations as suggested.

One issue that I found confusing is that the authors use two different metrics to calculate drained lake density, but this differentiation is not explained until the Methods (lines

575-579). I think if the difference between 'spatial density' and 'probability of lake drainage' is defined much earlier in the manuscript, this would clear up confusion. Similarly, since density could be calculated on an area basis (drained lakes/unit area) or lake basis (drained lakes/number of lakes in a region), I suggest the authors always specify 'spatial density' rather than simply 'density' to aid reader comprehension. Calling the other metric 'lake-wise density' or something similar rather than 'probability of lake drainage' might also be helpful.

Thanks. Your input is very valuable. We acknowledge the potential confusion this may have caused and appreciate your suggestion for clarification.

To address this concern, we have adhered to your recommendation by consistently employing the terms "spatial density" and "lake-wise density" to differentiate these metrics and facilitate reader comprehension.

Additionally, we have provided an explanation: "We used two metrics to calculate the density of drained lakes: spatial density (number of drained lakes per unit area) and lake-wise density (calculated as the ratio of drained lakes to the total number of lakes in the region). These metrics aim to illustrate the spatial distribution patterns and likelihood of lake drainage events in different regions."

The Zenodo repository with the drained lake dataset link is broken.

Apologies for the line breaks associated with PDF conversion, we have added hyperlinks to the manuscript.

Line-by-line comments

1 Did the authors mean 'tracking' instead of 'tracing'?

Yes, we do track and monitor, not trace. Thanks a lot for the correction.

84 conversion info between ha and m2 seems unnecessary

Removed as per comment.

85 'in area' should be 'of original lake area.'

Done as suggested.

89 are there more drained lakes in coastal lowlands/river deltas, or just more lakes there

in general? If the authors are going to make a statement about the spatial clustering of DLBs, they should also provide an analysis backing up this claim.

We have revised it to: "The distribution of drained lakes appears to exhibit spatial clustering, with a concentration in coastal lowlands and river delta areas (figs. S1–S3). These are generally lake-rich regions, as lakes and DLBs account for 21% of the northern permafrost regions and 49% of the lowland permafrost regions. Typically, lake drainage is thought to be dominated by thermokarst lake processes, which cover about 20% of the continuous permafrost zone. However, we have found that only about half of the drained lakes are situated within thermokarst lake landscapes (Fig. 1D), with the other half having non-thermokarst origins."

The analysis of drained lake density in coastal lowlands can be found in fig. S8.

101 what is the 'lake rich zone?'

We have modified this to "lake-rich areas" and added a definition in line 25: "Arctic lake-rich areas (areas with at least 5% lake coverage)".

105 what does 'regional' mean in this context?

We have modified this to "study area average".

150 $p=0.08$ could be considered significant in some contexts. I suggest rewording this sentence to: 'the lake drainage frequency exhibited a slight upward trend (slope of 23; $p=0.08$) throughout the study period.'

Done as suggested.

164 Citing some studies that do track non-thermokarst lake dynamic would be appropriate here

Thanks for the suggestion. Done as suggested.

176 Figure title should be '2001-2020' rather than '2021-2020'

Thanks for the correction.

186 I suggest reporting model performance statistics here

Thanks for the suggestion. We have revised it to: "We conducted a diagnostic evaluation of the lake drainage prediction model and obtained an area under the curve (AUC) value of 0.92 and an average precision (AP) value of 0.88 (fig. S7). The results showed that the model has a strong classification ability and can effectively

use explanatory variables to predict whether or not individual lakes will be drained."

188 Either use 'trend' or 'slope' but not both. Additionally, this sentence should be reworded, as it currently implies that the trend in elevation was an explanatory variable.

Thanks for the correction. We have replaced the "trend slope" with "slope" where it appeared in the manuscript. We have also rewritten the sentence: "Further feature importance assessment indicates that air temperature and elevation are important drivers of lake drainage events (Fig. 3A)."

191 What is the difference between 'promoting lake drainage' and 'having an overall negative impact.' (what is 'negative impact' in this context?)

We have revised it to: "...with the primary clusters between 0–0.04 °C/year inhibiting lake drainage, and secondary clusters near 0.1 °C/year promoting lake drainage."

200 Perhaps this is just because there are more lakes in lowland regions

We have revised it to: "...with both spatial and lake-wise densities of drained lakes exceeding the average level (fig. S8)."

204 This could also be due to increased erosion and other thermal processes. Why would taliks be the only permafrost process affected by deeper active layer depth?

Yes. We have revised it to: "This may be attributed to increased erosion and other thermal processes, such as the formation of taliks at the lake bottom^{48,49}."

225 I do not understand this sentence. Why would the ML model underestimate the importance of categorical features?

We have revised it to: "For categorical features such as permafrost extent, ground ice content, and thermokarst lake likelihood, their relative importance may be underestimated due to the fact that they are treated as time-invariant attributes and the imbalanced distribution of categories."

236-237 This sentence needs a citation.

Done as suggested.

268 Just because a dataset is old does not make it wrong or outdated. Similarly, a new dataset is not good just because it is new.

Agreed. We have revised it to: "Considering that the ground ice content dataset covering the northern permafrost region has not been updated for over 20 years, and

during this period, the permafrost region has undergone significant warming, we need to exercise caution regarding the timeliness of the data and its potential impact on the current research findings."

297 suggested edit to this sentence: 'taking into account the potential modulation of NDVI by standing water,...'"

Modified as suggested.

337 what is 'drainage ratio'?

We have revised it to: "drainage area ratio".

350 I would have thought rocky conditions would promote drainage. Please provide a citation for the idea that rocky soils would lead to more water logging.

We have added a citation.

356 I do not understand the point the authors are trying to make here. What does it mean for microtopography of DLBs to lead to non-significant differences between flood prone and non-flood prone regions?

We have deleted the sentence. And we have added in lines 419–424: "Additionally, due to permafrost aggradation and ice wedge growth, DLBs exhibit a highly heterogeneous geomorphic mosaic characterized by fine-scale variations in topography²². The micro-topography of DLBs effectively directs runoff, facilitates nutrient accumulation, and supports vegetation growth within DLBs^{17,22}. Simultaneously, it fosters diverse tundra ecosystems within DLBs, making them ecological hotspots in the permafrost regions^{4,16}."

362 Is the underlying assumption here that yedoma soils have more C and N? If so, a citation is needed. If not, clarification (and a citation) is needed.

Yes. Cited as suggested.

374 delete 'further'

Done as suggested.

379 delete 'indeed'

Done as suggested.

391 consider re-wording to '..in regions where summer air temperatures and annual soil temperatures are increasing.'

Thanks for the suggestion. Revised as suggested.

394 yes, but the trend in annual air temperature also explains a good deal of the variation, and this implies that increasing temps benefit plant growth.

Agreed. We have revised this to: "The relationship between summer air temperature and NDVI predictions displays an almost monotonically increasing trend, suggesting that warmer locations tend to show enhanced plant growth."

404 How do the authors know this is not easily observable through conventional statistical analyses? I suggest deleting this.

Deleted as per comment.

428 As a reader, I want to know about the implications of the study's findings in the concluding section. These sentences instead reiterate methods and model performance. I suggest deleting.

Deleted as per comment.

444 Since this is not a study of one specific lake, it is out of place to discuss one lake in the conclusions, especially since this study contains no direct measurements of the lake other than its drainage date (i.e., everything else discussed here came from a different source - not cited - and is speculative). I suggest deleting.

Deleted as per comment.

457 citation needed

Done as suggested.

460 replace 'disasters' with 'events'

Done as suggested.

576 collected need not be capitalized

Done as suggested.

662 I do not understand what the authors did here. From the text, it seems the authors are evaluating the accuracy of existing data sets, but I am not sure why that would be necessary. It would be more pertinent to use existing datasets to evaluate the false negative rate of the author's data set.

Yes, the reviewer is right, a comparative assessment is not necessary here.

Since the existing datasets do not detect drained lakes as comprehensively as our

detections (which is not only dependent on algorithm accuracy but also on the study period), it is not possible to provide a false-negative rate test.

We have removed this sentence.

Figures:

Figure 2: I suggest using a different color scheme for panels showing TK probabilities, ground ice content, and permafrost extent

Thanks to the suggestion, we have updated the color scheme of Figure 2 to avoid confusion.

Figure 3: Model diagnostic panels could be moved to the supplementary material.

Thanks to the suggestion, we have moved the model diagnostic panels from Figures 3 and 6 to the supplementary material.

Thanks again to the Reviewer for providing us with valuable comments and suggestions.

III. Response to comments and suggestions of Reviewer #3

The authors have done a thorough job of addressing prior review comments. I only have some minor additional comments based on my prior review

Thank you for acknowledging our efforts in addressing the previous review comments. We appreciate your continued engagement with our manuscript. We have further improved our work based on your comments.

Choice of ERA-5:

The authors justify the choice of ERA-5 as a dataset, by pointing out that primarily most lakes occur in one grid cell, and that the ERA-5 data are comparable to the high resolution Daymet dataset. There are a couple of caveats to this that need to be acknowledged

1) Even if there is only 1 lake per grid cell, the averaging that results from using a coarse 80 sq km dataset applied to an ~1sq km (or less) area will result in uncertainty in the modeling/relations determined between the climate drivers and the changes seen for a particular lake.

2) While the ERA-5 and Daymet data are consistent, there is significant scatter. Particularly the differences in precipitation are notable (+/- 250 to 500 mm – the scales are hard to see in Fig S11), which is significant given that median precip is ~500 mm in the region.

This doesn't have to change the analysis or the results, but some statements regarding the uncertainties introduced by the choice of ERA-5 should be included. The terms "strong consistency and low variability" might be too strong given the scatter in Fig S11. If the authors wanted to make a definitive statement, a statistical test such as a Mann-Whitney or Kolmogorov-Smirnov test can be used to compare the two distributions instead of a visual comparison.

Thanks for the comments. We agree with what the reviewers have mentioned.

We have reworded the terms "strong consistency and low variability" to "consistency and relatively low variability".

We have also added a statement about the uncertainties introduced by the choice of ERA5-Land as suggested: "It should be noted that utilizing coarse-resolution

ERA5-Land data introduces uncertainties into the modeling and relationships established between climate drivers and observed changes for individual lakes."

The description of the ML models are now sufficient, and the new classification model to determine between climate drivers and lake drainage will occur is interesting. I would still suggest being careful about overinterpreting highly correlated features (delta air temperature vs. air temperature) and trying to distinguish between the two as in the statement:

(e.g . Further feature importance assessment indicates that trend slope of annual air temperature, annual air temperature and elevation are the most important drivers of lake drainage events (Fig. 3C).)

It would suffice to say that air temperature and elevation are important drivers of drainage events.

We appreciate your acknowledgment of the improvements and the novelty of our machine learning models. Regarding your suggestion to avoid overinterpreting highly correlated features, we understand your concern.

To address this, we have revised the statement as suggested: "Further feature importance assessment indicates that air temperature and elevation are important drivers of lake drainage events (Fig. 3A)."

Once again, we would like to express our gratitude for your valuable comments.

REVIEWERS' COMMENTS

Reviewer #1 (Remarks to the Author):

Thank you for addressing all of my concerns. However, I want to clarify one point. The smoothing technique used in the GLAD product doesn't cause "water loss" to be confused as "wet period". If anything, this smoothing would cause the opposite effect. The example shown in fig S11 is a case where there was only one clear observation in 1999 and it was marked as land (see time-series graph: <https://glad.earthengine.app/view/surface-water-dynamics#lon=-161.27084909393497;lat=69.873842208242;zoom=4;timeseries=1;>). The lake area was very shallow or turbid in 1999 and 2000 as can also be seen in the figure. Please remove this sentence and figure. Thanks!

I. Response to comments and suggestions of Reviewer #1

Thank you for addressing all of my concerns. However, I want to clarify one point. The smoothing technique used in the GLAD product doesn't cause "water loss" to be confused as "wet period". If anything, this smoothing would cause the opposite effect. The example shown in fig S11 is a case where there was only one clear observation in 1999 and it was marked as land (see time-series graph: <https://glad.earthengine.app/view/surface-water-dynamics#lon=-161.27084909393497;lat=69.873842208242;zoom=4;timeseries=1;>). The lake area was very shallow or turbid in 1999 and 2000 as can also be seen in the figure. Please remove this sentence and figure. Thanks!

Yes. Thank you very much for your feedback.

We have removed the sentence and fig. S11 as suggested.